# Convergence Analysis of Natural Gradient Descent for Over-parameterized Physics-Informed Neural Networks

## Abstract

In the context of over-parameterization, there is a line of work demonstrating that randomly initialized (stochastic) gradient descent (GD) converges to a globally optimal solution at a linear convergence rate for the quadratic loss function. However, the learning rate of GD for training two-layer neural networks exhibits poor dependence on the sample size and the Gram matrix, leading to a slow training process. In this paper, we show that for training two-layer ReLU[3] Physics-Informed Neural Networks (PINNs), the learning rate can be improved from $\mathcal{O}(\lambda_0)$ to $\mathcal{O}(1/\|\boldsymbol{H}^\infty\|_2)$, implying that GD actually enjoys a faster convergence rate. Despite such improvements, the convergence rate is still tied to the least eigenvalue of the Gram matrix, leading to slow convergence. We then develop the positive definiteness of Gram matrices with general smooth activation functions and provide the convergence analysis of natural gradient descent (NGD) in training two-layer PINNs, demonstrating that the learning rate can be $\mathcal{O}(1)$ and at this rate, the convergence rate is independent of the Gram matrix. In particular, for smooth activation functions, the convergence rate of NGD is quadratic.

## 1. Introduction

In recent years, neural networks have achieved remarkable breakthroughs in the fields of image recognition (He et al., 2016), natural language processing (Devlin et al., 2018), reinforcement learning (Silver et al., 2016), and so on. Moreover, due to the flexibility and scalability of neural networks, researchers are paying much attention in exploring new methods involving neural networks for handling problems in scientific computing. One long-standing and essential problem in this area is solving partial differential equations (PDEs) numerically. Classical numerical methods, such as finite difference, finite volume and finite elements methods, suffer from the curse of dimensionality when solving high-dimensional PDEs. Due to this drawback, various methods involving neural networks have been proposed for solving different type PDEs (Müller & Zeinhofer, 2023; Raissi et al., 2019; Yu et al., 2018; Zang et al., 2020; Siegel et al., 2023). Among them, the most representative approach is Physics-Informed Neural Networks (PINNs) (Raissi et al., 2019). In the framework of PINNs, one incorporate PDE constraints into the loss function and train the neural network with it. With the use of automatic differentiation, the neural network can be efficiently trained by first-order or second-order methods.

In the applications of neural networks, one inevitable issue is the selection of the optimization methods. First-order methods, such as gradient descent (GD) and stochastic gradient descent (SGD), are widely used in optimizing neural networks as they only calculate the gradient, making them computationally efficient. In addition to first-order methods, there has been significant interest in utilizing second-order optimization methods to accelerate training. These methods have proven to be applicable not only to regression problems, as demonstrated in Martens & Grosse (2015), but also to problems related to PDEs, as shown in Müller & Zeinhofer (2023); Raissi et al. (2019).

As for the convergence aspect of the optimization methods, it has been shown that gradient descent algorithm can even achieve zero training loss under the setting of over-parameterization, which refers to a situation where a model has more parameters than necessary to fit the data (Du et al., 2018; 2019; Allen-Zhu et al., 2019a;b; Arora et al., 2019; Li & Liang, 2018; Zou et al., 2020; Cao & Gu, 2019). These works are based on the idea of neural tangent kernel (NTK)(Jacot et al., 2018), which shows that training multi-layer fully-connected neural networks via gradient descent is equivalent to performing a certain kernel method as the width of every layer goes to infinity. As for the finite width neural networks, with more refined analysis, it can be shown that the parameters are closed to the initializations throughout the entire training process when

---

[1]Anonymous Institution, Anonymous City, Anonymous Region, Anonymous Country. Correspondence to: Anonymous Author <anon.email@domain.com>.

Preliminary work. Under review by the International Conference on Machine Learning (ICML). Do not distribute.

the width is large enough. This directly leads to the linear convergence for GD. Despite these attractive convergence results, the learning rate depends on the sample size and the Gram matrix, so it needs to be sufficiently small to guarantee convergence in practice. However, doing so results in a slow training process. In contrast to first-order methods, the second-order method NGD has been shown to enjoy fast convergence for the $L^2$ regression problems, as demonstrated in Zhang et al. (2019); Cai et al. (2019). However, the convergence of NGD in the context of training PINNs is still an open problem. In this paper, we demonstrate that when training PINNs, NGD indeed enjoys a faster convergence rate.

### 1.1. Contributions

The main contributions of our work can are summarized as follows:

- For the PINNs, we simultaneously improve both the learning rate $\eta$ of gradient descent and the requirement for the width $m$. The improvements rely on a new recursion formula for gradient descent, which is similar to that for regression problems. Specifically, we can improve the learning rate $\eta = \mathcal{O}(\lambda_0)$ required in Gao et al. (2023) to $\eta = \mathcal{O}(1/\|\boldsymbol{H}^\infty\|_2)$ and the requirement for the width $m$, i.e. $m = \widetilde{\Omega}\left(\frac{(n_1+n_2)^2}{\lambda_0^4 \delta^3}\right)$, can be improved to $m = \widetilde{\Omega}\left(\frac{1}{\lambda_0^4}(\log(\frac{n_1+n_2}{\delta}))\right)$, where $\widetilde{\Omega}$ indicates that some terms involving $\log(m)$ are omitted.

- We present a framework for demonstrating the positive definiteness of Gram matrices for a variety of commonly used smooth activation functions, including the logistic function, softplus function, hyperbolic tangent function, and others. This conclusion is not only applicable to the PDE we have considered but can also be naturally extended to other forms of PDEs.

- We provide the convergence results for natural gradient descent (NGD) in training over-parameterized two-layer PINNs with ReLU[3] activation functions and smooth activation functions. Due to the distinct optimization dynamics of NGD compared to GD, the learning rate can be $\mathcal{O}(1)$. Consequently, the convergence rate is independent of $n$ and $\lambda_0$, leading to faster convergence. Moreover, when the activation function is smooth, NGD can achieve a quadratic convergence rate.

### 1.2. Related Works

**First-order methods.** There are mainly two approaches to studying the optimization of neural networks and understanding why first-order methods can find a global minimum.

One approach is to analyze the optimization landscape, as demonstrated in Jin et al. (2017); Ge et al. (2015). It has been shown that gradient descent can find a global minimum in polynomial time if the optimization landscape possesses certain favorable geometric properties. However, some unrealistic assumptions in these works make it challenging to generalize the findings to practical neural networks. Another approach to understand the optimization of neural networks is by analyzing the optimization dynamics of first-order methods. For the two-layer ReLU neural networks, as shown in Du et al. (2018), randomly initialized gradient descent converges to a globally optimal solution at a linear rate, provided that the width $m$ is sufficiently large and no two inputs are parallel. Later, these results were extended to deep fully-connected feedforward neural networks and ResNet with smooth activation functions (Du et al., 2019). Results for both shallow and deep neural networks depend on the stability of the Gram matrices throughout the training process, which is crucial for convergence to the global minimum. In addition to regression and classification problems, Gao et al. (2023) demonstrated the convergence of the gradient descent for two-layer PINNs through a similar analysis of optimization dynamics. However, both Du et al. (2018) and Gao et al. (2023) require a sufficiently small learning rate and a large enough network width to achieve convergence. In this work, we conduct a refined analysis of gradient descent for PINNs, resulting in milder requirements for the learning rate and network width.

**Second-order methods.** Although second-order methods possess better convergence rate, they are rarely used in training deep neural networks due to the prohibitive computational cost. As a variant of the Gauss-Newton method, natural gradient descent (NGD) is more efficient in practice. Meanwhile, as shown in Zhang et al. (2019) and Cai et al. (2019), NGD also enjoys faster convergence rate for the $L^2$ regression problems compared to gradient descent. Müller & Zeinhofer (2023) proposed energy natural gradient descent for PINNs and deep Ritz method, demonstrating experimentally that this method yields solutions that are more accurate than those obtained through GD, Adam or BFGS. After observing the ill-conditioned loss landscape of PINNs, Rathore et al. (2024) introduced a novel second-order optimizer, Nys-NewtonCG (NNCG), showing that NNCG can significantly improve the solution returned by Adam+L-BFGS. Moreover, under the assumption that the PŁ$^\star$-condition holds, Rathore et al. (2024) demonstrated that the convergence rate of their algorithm is independent of the condition number, which is similar with our result. However, although the PŁ$^\star$-condition holds for over-parameterized neural networks in the context of regression problems (Liu et al., 2022), it remains unclear whether this condition holds for PINNs. In this paper, we provide the convergence analysis for NGD in training two-layer PINNs with ReLU[3] activation func-

tions or smooth activation functions, showing that it indeed converges at a faster rate.

### 1.3. Notations

We denote $[n] = \{1, 2, \cdots, n\}$ for $n \in \mathbb{N}$. Given a set $S$, we denote the uniform distribution on $S$ by $Unif\{S\}$. We use $I\{E\}$ to denote the indicator function of the event $E$. For two positive functions $f_1(n)$ and $f_2(n)$, we use $f_1(n) = \mathcal{O}(f_2(n))$, $f_2(n) = \Omega(f_1(n))$ or $f_1(n) \lesssim f_2(n)$ to represent $f_1(n) \leq C f_2(n)$, where $C$ is a universal constant $C$. A universal constant means a constant independent of any variables. Throughout the paper, we use boldface to denote vectors. Given $x_1, \cdots, x_d \in \mathbb{R}$, we use $(x_1, \cdots, x_d)$ or $[x_1, \cdots, x_d]$ to denote a row vector with $i$-th component $x_i$ for $i \in [d]$ and then $(x_1, \cdots, x_d)^T \in \mathbb{R}^d$ is a column vector.

### 1.4. Organization of this Paper

In Section 2, we provide the problem setup for training two-layer PINNs. We then present the improved convergence results of gradient descent for PINNs in Section 3. In Section 4, we analyze the convergence of natural gradient descent in training two-layer PINNs with ReLU$^3$ activation functions and smooth activation functions. We conclude in Section 5, and the detailed proofs are provided in the Appendix for readability and brevity.

## 2. Problem Setup

In this section, we consider the same setup as Gao et al. (2023), focusing on the PDE with the following form.

$$\begin{cases} \dfrac{\partial u}{\partial x_0}(\boldsymbol{x}) - \displaystyle\sum_{i=1}^{d} \dfrac{\partial^2 u}{\partial x_i^2}(\boldsymbol{x}) = f(\boldsymbol{x}), \ \boldsymbol{x} \in (0, T) \times \Omega, \\ u(\boldsymbol{x}) = g(\boldsymbol{x}), \ \boldsymbol{x} \in \{0\} \times \Omega \cup [0, T] \times \partial\Omega, \end{cases} \quad (1)$$

where $\Omega \subset \mathbb{R}^d$ is an open and bounded domain, $\boldsymbol{x} = (x_0, x_1, \cdots, x_d)^T \in \mathbb{R}^{d+1}$ and $x_0 \in [0, T]$ is the time variable. In the following, we assume that $\|\boldsymbol{x}\|_2 \leq 1$ for $\boldsymbol{x} \in [0, T] \times \bar{\Omega}$ and $f, g$ are bounded continuous functions.

Moreover, we consider a two-layer neural network of the following form.

$$\phi(\boldsymbol{x}; \boldsymbol{w}, \boldsymbol{a}) = \frac{1}{\sqrt{m}} \sum_{r=1}^{m} a_r \sigma(\boldsymbol{w}_r^T \tilde{\boldsymbol{x}}), \quad (2)$$

where $\boldsymbol{w} = (\boldsymbol{w}_1^T, \cdots, \boldsymbol{w}_m^T)^T \in \mathbb{R}^{m(d+2)}$, $\boldsymbol{a} = (a_1, \cdots, a_m)^T \in \mathbb{R}^m$ and for $r \in [m]$, $\boldsymbol{w}_r \in \mathbb{R}^{d+2}$ is the weight vector of the first layer, $a_r$ is the output weight and $\sigma(\cdot)$ is the ReLU$^3$ activation function. Here, $\tilde{\boldsymbol{x}} = (\boldsymbol{x}^T, 1)^T \in \mathbb{R}^{d+2}$ is the augmented vector from $\boldsymbol{x}$ and in the following, we write $\boldsymbol{x}$ for $\tilde{\boldsymbol{x}}$ for brevity.

In the framework of PINNs, given training samples $\{\boldsymbol{x}_p\}_{p=1}^{n_1}$ and $\{\boldsymbol{y}_j\}_{j=1}^{n_2}$ that are from interior and boundary respectively, we aim to minimize the following empirical loss function.

$$L(\boldsymbol{w}, \boldsymbol{a}) :=$$

$$\sum_{p=1}^{n_1} \frac{1}{2n_1} \left( \frac{\partial\phi}{\partial x_0}(\boldsymbol{x}_p; \boldsymbol{w}, \boldsymbol{a}) - \sum_{i=1}^{d} \frac{\partial^2\phi}{\partial x_i^2}(\boldsymbol{x}_p; \boldsymbol{w}, \boldsymbol{a}) - f(\boldsymbol{x}_p) \right)^2$$

$$+ \sum_{j=1}^{n_2} \frac{1}{2n_2} \left( \phi(\boldsymbol{y}_j; \boldsymbol{w}, \boldsymbol{a}) - g(\boldsymbol{y}_j) \right)^2. \quad (3)$$

Similar to that for the $L^2$ regression problems, we initialize the first layer vector $\boldsymbol{w}_r(0) \sim \mathcal{N}(\boldsymbol{0}, \boldsymbol{I})$, output weight $a_r \sim Unif(\{-1, 1\})$ for $r \in [m]$ and fix the output weights. Then the gradient descent updates the hidden weights by the following formulations:

$$\boldsymbol{w}_r(k+1) = \boldsymbol{w}_r(k) - \eta \frac{\partial L(\boldsymbol{w}(k), \boldsymbol{a})}{\partial \boldsymbol{w}_r} \quad (4)$$

for all $r \in [m]$ and $k \in \mathbb{N}$, where $\eta > 0$ is the learning rate. For brevity, we write $L(\boldsymbol{w})$ for $L(\boldsymbol{w}, \boldsymbol{a})$.

To simplify the notations, for the residuals of interior and boundary, we denote them by $s_p(\boldsymbol{w})$ and $h_j(\boldsymbol{w})$ respectively, i.e.,

$$s_p(\boldsymbol{w}) = \frac{1}{\sqrt{n_1}} \left( \frac{\partial\phi}{\partial x_0}(\boldsymbol{x}_p; \boldsymbol{w}) - \sum_{i=1}^{d} \frac{\partial^2\phi}{\partial x_i^2}(\boldsymbol{x}_p; \boldsymbol{w}) - f(\boldsymbol{x}_p) \right) \quad (5)$$

and

$$h_j(\boldsymbol{w}) = \frac{1}{\sqrt{n_2}} (\phi(\boldsymbol{y}_j; \boldsymbol{w}) - g(\boldsymbol{y}_j)). \quad (6)$$

Then the empirical loss function can be written as

$$L(\boldsymbol{w}) = \frac{1}{2} \left( \|\boldsymbol{s}(\boldsymbol{w})\|_2^2 + \|\boldsymbol{h}(\boldsymbol{w})\|_2^2 \right), \quad (7)$$

where

$$\boldsymbol{s}(\boldsymbol{w}) = (s_1(\boldsymbol{w}), \cdots, s_{n_1}(\boldsymbol{w}))^T \in \mathbb{R}^{n_1} \quad (8)$$

and

$$\boldsymbol{h}(\boldsymbol{w}) = (h_1(\boldsymbol{w}), \cdots, h_{n_2}(\boldsymbol{w}))^T \in \mathbb{R}^{n_2}. \quad (9)$$

At this time, we have

$$\frac{\partial L(\boldsymbol{w})}{\partial \boldsymbol{w}_r} = \sum_{p=1}^{n_1} s_p(\boldsymbol{w}) \frac{\partial s_p(\boldsymbol{w})}{\partial \boldsymbol{w}_r} + \sum_{j=1}^{n_2} h_j(\boldsymbol{w}) \frac{\partial h_j(\boldsymbol{w})}{\partial \boldsymbol{w}_r} \quad (10)$$

and the Gram matrix $\boldsymbol{H}(\boldsymbol{w})$ is defined as $\boldsymbol{H}(\boldsymbol{w}) = \boldsymbol{D}^T \boldsymbol{D}$, where

$$\boldsymbol{D} := \left( \frac{\partial s_1(\boldsymbol{w})}{\partial \boldsymbol{w}}, \cdots, \frac{\partial s_{n_1}(\boldsymbol{w})}{\partial \boldsymbol{w}}, \frac{\partial h_1(\boldsymbol{w})}{\partial \boldsymbol{w}}, \cdots, \frac{\partial h_{n_2}(\boldsymbol{w})}{\partial \boldsymbol{w}} \right). \quad (11)$$

## 3. Improved Results of GD for Two-Layer PINNs

To simplify the analysis, we make the following assumptions on the training data.

**Assumption 3.1.** For $p \in [n_1]$ and $j \in [n_2]$, $\|\boldsymbol{x}_p\|_2 \leq \sqrt{2}$, $\|\boldsymbol{y}_j\|_2 \leq \sqrt{2}$, where all inputs have been augmented.

**Assumption 3.2.** No two samples in $\{\boldsymbol{x}_p\}_{p=1}^{n_1} \cup \{\boldsymbol{y}_j\}_{j=1}^{n_2}$ are parallel.

Under Assumption 3.2, Lemma 3.3 in Gao et al. (2023) implies that the Gram matrix $\boldsymbol{H}^\infty := \mathbb{E}_{\boldsymbol{w} \sim \mathcal{N}(\boldsymbol{0},\boldsymbol{I})}[\boldsymbol{H}(\boldsymbol{w})]$ is strictly positive definite and we let $\lambda_0 = \lambda_{min}(\boldsymbol{H}^\infty)$. Similar to the case of the regression problem in Du et al. (2018), $\boldsymbol{H}^\infty$ plays an important role in the optimization process. Specifically, under over-parameterization and random initialization, we have two facts that (1) at initialization $\|\boldsymbol{H}(0) - \boldsymbol{H}^\infty\|_2 = \mathcal{O}(1/\sqrt{m})$ and (2) for any iteration $k \in \mathbb{N}$, $\|\boldsymbol{H}(k) - \boldsymbol{H}(0)\|_2 = \mathcal{O}(1/\sqrt{m})$. The following two lemmas can be used to verify these two facts, which are crucial in the convergence analysis.

**Lemma 3.3.** *If $m = \Omega\left(\frac{d^4}{\lambda_0^2} \log\left(\frac{n_1+n_2}{\delta}\right)\right)$, we have that with probability at least $1 - \delta$, $\|\boldsymbol{H}(0) - \boldsymbol{H}^\infty\|_2 \leq \frac{\lambda_0}{4}$ and $\lambda_{min}(\boldsymbol{H}(0)) \geq \frac{3}{4}\lambda_0$.*

*Remark* 3.4. Under the premise of deriving the same conclusion as Lemma 3.3, Lemma 3.5 in Gao et al. (2023) requires that $m = \tilde{\Omega}\left(\frac{(n_1+n_2)^4}{(n_1 n_2)^2 \lambda_0^2}\left(\log(\frac{1}{\delta})\right)^7\right)$, where some terms involving $\log(m)$ are omitted. In contrast, on one hand, our conclusion is independent of $n_1$ and $n_2$, and on the other hand, our conclusion exhibits a clear dependence on $d$. Moreover, the method in Gao et al. (2023) involves truncating the Gaussian distribution and then applying Hoeffding's inequality, which is quite complicated. In contrast, we utilize the concentration inequality for sub-Weibull random variables, which serves as a simple framework for this class of problems.

**Lemma 3.5.** *Let $R \in (0,1]$, if $\boldsymbol{w}_1(0), \cdots, \boldsymbol{w}_m(0)$ are i.i.d. generated from $\mathcal{N}(\boldsymbol{0},\boldsymbol{I})$, then with probability at least $1 - \delta - n_1 e^{-mR}$, the following holds. For any set of weight vectors $\boldsymbol{w}_1, \cdots, \boldsymbol{w}_m \in \mathbb{R}^{d+1}$ that satisfy for any $r \in [m]$, $\|\boldsymbol{w}_r - \boldsymbol{w}_r(0)\|_2 < R$, then*

$$\|\boldsymbol{H}(\boldsymbol{w}) - \boldsymbol{H}(0)\|_F < CM^2 R, \quad (12)$$

*where $M = 2(d+2)\log(2m(d+2)/\delta)$ and $C$ is a universal constant.*

*Remark* 3.6. Lemma 3.6 in Gao et al. (2023) shows that when $\|\boldsymbol{w}_r - \boldsymbol{w}_r(0)\|_2 \leq R = \tilde{\mathcal{O}}\left(\frac{\lambda_0 \delta}{(n_1+n_2)(\log m)^3}\right)$ holds for all $r \, in[m]$, then $\|\boldsymbol{H}(\boldsymbol{w}) - \boldsymbol{H}(0)\|_2 \leq \frac{\lambda_0}{4}$. In contrast, Lemma 3.5 only requires $R = \mathcal{O}\left(\frac{\lambda_0}{d^2(\log(m/\delta)^2}\right)$ to reach same result.

For the $L^2$ regression problem, as shown in Du et al. (2018), the convergence of gradient descent requires that the learning rate $\eta = \mathcal{O}(\lambda_0/n^2)$, where $n$ is the sample size of the regression problem. It is evident that this requirement on the learning rate is difficult to satisfy in practical scenarios, since $\lambda_0$ is unknown and $n^2$ is too large . For PINNs, Gao et al. (2023) follows the methodology of Du et al. (2018), thus inheriting similarly stringent requirements on the learning rate. Indeed, such stringent requirement stems from an inadequate decomposition method for the residual. Specifically, in Gao et al. (2023), the decomposition for the residual in the $(k+1)$-th iteration is same as the one in Du et al. (2018), i.e.,

$$\begin{pmatrix} \boldsymbol{s}(k+1) \\ \boldsymbol{h}(k+1) \end{pmatrix} = \begin{pmatrix} \boldsymbol{s}(k) \\ \boldsymbol{h}(k) \end{pmatrix} + \left[ \begin{pmatrix} \boldsymbol{s}(k+1) \\ \boldsymbol{h}(k+1) \end{pmatrix} - \begin{pmatrix} \boldsymbol{s}(k) \\ \boldsymbol{h}(k) \end{pmatrix} \right], \quad (13)$$

which leads to the requirements that $\eta = \mathcal{O}(\lambda_0)$ and $m = Poly(n_1, n_2, 1/\delta)$. Thus, it requires a new approach to achieve the improvements for $\eta$ and $m$. In fact, we can derive the following recursion formula.

**Lemma 3.7.** *For all $k \in \mathbb{N}$, we have*

$$\begin{pmatrix} \boldsymbol{s}(k+1) \\ \boldsymbol{h}(k+1) \end{pmatrix} = (\boldsymbol{I} - \eta \boldsymbol{H}(k)) \begin{pmatrix} \boldsymbol{s}(k) \\ \boldsymbol{h}(k) \end{pmatrix} + \boldsymbol{I}_1(k), \quad (14)$$

*where*

$$\boldsymbol{I}_1(k) = (I_1^1(k), \cdots, I_1^{n_1+n_2}(k))^T \in \mathbb{R}^{n_1+n_2}$$

*and for $p \in [n_1]$,*

$$I_1^p(k) = s_p(k+1) - s_p(k) - \left\langle \frac{\partial s_p(k)}{\partial \boldsymbol{w}}, \boldsymbol{w}(k+1) - \boldsymbol{w}(k) \right\rangle, \quad (15)$$

*for $j \in [n_2]$,*

$$I_1^{n_1+j}(k) = h_j(k+1) - h_j(k) - \left\langle \frac{\partial h_j(k)}{\partial \boldsymbol{w}}, \boldsymbol{w}(k+1) - \boldsymbol{w}(k) \right\rangle. \quad (16)$$

In the recursion formula (14), $\boldsymbol{I}_1(k)$ serves as a residual term. From the proof, we can see that $\|\boldsymbol{I}_1(k)\|_2 = \mathcal{O}(1/\sqrt{m})$ and thus, as $m$ becomes large enough, only the term $\boldsymbol{I} - \eta \boldsymbol{H}(k)$ is significant. This observation is the reason for the requirement of $\eta$. With these facts in mind, we arrive at our main result.

**Theorem 3.8.** *Under Assumption 3.1 and Assumption 3.2, if we set the number of hidden nodes*

$$m = \Omega\left(\frac{d^8}{\lambda_0^4} \log^6\left(\frac{md}{\delta}\right) \log\left(\frac{n_1+n_2}{\delta}\right)\right)$$

*and the learning rate $\eta = \mathcal{O}\left(\frac{1}{\|\boldsymbol{H}^\infty\|_2}\right)$, then with probability at least $1 - \delta$ over the random initialization, the gradient descent algorithm satisfies*

$$\left\| \begin{pmatrix} \boldsymbol{s}(k) \\ \boldsymbol{h}(k) \end{pmatrix} \right\|_2^2 \leq \left(1 - \frac{\eta \lambda_0}{2}\right)^k \left\| \begin{pmatrix} \boldsymbol{s}(0) \\ \boldsymbol{h}(0) \end{pmatrix} \right\|_2^2 \quad (17)$$

for all $k \in \mathbb{N}$.

*Remark* 3.9. It may be confusing that Gao et al. (2023) has used the same method in Du et al. (2018), yet it only requires $\eta = \mathcal{O}(\lambda_0)$. Actually, it is because that the loss function of PINN has been normalized. If we let $n_1 = n_2 = n$ and $\widetilde{\boldsymbol{H}}^\infty$ be the Gram matrix induced by unnormalized loss function of PINN, then $\lambda_{min}(\boldsymbol{H}^\infty) = \lambda_{min}(\widetilde{\boldsymbol{H}}^\infty)/n$, leading to the convergence rate similar to that of regression problem. At this point, due to the normalization of loss function, $\|\boldsymbol{H}^\infty\|_2$ can be bounded by the trace of $\boldsymbol{H}^\infty$, which is an explicit constant that is independent of the sample size $n_1, n_2$. Therefore, in practice, we can set the learning rate to satisfy the theoretical convergence requirement, bridging the gap between theory and practice.

*Remark* 3.10. Regarding the impact of dimensionality on the convergence of GD for PINNs, Theorem 3.8 consists of two parts: one is explicit, namely $d^8$, and the other is implicit, specifically $\lambda_0^4$, whose relationship with dimensionality remains unclear. The explicit impact arises from the form of the PDE. For instance, the PDE (1) we consider contains $\mathcal{O}(d)$ terms. In concurrent work, hoon Song et al. (2024) investigated the impact of dimensionality and the order of PDEs on convergence under the setting of continuous gradient flow. Specifically, for PDEs of order 2, the form they considered includes $\mathcal{O}(d^2)$ terms, and the explicit dependence on dimensionality is $d^{28}$. On the one hand, hoon Song et al. (2024) only addressed the continuous gradient flow case, while the discrete case requires more refined analysis as stated in Du et al. (2018). On the other hand, our results can naturally extend to the PDEs they considered, where the explicit dependence on dimensionality is $d^{16}$, which is better than the result in hoon Song et al. (2024). Investigating the lower bounds of the smallest eigenvalue of the NTK for PINNs, similar to what has been done for deep ReLU neural networks in Nguyen et al. (2021), represents a promising direction for future research.

Similar to Du et al. (2018) and Gao et al. (2023), we prove Theorem 3.8 by induction. Our induction hypothesis is the following convergence rate of the empirical loss and upper bounds for the weights.

*Condition* 1. At the $t$-th iteration, we have that for each $r \in [m]$, $\|\boldsymbol{w}_r(t)\|_2 \leq B$ and

$$L(t) \leq \left(1 - \frac{\eta\lambda_0}{2}\right)^t L(0), \qquad (18)$$

where $B = \sqrt{2(d+2)\log\left(\frac{2m(d+2)}{\delta}\right)} + 1$ and $L(k)$ is an abbreviation of $L(\boldsymbol{w}(k))$.

From the update formula of gradient descent, we can directly derive the following corollary, which indicates that under over-parameterization, the weights are closed to their initializations.

**Corollary 3.11.** *If Condition 1 holds for* $t = 0, \cdots, k$, *then we have for every* $r \in [m]$,

$$\|\boldsymbol{w}_r(k+1) - \boldsymbol{w}_r(0)\|_2 \leq \frac{CB^2\sqrt{L(0)}}{\sqrt{m}\lambda_0}, \qquad (19)$$

*where* $C$ *is a universal constant.*

**Proof Sketch:** Assume that Condition 1 holds for $t = 0, \cdots, k$, it suffices to demonstrate that Condition 1 also holds for $t = k + 1$.

From the recursion formula (14), we have that

$$
\begin{aligned}
&\left\|\begin{pmatrix}\boldsymbol{s}(k+1)\\\boldsymbol{h}(k+1)\end{pmatrix}\right\|_2^2 \\
&= \left\|(\boldsymbol{I} - \eta\boldsymbol{H}(k))\begin{pmatrix}\boldsymbol{s}(k)\\\boldsymbol{h}(k)\end{pmatrix} + \boldsymbol{I}_1(k)\right\|_2^2 \\
&\leq \|\boldsymbol{I} - \eta\boldsymbol{H}(k)\|_2^2 \left\|\begin{pmatrix}\boldsymbol{s}(k)\\\boldsymbol{h}(k)\end{pmatrix}\right\|_2^2 + \|\boldsymbol{I}_1(k)\|_2^2 \\
&\quad + 2\|\boldsymbol{I} - \eta\boldsymbol{H}(k)\|_2 \left\|\begin{pmatrix}\boldsymbol{s}(k)\\\boldsymbol{h}(k)\end{pmatrix}\right\|_2 \|\boldsymbol{I}_1(k)\|_2,
\end{aligned}
\qquad (20)
$$

where the inequality follows from the Cauchy's inequality.

Combining Corollary 3.11 with Lemma 3.5, we can deduce that when $m$ is large enough, we have $\|\boldsymbol{H}(k) - \boldsymbol{H}(0)\|_2 \leq \lambda_0/4$. Thus, $\lambda_{min}(\boldsymbol{H}(k)) \geq \lambda_0/2$ and $\boldsymbol{I} - \eta\boldsymbol{H}(k)$ is positive definite when $\eta = \mathcal{O}(1/\|\boldsymbol{H}^\infty\|_2)$. On the other hand, with Corollary 3.11, we can derive that $\|\boldsymbol{I}_1(k)\|_2 = \mathcal{O}(\eta\sqrt{L(k)}/\sqrt{m})$. Plugging these results into (20), we have

$$
\begin{aligned}
&\left\|\begin{pmatrix}\boldsymbol{s}(k+1)\\\boldsymbol{h}(k+1)\end{pmatrix}\right\|_2^2 \\
&= \left(\left(1 - \frac{\eta\lambda_0}{2}\right)^2 + \mathcal{O}\left(\frac{\eta^2}{m}\right) + \mathcal{O}\left(\frac{\eta}{\sqrt{m}}\right)\right)\left\|\begin{pmatrix}\boldsymbol{s}(k)\\\boldsymbol{h}(k)\end{pmatrix}\right\|_2^2 \\
&\leq \left(1 - \frac{\eta\lambda_0}{2}\right)\left\|\begin{pmatrix}\boldsymbol{s}(k)\\\boldsymbol{h}(k)\end{pmatrix}\right\|_2^2,
\end{aligned}
\qquad (21)
$$

where the last inequality holds when $m$ is large enough.

## 4. Convergence of NGD for Two-Layer PINNs

Although we have improved the learning rate of gradient descent for PINNs, it may still be necessary to set the learning rates to be sufficiently small for some complex PDEs. Because, although for all PDEs, $Trace(\boldsymbol{H}^\infty)$ is an explicit constant, it depends on the form of the PDE, and for complex PDEs, it may be quite large. Moreover, the convergence rate $1 - \frac{\eta\lambda_0}{2}$ also depends on $\lambda_0$, which depends on the sample size and may be extremely small. Zhang et al. (2019) and Cai et al. (2019) have provided the convergence results for natural gradient descent (NGD) in training over-parameterized two-layer neural networks for $L^2$ regression

problems. They showed that the maximal learning rate can be $\mathcal{O}(1)$ and the convergence rate is independent of $\lambda_0$, which result in a faster convergence rate. However, their methods cannot generalize directly to PINNs. In the section, we conduct the convergence analysis of NGD for PINNs and demonstrate that it results in a faster convergence rate for PINNs compared to gradient descent.

In this section, we consider the same setup as described in Section 2. Specifically, we focus on the PDE of the form given in (1) and follow the same initialization as described in Section 2. During the training process, we fix the output weight $\boldsymbol{a}$ and update the hidden weights via NGD. The optimization objective is the empirical loss function presented in (7), which is defined as follows:

$$L(\boldsymbol{w}) = \frac{1}{2}\left(\|\boldsymbol{s}(\boldsymbol{w})\|_2^2 + \|\boldsymbol{h}(\boldsymbol{w})\|_2^2\right), \qquad (22)$$

where $\boldsymbol{s}(\boldsymbol{w})$ and $\boldsymbol{h}(\boldsymbol{w})$ are defined in (8) and (9), respectively.

The NGD gives the following update rule:

$$\boldsymbol{w}(k+1) = \boldsymbol{w}(k) - \eta \boldsymbol{J}(k)^T \left(\boldsymbol{J}(k)\boldsymbol{J}(k)^T\right)^{-1}\begin{pmatrix} \boldsymbol{s}(k) \\ \boldsymbol{h}(k) \end{pmatrix}, \qquad (23)$$

where

$$\boldsymbol{J}(k) = \left(\boldsymbol{J}_1(k)^T, \cdots, \boldsymbol{J}_{n_1+n_2}(k)^T\right)^T \in \mathbb{R}^{(n_1+n_2)\times m(d+2)}$$

is the Jacobian matrix for the whole dataset and $\eta > 0$ is the learning rate. Specifically, for $p \in [n_1]$,

$$\boldsymbol{J}_p(k) = \left[\left(\frac{\partial s_p(k)}{\partial \boldsymbol{w}_1}\right)^T, \cdots, \left(\frac{\partial s_p(k)}{\partial \boldsymbol{w}_m}\right)^T\right] \in \mathbb{R}^{1\times m(d+2)} \qquad (24)$$

and for $j \in [n_2]$,

$$\boldsymbol{J}_{n_1+j}(k) = \left[\left(\frac{\partial h_j(k)}{\partial \boldsymbol{w}_1}\right)^T, \cdots, \left(\frac{\partial h_j(k)}{\partial \boldsymbol{w}_m}\right)^T\right] \in \mathbb{R}^{1\times m(d+2)} \qquad (25)$$

*Remark* 4.1. Zhang et al. (2019) and Cai et al. (2019) have independently and concurrently established the convergence of NGD in the context of regression problems. The difference lies in the fact that Zhang et al. (2019) focused on ReLU activation functions, whereas Cai et al. (2019) considered smooth activation functions and consistently set the learning rate to 1. Here, following Zhang et al. (2019), we refer to this approach as NGD. In Cai et al. (2019), the authors derived this method based on NTK kernel regression and termed it the Gram-Gauss-Newton (GGN) method.

For the activation function of the two-layer neural network

$$\phi(\boldsymbol{x}; \boldsymbol{w}, \boldsymbol{a}) = \frac{1}{\sqrt{m}}\sum_{r=1}^m a_r \sigma(\boldsymbol{w}_r^T \boldsymbol{x}), \qquad (26)$$

we consider settings where $\sigma(\cdot)$ is either the ReLU[3] activation function or a smooth activation function satisfying the following assumption.

**Assumption 4.2.** There exists a constant $c > 0$ such that $\sup_{z\in\mathbb{R}}|\sigma^{(3)}(z)| \leq c$ and for any $z, z' \in \mathbb{R}$,

$$|\sigma^{(k)}(z) - \sigma^{(k)}(z')| \leq c|z - z'|, \qquad (27)$$

where $k \in \{0, 1, 2, 3\}$. Moreover, $\sigma(\cdot)$ is analytic and is not a polynomial function.

**Lemma 4.3.** *If no two samples in $\{\boldsymbol{x}_p\}_{p=1}^{n_1} \cup \{\boldsymbol{y}_j\}_{j=1}^{n_2}$ are parallel, then the Gram matrix $\boldsymbol{H}^\infty$ is strictly positive definite for activation functions that satisfy Assumption 4.2, i.e., $\lambda_0 := \lambda_{min}(\boldsymbol{H}^\infty) > 0$.*

*Remark* 4.4. Assumption 4.2 holds for various commonly used activation function, including logistic function $\sigma(z) = 1/(1 + e^{-z})$, softplus function $\sigma(z) = \log(1 + e^z)$, hyperbolic tangent function $\sigma(z) = (e^z - e^{-z})/(e^z + e^{-z})$ and others. Compared to the ReLU[3] activation function, these smooth activation functions are more popular for PINNs because solving PDEs typically requires high-order derivatives.

Unlike the approach for gradient descent, Zhang et al. (2019) focus on the change of the Jacobian matrix for NGD rather than the Gram matrix. More precisely, they demonstrate that $\boldsymbol{J}(\boldsymbol{w})$ is stable with respect to $\boldsymbol{w}$, where $\boldsymbol{J}(\boldsymbol{w})$ is the Jacobian matrix with weight vector $\boldsymbol{w} = (\boldsymbol{w}_1^T, \cdots, \boldsymbol{w}_m^T)^T$. Roughly speaking, they show that when $\|\boldsymbol{w} - \boldsymbol{w}(0)\|_2$ is small, then $\|\boldsymbol{J}(\boldsymbol{w}) - \boldsymbol{J}(0)\|_2$ is also proportionately small. However, this approach is not applicable to PINNs, because the loss function involves derivatives. Roughly speaking, the stability considered in Zhang et al. (2019) is more global in nature, whereas ours is local. Since the subsequent conclusions require the boundedness of local weights, we do not use this stability. Moreover, from Theorem 1 in Zhang et al. (2019), we can see that this stability imposes additional constraints on the learning rate. Therefore, we instead focus on the stability of $\boldsymbol{J}(\boldsymbol{w})$ with respect to each individual weight vector $\boldsymbol{w}_r$, which provides a more targeted approach.

**Lemma 4.5.** *Let $R \in (0, 1]$, if $\boldsymbol{w}_1(0), \cdots, \boldsymbol{w}_m(0)$ are i.i.d. generated $\mathcal{N}(\boldsymbol{0}, \boldsymbol{I})$, then with probability at least $1 - P(\delta, m, R)$ the following holds. For any set of weight vectors $\boldsymbol{w}_1, \cdots, \boldsymbol{w}_m \in \mathbb{R}^{d+2}$ that satisfy for any $r \in [m]$, $\|\boldsymbol{w}_r - \boldsymbol{w}_r(0)\|_2 < R$, then*

*(1) when $\sigma(\cdot)$ is the ReLU[3] activation function, we have that*

$$\|\boldsymbol{J}(\boldsymbol{w}) - \boldsymbol{J}(0)\|_2 \leq CM\sqrt{R}, \qquad (28)$$

*where $C$ is a universal constant, $M = 2(d+2)\log(2m(d+2)/\delta)$ and*

$$P(\delta, m, R) = \delta + n_1 e^{-mR}; \qquad (29)$$

*(2) when $\sigma(\cdot)$ satisfies Assumption 4.2, we have that*

$$\|\boldsymbol{J}(\boldsymbol{w}) - \boldsymbol{J}(0)\|_2 \leq CdR \qquad (30)$$

for $m \geq \log^2(1/\delta)$, where $C$ is a universal constant and $P(\delta, m, R) = \delta$.

As shown in Lemma 4.5, for ReLU$^3$ activation function, $\|\boldsymbol{J}(\boldsymbol{w}) - \boldsymbol{J}(0)\|_2 = \mathcal{O}(\sqrt{R})$, whereas for smooth activation function, $\|\boldsymbol{J}(\boldsymbol{w}) - \boldsymbol{J}(0)\|_2 = \mathcal{O}(R)$. Since $R$ is is sufficiently small, $\mathcal{O}(R)$ is more favorable than $\mathcal{O}(\sqrt{R})$. In fact, the difference of (28) and (30) arises from the continuity of $\sigma'''(\cdot)$.

*Remark* 4.6. For the regression problems, it is shown in Zhang et al. (2019) that when $\sigma(\cdot)$ is the ReLU activation function, then with probability at least $1 - \delta$, for all weight vectors $\boldsymbol{w}$ that satisfy $\|\boldsymbol{w} - \boldsymbol{w}(0)\|_2 \leq R'$, the following holds.

$$\|\boldsymbol{J}(\boldsymbol{w}) - \boldsymbol{J}(0)\|_2 \lesssim \frac{(R')^{1/3}}{\delta^{1/3}m^{1/6}}.$$

Setting $R = R'/\sqrt{m}$ in Lemma 4.5, then $\|\boldsymbol{w} - \boldsymbol{w}(0)\|_2 \leq R'$ and (28) becomes

$$\|\boldsymbol{J}(\boldsymbol{w}) - \boldsymbol{J}(0)\|_2 \lesssim \frac{\log(\frac{1}{\delta})(R')^{1/2}}{m^{1/4}}.$$

Since $R' = \mathcal{O}(\|\boldsymbol{y} - \boldsymbol{u}(0)\|_2/\sqrt{\lambda_0})$ for regression problems, our method results in a less favorable dependence on $R'$ and more favorable dependence on $m$ and $\delta$. This can improve $m = Poly(1/\delta)$ to $m = Poly(\log(1/\delta))$ for the regression problems.

More importantly, the stability considered in Zhang et al. (2019) results in that the learning rate must satisfy that $\eta \leq \frac{1-C}{(1+C)^2}$, where $0 \leq C < 1/2$ is a constant appearing in the stability of Jacobian matrix. This requirement for the learning rate may be difficult to satisfy, as $C$ is unknown.

With the stability of Jacobian matrix, we can derive the following convergence results.

**Theorem 4.7.** *Let $L(k) = L(\boldsymbol{w}(k))$, then the following conclusions hold.*

*(1) When $\sigma(\cdot)$ is the ReLU$^3$ activation function, under Assumption 3.2, we set*

$$m = \Omega\left(\frac{1}{(1-\eta)^2}\frac{d^8}{\lambda_0^4}\log^6\left(\frac{md}{\delta}\right)\log\left(\frac{n_1+n_2}{\delta}\right)\right)$$

*and $\eta \in (0, 1)$, then with probability at least $1 - \delta$ over the random initialization for all $k \in \mathbb{N}$*

$$L(k) \leq (1-\eta)^k L(0). \tag{31}$$

*(2) When $\sigma(\cdot)$ satisfies Assumption 4.2, under Assumption 3.2, we set*

$$m = \Omega\left(\frac{1}{1-\eta}\frac{d^6}{\lambda_0^3}\log^2\left(\frac{md}{\delta}\right)\log\left(\frac{n_1+n_2}{\delta}\right)\right)$$

*and $\eta \in (0, 1)$, then with probability at least $1 - \delta$ over the random initialization for all $k \in \mathbb{N}$*

$$L(k) \leq (1-\eta)^k L(0). \tag{32}$$

In Theorem 4.7, the requirements of $m$ with ReLU$^3$ and smooth activation functions exhibit different dependencies on $\lambda_0$ and $d$. The discrepancy is primarily due to the distinct formulations presented in (28) and (30) of Lemma 4.5.

*Remark* 4.8. We first compare our results with those of NGD for $L^2$ regression problems. Given that the convergence results are the same, our focus shifts to examining the necessary conditions for the width $m$. As demonstrated in Zhang et al. (2019) and Cai et al. (2019), it is required that $m = \Omega\left(\frac{n^4}{\lambda_0^4\delta^3}\right)$ for ReLU activation function and $m = \Omega\left(\max\left\{\frac{n^4}{\lambda_0^4}, \frac{n^2 d\log(n/\delta)}{\lambda_0^2}\right\}\right)$ for smooth activation function. Clearly, our result has a worse dependence on $d$, which is inevitable due to the involvement of derivatives in the loss function. Moreover, our requirement for $m$ appears to be almost independent of $n$, primarily because our loss function has been normalized. With smooth activation functions, in addition to the dependence on $d$, Theorem 4.7 (2) only requires that $m = \Omega(\lambda_0^{-3})$. However, Cai et al. (2019) demands a more stringent condition, requiring that $m = \Omega(\lambda_0^{-4})$.

Continuing our analysis, we contrast our results with those of GD for PINNs. Roughly speaking, Gao et al. (2023) has shown that when $\sigma(\cdot)$ is the ReLU$^3$ activation function, $m = \widetilde{\Omega}\left(\frac{(n_1+n_2)^2}{\lambda_0^4\delta^3}\right)$ and $\eta = \mathcal{O}(\lambda_0)$, then the convergence result (17) holds. It is evident that our result, i.e, Theorem 4.7 (1), has a milder dependence on $n_1, n_2$ and $\delta$. Furthermore, the learning rate and convergence rate are independent of $\lambda_0$, resulting in faster convergence.

Comparing with our results in Section 3, the requirement for $m$ in Theorem 4.7 (1) is the same as in Theorem 3.8, when we make $\eta$ less close to 1. On the other hand, since $\eta = \mathcal{O}(1)$ and the convergence rate only depends on $\eta$, NGD can lead to faster convergence than GD.

Note that as $\eta$ approaches 1, the width $m$ tends to infinity, thus, the convergence results in Theorem 4.7 become vacuous. In fact, when $\eta = 1$, NGD can enjoy a second-order convergence rate even though $m$ is finite, provided that $\sigma(\cdot)$ satisfies Assumption 4.2.

**Corollary 4.9.** *Under Assumption 3.2 and Assumption 4.2, set $\eta = 1$ and*

$$m = \Omega\left(\frac{d^6}{\lambda_0^3}\log^2\left(\frac{md}{\delta}\right)\log\left(\frac{n_1+n_2}{\delta}\right)\right),$$

*then with probability at least $1 - \delta$, we have*

$$\left\|\begin{pmatrix}\boldsymbol{s}(t+1)\\\boldsymbol{h}(t+1)\end{pmatrix}\right\|_2 \leq \frac{CB^4}{\sqrt{m\lambda_0^3}}\left\|\begin{pmatrix}\boldsymbol{s}(t)\\\boldsymbol{h}(t)\end{pmatrix}\right\|_2^2$$

*for all $t \in \mathbb{N}$, where $C$ is a universal constant and $B = \sqrt{2(d+2)\log(2m(d+2)/\delta)} + 1$.*

*Remark* 4.10. Cai et al. (2019) has demonstrated the second-order convergence for regression problems with smooth activation functions. Specifically, it is shown in Cai et al. (2019) that

$$\|\boldsymbol{y} - \boldsymbol{u}(t+1)\|_2 \lesssim \frac{n^{3/2}}{\sqrt{m}\lambda_0^2}\|\boldsymbol{y} - \boldsymbol{u}(t)\|_2^2.$$

Actually, when applying our method used in Corollary 4.9, we can get a more satisfactory result as follows.

$$\|\boldsymbol{y} - \boldsymbol{u}(t+1)\|_2 \lesssim \frac{n^{3/2}}{\sqrt{m}\lambda_0^3}\|\boldsymbol{y} - \boldsymbol{u}(t)\|_2^2.$$

Instead of inducing on the convergence rate of the empirical loss function, as shown in Condition 1, we perform induction on the movements of the hidden weights as follows.

*Condition* 2. At the $t$-th iteration, we have $\|\boldsymbol{w}_r(t)\|_2 \leq B$ and

$$\|\boldsymbol{w}_r(t) - \boldsymbol{w}_r(0)\|_2 \leq \frac{CB^2\sqrt{L(0)}}{\sqrt{m}\lambda_0} := R'$$

for all $r \in [m]$, where $C$ is a universal constant and $B = \sqrt{2(d+2)\log\left(\frac{2m(d+2)}{\delta}\right)} + 1$.

With Condition 2, we can directly derive the following convergence rate of the empirical loss function.

**Corollary 4.11.** *If Condition 2 holds for $t = 0, \cdots, k$ and $R' \leq R$ and $R'' \lesssim \sqrt{1-\eta}\sqrt{\lambda_0}$, then*

$$L(t) \leq (1-\eta)^t L(0),$$

*holds for $t = 0, \cdots, k$, where $R$ is the constant in Lemma 4.5 and $R'' = CM\sqrt{R}$ is in (28) when $\sigma$ is the ReLU$^3$ activation function, $R'' = CdR$ is in (30) when $\sigma$ satisfies Assumption 4.2.*

**Proof Sketch:** First, let $\boldsymbol{u}(t) = \begin{pmatrix} \boldsymbol{s}(t) \\ \boldsymbol{h}(t) \end{pmatrix}$, then from the updating formula of NGD (23), we have

$$\boldsymbol{u}(t+1) - \boldsymbol{u}(t)$$
$$= \boldsymbol{u}\left(\boldsymbol{w}(t) - \eta\boldsymbol{J}(t)^T\boldsymbol{H}(t)^{-1}\boldsymbol{u}(\boldsymbol{w}(t))\right) - \boldsymbol{u}(\boldsymbol{w}(t))$$
$$= -\int_0^1 \left\langle \frac{\partial\boldsymbol{u}(\boldsymbol{w}(s))}{\partial\boldsymbol{w}}, \eta\boldsymbol{J}(t)^T\boldsymbol{H}(t)^{-1}\boldsymbol{u}(\boldsymbol{w}(t)) \right\rangle ds$$
$$= -\int_0^1 \left\langle \frac{\partial\boldsymbol{u}(\boldsymbol{w}(t))}{\partial\boldsymbol{w}}, \eta\boldsymbol{J}(t)^T\boldsymbol{H}(t)^{-1}\boldsymbol{u}(\boldsymbol{w}(t)) \right\rangle ds$$
$$+ \int_0^1 \left\langle \frac{\partial\boldsymbol{u}(\boldsymbol{w}(t))}{\partial\boldsymbol{w}} - \frac{\partial\boldsymbol{u}(\boldsymbol{w}(s))}{\partial\boldsymbol{w}}, \eta\boldsymbol{J}(t)^T\boldsymbol{H}(t)^{-1}\boldsymbol{u}(t) \right\rangle ds$$
$$:= \boldsymbol{I}_1(t) + \boldsymbol{I}_2(t), \tag{33}$$

where the second equality is from the fundamental theorem of calculus and $\boldsymbol{w}(s) = s\boldsymbol{w}(t+1) + (1-s)\boldsymbol{w}(t) = \boldsymbol{w}(t) - s\eta\boldsymbol{J}(t)^T\boldsymbol{H}(t)^{-1}\boldsymbol{u}(t)$.

In the proof, we assume that Condition 2 holds for $t = 0, \cdots, k$. Then from Corollary 4.11, to prove Theorem 4.7, it suffices to demonstrate that this condition also holds for $t = k+1$. Here, we primarily explain the process from Condition 2 to Corollary 4.11, while other content is placed in the appendix.

Note that $\frac{\partial\boldsymbol{u}(\boldsymbol{w}(t))}{\partial\boldsymbol{w}} = \boldsymbol{J}(t)$, thus $\boldsymbol{I}_1(t) = \eta\boldsymbol{u}(t)$. Plugging this into (33) yields that

$$\boldsymbol{u}(t+1) = (1-\eta)\boldsymbol{u}(t) + \boldsymbol{I}_2(t). \tag{34}$$

From equation (34), we can see the difference between NGD and GD. Recall that the iteration formula for GD is

$$\boldsymbol{u}(t+1) = (1-\eta\boldsymbol{H}(t))\boldsymbol{u}(t) + \boldsymbol{I}_1(t).$$

Precisely because of this, the convergence rate of GD is inevitably influenced by $\lambda_0$, whereas that of NGD is not.

From the stability of the Jacobian matrix, we can deduce that $\|\boldsymbol{I}_2(t)\|_2 = \mathcal{O}(\eta\|\boldsymbol{u}(t)\|_2/\sqrt{m})$. Plugging this into (34) yields that

$$\|\boldsymbol{u}(t+1)\|_2^2$$
$$\leq \|(1-\eta)\boldsymbol{u}(t)\|_2^2 + \|\boldsymbol{I}_2(t)\|_2^2 + 2(1-\eta)\|\boldsymbol{u}(t)\|_2\|\boldsymbol{I}_2(t)\|_2$$
$$= \left((1-\eta)^2 + \mathcal{O}\left(\frac{\eta^2}{m}\right) + 2(1-\eta)\mathcal{O}\left(\frac{\eta}{\sqrt{m}}\right)\right)\|\boldsymbol{u}(t)\|_2^2$$
$$\leq (1-\eta)\|\boldsymbol{u}(t)\|_2^2, \tag{35}$$

where the last inequality holds if $m$ is large enough.

## 5. Conclusion and Discussion

In this paper, we have improved the conditions required for the convergence of gradient descent for PINNs, showing that gradient descent actually achieves a better convergence rate. Furthermore, we demonstrate that natural gradient descent can find the global optima of two-layer PINNs with ReLU$^3$ or smooth activation functions for a class of second-order linear PDEs. Compared to gradient descent, natural gradient descent exhibits a faster convergence rate and its maximal learning rate is $\mathcal{O}(1)$. However, natural gradient descent is quite expensive in terms of computation and memory in training neural networks. As a result, several cost-effective variants have been proposed, such as K-FAC (Martens & Grosse, 2015) and mini-batch natural gradient descent. It would be interesting to investigate the convergence of these methods for PINNs. Additionally, extending the convergence analysis to deep neural networks and studying the generalization bounds of trained PINNs are important directions for future research.

## Impact Statement

This paper presents work whose goal is to advance the field of Machine Learning. There are many potential societal consequences of our work, none which we feel must be specifically highlighted here.

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

## Appendix

Before the proofs, we first define the event

$$A_{ir} := \{\exists \boldsymbol{w} : \|\boldsymbol{w} - \boldsymbol{w}_r(0)\|_2 \le R, I\{\boldsymbol{w}^T \boldsymbol{x}_i \ge 0\} \ne I\{\boldsymbol{w}_r(0)^T \boldsymbol{x}_i \ge 0\}\} \tag{36}$$

for all $i \in [n]$.

Note that the event happens if and only if $|\boldsymbol{w}_r(0)^T \boldsymbol{x}_i| < \|\boldsymbol{x}_i\|_2 R$, thus by the anti-concentration inequality of Gaussian distribution, we have

$$P(A_{ir}) = P_{z \sim \mathcal{N}(0, \|\boldsymbol{x}_i\|_2^2)}(|z| < R) = P_{z \sim \mathcal{N}(0,1)}(|z| < R) \le \frac{2R}{\sqrt{2\pi}}. \tag{37}$$

Let $S_i = \{r \in [m] : I\{A_{ir}\} = 0\}$ and $S_i^\perp = [m] \backslash S_i$.

Then, we need to recall that

$$\frac{\partial s_p(\boldsymbol{w})}{\partial \boldsymbol{w}_r} = \frac{a_r}{\sqrt{mn_1}} \left[ \sigma''(\boldsymbol{w}_r^T \boldsymbol{x}_p) w_{r0} \boldsymbol{x}_p + \sigma'(\boldsymbol{w}_r^T \boldsymbol{x}_p) \begin{pmatrix} 1 \\ \boldsymbol{0}_{d+1} \end{pmatrix} - \sigma'''(\boldsymbol{w}_r^T \boldsymbol{x}_p) \|\boldsymbol{w}_{r1}\|_2^2 \boldsymbol{x}_p - 2\sigma''(\boldsymbol{w}_r^T \boldsymbol{x}_p) \begin{pmatrix} 0 \\ \boldsymbol{w}_{r1} \end{pmatrix} \right] \tag{38}$$

and

$$\frac{\partial h_j(\boldsymbol{w})}{\partial \boldsymbol{w}_r} = \frac{a_r}{\sqrt{mn_2}} \sigma'(\boldsymbol{w}_r^T \boldsymbol{y}_j) \boldsymbol{y}_j. \tag{39}$$

## A. Proof of Section 3

### A.1. Proof of Lemma 3.3

*Proof.* In the following, we aim to bound $\|\boldsymbol{H}(0) - \boldsymbol{H}^\infty\|_F$, as $\|\boldsymbol{H}(0) - \boldsymbol{H}^\infty\|_2 \le \|\boldsymbol{H}(0) - \boldsymbol{H}^\infty\|_F$. Note that the entries of $\boldsymbol{H}(0) - \boldsymbol{H}^\infty$ have three forms as follows.

$$\sum_{r=1}^m \left\langle \frac{\partial s_i(\boldsymbol{w}(0))}{\partial \boldsymbol{w}_r}, \frac{\partial s_j(\boldsymbol{w}(0))}{\partial \boldsymbol{w}_r} \right\rangle - \mathbb{E}_{\boldsymbol{w}(0)} \left[ \sum_{r=1}^m \left\langle \frac{\partial s_i(\boldsymbol{w}(0))}{\partial \boldsymbol{w}_r}, \frac{\partial s_j(\boldsymbol{w}(0))}{\partial \boldsymbol{w}_r} \right\rangle \right], \tag{40}$$

$$\sum_{r=1}^m \left\langle \frac{\partial s_i(\boldsymbol{w}(0))}{\partial \boldsymbol{w}_r}, \frac{\partial h_j(\boldsymbol{w}(0))}{\partial \boldsymbol{w}_r} \right\rangle - \mathbb{E}_{\boldsymbol{w}(0)} \left[ \sum_{r=1}^m \left\langle \frac{\partial s_i(\boldsymbol{w}(0))}{\partial \boldsymbol{w}_r}, \frac{\partial h_j(\boldsymbol{w}(0))}{\partial \boldsymbol{w}_r} \right\rangle \right] \tag{41}$$

and

$$\sum_{r=1}^m \left\langle \frac{\partial h_i(\boldsymbol{w}(0))}{\partial \boldsymbol{w}_r}, \frac{\partial h_j(\boldsymbol{w}(0))}{\partial \boldsymbol{w}_r} \right\rangle - \mathbb{E}_{\boldsymbol{w}} \left[ \sum_{r=1}^m \left\langle \frac{\partial h_i(\boldsymbol{w}(0))}{\partial \boldsymbol{w}_r}, \frac{\partial h_j(\boldsymbol{w}(0))}{\partial \boldsymbol{w}_r} \right\rangle \right]. \tag{42}$$

For the first form (40), to simplify the analysis, we let

$$\boldsymbol{Z}_r(i) = \sigma''(\boldsymbol{w}_r(0)^T \boldsymbol{x}_i) w_{r0}(0) \boldsymbol{x}_i + \sigma'(\boldsymbol{w}_r(0)^T \boldsymbol{x}_i) \begin{pmatrix} 1 \\ \boldsymbol{0}_{d+1} \end{pmatrix}$$

$$- \sigma'''(\boldsymbol{w}_r(0)^T \boldsymbol{x}_p) \|\boldsymbol{w}_{r1}(0)\|_2^2 \boldsymbol{x}_p - 2\sigma''(\boldsymbol{w}_r(0)^T \boldsymbol{x}_i) \begin{pmatrix} 0 \\ \boldsymbol{w}_{r1}(0) \end{pmatrix}$$

and

$$X_r(ij) = \langle \boldsymbol{Z}_r(i), \boldsymbol{Z}_r(j) \rangle,$$

then

$$\sum_{r=1}^m \left\langle \frac{\partial s_p(\boldsymbol{w}(0))}{\partial \boldsymbol{w}_r}, \frac{\partial s_j(\boldsymbol{w}(0))}{\partial \boldsymbol{w}_r} \right\rangle - \mathbb{E}_{\boldsymbol{w}} \left[ \sum_{r=1}^m \left\langle \frac{\partial s_p(\boldsymbol{w}(0))}{\partial \boldsymbol{w}_r}, \frac{\partial s_j(\boldsymbol{w}(0))}{\partial \boldsymbol{w}_r} \right\rangle \right] = \frac{1}{n_1 m} \sum_{r=1}^m [X_r(ij) - \mathbb{E} X_r(ij)].$$

Note that $|X_r(ij)| \lesssim 1 + \|\boldsymbol{w}_r(0)\|_2^4$, thus

$$\|X_r(ij)\|_{\psi_{\frac{1}{2}}} \lesssim 1 + \left\| \|\boldsymbol{w}_r(0)\|_2^4 \right\|_{\psi_{\frac{1}{2}}} \lesssim 1 + \left\| \|\boldsymbol{w}_r(0)\|_2^2 \right\|_{\psi_1}^2 \lesssim d^2.$$

Here, for more details on the Orlicz norm, see the remarks after Lemma C.1.

For the centered random variable, the property of $\psi_{\frac{1}{2}}$ quasi-norm implies that

$$\|X_r(ij) - \mathbb{E}[X_r(ij)]\|_{\psi_{\frac{1}{2}}} \lesssim \|X_r(ij)\|_{\psi_{\frac{1}{2}}} + \|\mathbb{E}[X_r(ij)]\|_{\psi_{\frac{1}{2}}} \lesssim d^2.$$

Therefore, applying Lemma C.1 yields that with probability at least $1 - \delta$,

$$\left|\sum_{r=1}^{m} \frac{1}{m} \left[X_r(ij) - \mathbb{E}X_r(ij)\right]\right| \lesssim \frac{d^2}{\sqrt{m}} \sqrt{\log\left(\frac{1}{\delta}\right)} + \frac{d^2}{m} \left(\log\left(\frac{1}{\delta}\right)\right)^2,$$

which directly yields that

$$\left|\sum_{r=1}^{m} \left\langle \frac{\partial s_p(\boldsymbol{w}(0))}{\partial \boldsymbol{w}_r}, \frac{\partial s_j(\boldsymbol{w}(0))}{\partial \boldsymbol{w}_r} \right\rangle - \mathbb{E}_{\boldsymbol{w}(0)} \left[\sum_{r=1}^{m} \left\langle \frac{\partial s_p(\boldsymbol{w}(0))}{\partial \boldsymbol{w}_r}, \frac{\partial s_j(\boldsymbol{w}(0))}{\partial \boldsymbol{w}_r} \right\rangle\right]\right| \lesssim \frac{d^2}{n_1\sqrt{m}} \sqrt{\log\left(\frac{1}{\delta}\right)} + \frac{d^2}{n_1 m} \left(\log\left(\frac{1}{\delta}\right)\right)^2.$$
$$\tag{43}$$

Similarly, for the second form (41) and third form (42), we can deduce that

$$\left\|\left\langle \frac{\partial s_i(\boldsymbol{w}(0))}{\partial \boldsymbol{w}_r}, \frac{\partial h_j(\boldsymbol{w}(0))}{\partial \boldsymbol{w}_r} \right\rangle - \mathbb{E}_{\boldsymbol{w}(0)} \left[\left\langle \frac{\partial s_i(\boldsymbol{w}(0))}{\partial \boldsymbol{w}_r}, \frac{\partial h_j(\boldsymbol{w}(0))}{\partial \boldsymbol{w}_r} \right\rangle\right]\right\|_{\psi_{\frac{1}{2}}} \lesssim \frac{d^2}{\sqrt{n_1 n_2}m}$$

and

$$\left\|\left\langle \frac{\partial h_i(\boldsymbol{w}(0))}{\partial \boldsymbol{w}_r}, \frac{\partial h_j(\boldsymbol{w}(0))}{\partial \boldsymbol{w}_r} \right\rangle - \mathbb{E}_{\boldsymbol{w}(0)} \left[\left\langle \frac{\partial h_i(\boldsymbol{w}(0))}{\partial \boldsymbol{w}_r}, \frac{\partial h_j(\boldsymbol{w}(0))}{\partial \boldsymbol{w}_r} \right\rangle\right]\right\|_{\psi_{\frac{1}{2}}} \lesssim \frac{d^2}{n_2 m}.$$

Thus applying Lemma C.1 yields that with probability at least $1 - \delta$,

$$\left|\sum_{r=1}^{m} \left\langle \frac{\partial s_i(\boldsymbol{w}(0))}{\partial \boldsymbol{w}_r}, \frac{\partial h_j(\boldsymbol{w}(0))}{\partial \boldsymbol{w}_r} \right\rangle - \mathbb{E}_{\boldsymbol{w}(0)} \left[\sum_{r=1}^{m} \left\langle \frac{\partial s_i(\boldsymbol{w}(0))}{\partial \boldsymbol{w}_r}, \frac{\partial h_j(\boldsymbol{w}(0))}{\partial \boldsymbol{w}_r} \right\rangle\right]\right| \lesssim \frac{d^2}{\sqrt{n_1 n_2}\sqrt{m}} \sqrt{\log\left(\frac{1}{\delta}\right)} + \frac{d^2}{\sqrt{n_1 n_2}m} \log\left(\frac{1}{\delta}\right)$$
$$\tag{44}$$

and with probability at least $1 - \delta$,

$$\left|\sum_{r=1}^{m} \left\langle \frac{\partial h_i(\boldsymbol{w}(0))}{\partial \boldsymbol{w}_r}, \frac{\partial h_j(\boldsymbol{w}(0))}{\partial \boldsymbol{w}_r} \right\rangle - \mathbb{E}_{\boldsymbol{w}(0)} \left[\sum_{r=1}^{m} \left\langle \frac{\partial h_i(\boldsymbol{w}(0))}{\partial \boldsymbol{w}_r}, \frac{\partial h_j(\boldsymbol{w}(0))}{\partial \boldsymbol{w}_r} \right\rangle\right]\right| \lesssim \frac{d^2}{n_2\sqrt{m}} \sqrt{\log\left(\frac{1}{\delta}\right)} + \frac{d^2}{n_2 m} \log\left(\frac{1}{\delta}\right).$$
$$\tag{45}$$

Combining (43), (44) and (45), we can deduce that with probability at least $1 - \delta$,

$$\begin{aligned}
\|\boldsymbol{H}(0) - \boldsymbol{H}^\infty\|_2^2 \\
\leq \|\boldsymbol{H}(0) - \boldsymbol{H}^\infty\|_F^2 \\
\lesssim \frac{d^4}{m} \log\left(\frac{n_1 + n_2}{\delta}\right) + \frac{d^4}{m^2} \left(\log\left(\frac{n_1 + n_2}{\delta}\right)\right)^4 \\
\lesssim \frac{d^4}{m} \log\left(\frac{n_1 + n_2}{\delta}\right).
\end{aligned}$$

Thus when $\sqrt{\frac{d^4}{m} \log\left(\frac{n_1 + n_2}{\delta}\right)} \lesssim \frac{\lambda_0}{4}$, i.e.,

$$m = \Omega\left(\frac{d^4}{\lambda_0^2} \log\left(\frac{n_1 + n_2}{\delta}\right)\right),$$

we have $\lambda_{min}(\boldsymbol{H}(0)) \geq \frac{3}{4}\lambda_0$.

$\square$

## A.2. Proof of Lemma 3.5

*Proof.* We first reformulate the term $\frac{\partial s_p(k)}{\partial \boldsymbol{w}_r}$ in (38) as follows.

$$\frac{\partial s_p(\boldsymbol{w})}{\partial \boldsymbol{w}_r} = \frac{a_r}{\sqrt{mn_1}} \left[ \sigma''(\boldsymbol{w}_r^T \boldsymbol{x}_p) \begin{pmatrix} w_{r0} x_{p0} \\ w_{r0} x_{p1} - 2\boldsymbol{w}_{r1} \end{pmatrix} + \sigma'(\boldsymbol{w}_r^T \boldsymbol{x}_p) \begin{pmatrix} 1 \\ \boldsymbol{0}_{d+1} \end{pmatrix} - \sigma'''(\boldsymbol{w}_r^T \boldsymbol{x}_p) \|\boldsymbol{w}_{r1}\|_2^2 \boldsymbol{x}_p \right].$$

It suffices to bound $\|\boldsymbol{H}(\boldsymbol{w}) - \boldsymbol{H}(0)\|_F$, which can in turn allows us to bound each entry of $\boldsymbol{H}(\boldsymbol{w}) - \boldsymbol{H}(0)$.

For $i \in [n_1]$ and $j \in [n_1]$, we have that

$$\begin{aligned} H_{ij}(\boldsymbol{w}) &= \sum_{r=1}^{m} \left\langle \frac{\partial s_i(\boldsymbol{w})}{\partial \boldsymbol{w}_r}, \frac{\partial s_j(\boldsymbol{w})}{\partial \boldsymbol{w}_r} \right\rangle \\ &= \frac{1}{n_1 m} \sum_{r=1}^{m} \left\langle \sigma''(\boldsymbol{w}_r^T \boldsymbol{x}_i) \begin{pmatrix} w_{r0} x_{i0} \\ w_{r0} \boldsymbol{x}_{i1} - 2\boldsymbol{w}_{r1} \end{pmatrix} + \sigma'(\boldsymbol{w}_r^T \boldsymbol{x}_i) \begin{pmatrix} 1 \\ \boldsymbol{0}_{d+1} \end{pmatrix} - \sigma'''(\boldsymbol{w}_r^T \boldsymbol{x}_i) \|\boldsymbol{w}_{r1}\|_2^2 \boldsymbol{x}_i, \right. \\ &\qquad \left. \sigma''(\boldsymbol{w}_r^T \boldsymbol{x}_j) \begin{pmatrix} w_{r0} x_{j0} \\ w_{r0} \boldsymbol{x}_{j1} - 2\boldsymbol{w}_{r1} \end{pmatrix} + \sigma'(\boldsymbol{w}_r^T \boldsymbol{x}_j) \begin{pmatrix} 1 \\ \boldsymbol{0}_{d+1} \end{pmatrix} - \sigma'''(\boldsymbol{w}_r^T \boldsymbol{x}_j) \|\boldsymbol{w}_{r1}\|_2^2 \boldsymbol{x}_j \right\rangle \end{aligned}$$

After expanding the inner product term, we can find that although it has nine terms, it only consists of six classes. For simplicity, we use the following six symbols to represent the corresponding classes.

$$\sigma'' \sigma'', \sigma'' \sigma', \sigma' \sigma', \sigma''' \sigma'', \sigma''' \sigma', \sigma''' \sigma'''.$$

For instance, $\sigma'' \sigma'$ represents

$$\left\langle \sigma''(\boldsymbol{w}_r^T \boldsymbol{x}_i) \begin{pmatrix} w_{r0} x_{i0} \\ w_{r0} \boldsymbol{x}_{i1} - 2\boldsymbol{w}_{r1} \end{pmatrix}, \sigma'(\boldsymbol{w}_r^T \boldsymbol{x}_j) \begin{pmatrix} 1 \\ \boldsymbol{0}_{d+1} \end{pmatrix} \right\rangle, \left\langle \sigma'(\boldsymbol{w}_r^T \boldsymbol{x}_i) \begin{pmatrix} 1 \\ \boldsymbol{0}_{d+1} \end{pmatrix}, \sigma''(\boldsymbol{w}_r^T \boldsymbol{x}_j) \begin{pmatrix} w_{r0} x_{j0} \\ w_{r0} \boldsymbol{x}_{j1} - 2\boldsymbol{w}_{r1} \end{pmatrix} \right\rangle.$$

In fact, when bounding the corresponding terms for $H_{ij}(\boldsymbol{w}) - H_{ij}(0)$, the first four classes can be grouped into one category. They are of the form $f_1(\boldsymbol{w}) f_2(\boldsymbol{w}) f_3(\boldsymbol{w}) f_4(\boldsymbol{w})$, where for each $i$ ($1 \le i \le 4$), $f_i(\boldsymbol{w})$ is Lipschitz continuous with respect to $\|\cdot\|_2$ and $|f_i(\boldsymbol{w})| \lesssim \|\boldsymbol{w}\|_2$ (Note that $\sigma'(\cdot) = (\sigma''(\cdot))^2$). On the other hand, when $\|\boldsymbol{w}_1 - \boldsymbol{w}_2\|_2 \le R \le 1$, we can deduce that

$$|f_1(\boldsymbol{w}_1) f_2(\boldsymbol{w}_1) f_3(\boldsymbol{w}_1) f_4(\boldsymbol{w}_1) - f_1(\boldsymbol{w}_2) f_2(\boldsymbol{w}_2) f_3(\boldsymbol{w}_2) f_4(\boldsymbol{w}_2)| \lesssim R(\|\boldsymbol{w}_1\|_2^3 + 1).$$

Thus, for the terms in $H_{ij}(\boldsymbol{w}) - H_{ij}(0)$ that belong to the first four classes, we can deduce that they are less than $CR(\|\boldsymbol{w}_r(0)\|_2^3 + 1)$, where $C$ is a universal constant.

For the classes $\sigma''' \sigma''$ and $\sigma''' \sigma'$, they are both involving $\sigma'''$ that is not Lipschitz continuous. To make it precise, we write the class $\sigma''' \sigma''$ explicitly as follows.

$$\sigma''(\boldsymbol{w}_r^T \boldsymbol{x}_i) \sigma'''(\boldsymbol{w}_r^T \boldsymbol{x}_j) \|\boldsymbol{w}_{r1}\|_2^2 \begin{pmatrix} w_{r0} x_{i0} \\ w_{r0} \boldsymbol{x}_{i1} - 2\boldsymbol{w}_{r1} \end{pmatrix}^T \boldsymbol{x}_j.$$

Note that when $\|\boldsymbol{w}_r - \boldsymbol{w}_r(0)\|_2 < R$, we have that

$$|\sigma'''(\boldsymbol{w}_r^T \boldsymbol{x}_j) - \sigma'''(\boldsymbol{w}_r(0)^T \boldsymbol{x}_j)| = |I\{\boldsymbol{w}_r^T \boldsymbol{x}_j \ge 0\} - I\{\boldsymbol{w}_r(0)^T \boldsymbol{x}_j \ge 0\}| \le I\{A_{jr}\},$$

where the event $A_{jr}$ has been defined in (36).

Thus, we can deduce that for the terms in $H_{ij}(\boldsymbol{w}) - H_{ij}(0)$ that belong to the classes $\sigma''' \sigma''$ and $\sigma''' \sigma'$, they are less than

$$C \left[ (I\{A_{ir}\} + I\{A_{jr}\})(\|\boldsymbol{w}_r(0)\|_2^3 + 1) + R(\|\boldsymbol{w}_r(0)\|_2^3 + 1) \right],$$

where $C$ is a universal constant.

Similarly, for the last class $\sigma''' \sigma'''$ that are of the form

$$\sigma'''(\boldsymbol{w}_r^T \boldsymbol{x}_i) \sigma'''(\boldsymbol{w}_r^T \boldsymbol{x}_j) \|\boldsymbol{w}_{r1}\|_2^4 \boldsymbol{x}_i^T \boldsymbol{x}_j,$$

we can deduce that

$$|\sigma'''(\boldsymbol{w}_r^T\boldsymbol{x}_i)\sigma'''(\boldsymbol{w}_r^T\boldsymbol{x}_j)\|\boldsymbol{w}_{r1}\|_2^4\boldsymbol{x}_i^T\boldsymbol{x}_j - \sigma'''(\boldsymbol{w}_r(0)^T\boldsymbol{x}_i)\sigma'''(\boldsymbol{w}_r(0)^T\boldsymbol{x}_j)\|\boldsymbol{w}_{r1}(0)\|_2^4\boldsymbol{x}_i^T\boldsymbol{x}_j|$$
$$\lesssim I\{A_{ir} \vee A_{jr}\}\|\boldsymbol{w}_r(0)\|_2^4 + R(\|\boldsymbol{w}_r(0)\|_2^3 + 1).$$

Combining the upper bounds for the terms in the six classes, we have that

$$|H_{ij}(\boldsymbol{w}) - H_{ij}(0)| \lesssim \frac{1}{n_1}\left[\frac{1}{m}\left(R\sum_{r=1}^m \|\boldsymbol{w}_r(0)\|_2^3\right) + \frac{1}{m}\sum_{r=1}^m(I\{A_{ir}\} + I\{A_{jr}\})(\|\boldsymbol{w}_r(0)\|_2^4 + \|\boldsymbol{w}_r(0)\|_2^3 + 1) + R\right]$$

$$\lesssim \frac{1}{n_1}\left[\frac{1}{m}\left(R\sum_{r=1}^m \|\boldsymbol{w}_r(0)\|_2^4\right) + \frac{1}{m}\sum_{r=1}^m(I\{A_{ir}\} + I\{A_{jr}\})(\|\boldsymbol{w}_r(0)\|_2^4 + 1) + R\right], \tag{46}$$

where the last inequality follows from that $\|\boldsymbol{w}_r(0)\|_2^3 \lesssim \|\boldsymbol{w}_r(0)\|_2^4 + 1$ due to Young's inequality for products.

Now, we focus on the term $\frac{1}{m}\sum_{r=1}^m I\{A_{ir}\}\|\boldsymbol{w}_r(0)\|_2^4$.

Since

$$P\left(|w_{ri}(0)|^2 \ge 2\log\left(\frac{2}{\delta}\right)\right) \le \delta$$

and then

$$P\left(\|\boldsymbol{w}_r(0)\|_2^2 \ge 2(d+2)\log\left(\frac{2(d+2)}{\delta}\right)\right) \le \delta.$$

This implies that

$$P\left(\exists r \in [m], \|\boldsymbol{w}_r(0)\|_2^2 \ge 2(d+2)\log\left(\frac{2m(d+2)}{\delta}\right)\right) \le \delta. \tag{47}$$

Let $M = 2(d+2)\log\left(\frac{2m(d+2)}{\delta}\right)$, then

$$\frac{1}{m}\sum_{r=1}^m I\{A_{ir}\}\|\boldsymbol{w}_r(0)\|_2^4$$

$$= \frac{1}{m}\sum_{r=1}^m I\{A_{ir}\}\|\boldsymbol{w}_r(0)\|_2^4 I\{\|\boldsymbol{w}_r(0)\|_2^2 \le M\} + \frac{1}{m}\sum_{r=1}^m I\{A_{ir}\}\|\boldsymbol{w}_r(0)\|_2^4 I\{\|\boldsymbol{w}_r(0)\|_2^2 > M\}$$

$$\le \frac{M^2}{m}\sum_{r=1}^m I\{A_{ir}\} + \frac{1}{m}\sum_{r=1}^m \|\boldsymbol{w}_r(0)\|_2^4 I\{\|\boldsymbol{w}_r(0)\|_2^2 > M\}.$$

Applying Bernstein's inequality for the first term yields that with probability at least $1 - e^{-mR}$,

$$\frac{1}{m}\sum_{r=1}^m I\{A_{ir}\} \le 4R.$$

Moreover, from (47), we have that with probability at least $1 - \delta$, $I\{\|\boldsymbol{w}_r(0)\|_2^2 > M\} = 0$ holds for all $r \in [m]$.

Thus from (46), with probability at least $1 - \delta - n_1 e^{-mR}$, we have that for any $i \in [n_1]$ and $j \in [n_1]$,

$$|H_{ij}(\boldsymbol{w}) - H_{ij}(0)| \lesssim \frac{1}{n_1}\left[RM^2 + RM^2 + R\right]$$

$$\lesssim \frac{1}{n_1}M^2 R.$$

For $i \in [n_1], j \in [n_1+2, n_2]$ and $i \in [n_1+1, n_2], j \in [n_2]$, from the form of $\frac{\partial h_j(\boldsymbol{w})}{\partial \boldsymbol{w}_r}$, i.e.,

$$\frac{\partial h_j(\boldsymbol{w})}{\partial \boldsymbol{w}_r} = \frac{a_r}{\sqrt{n_2 m}}\sigma'(\boldsymbol{w}_r^T\boldsymbol{y}_j)\boldsymbol{y}_j,$$

770 we can obtain similar results for the terms $\left\langle \frac{\partial s_i}{\partial \boldsymbol{w}}, \frac{\partial h_j}{\partial \boldsymbol{w}} \right\rangle$ and $\left\langle \frac{\partial h_i}{\partial \boldsymbol{w}}, \frac{\partial h_j}{\partial \boldsymbol{w}} \right\rangle$.

772 With all results above, we have that with probability at least $1 - \delta - n_1 e^{-mR}$,

$$\|\boldsymbol{H}(\boldsymbol{w}) - \boldsymbol{H}(0)\|_F \lesssim M^2 R.$$

$\square$

### A.3. Proof of Lemma 3.7

*Proof.* First, we have

$$
\begin{aligned}
s_p(k+1) - s_p(k) &= \left[ s_p(k+1) - s_p(k) - \left\langle \frac{\partial s_p(k)}{\partial \boldsymbol{w}}, \boldsymbol{w}(k+1) - \boldsymbol{w}(k) \right\rangle \right] + \left\langle \frac{\partial s_p(k)}{\partial \boldsymbol{w}}, \boldsymbol{w}(k+1) - \boldsymbol{w}(k) \right\rangle \\
&:= I_1^p(k) + I_2^p(k).
\end{aligned}
\tag{48}
$$

For the second term $I_2^p(k)$, from the updating rule of gradient descent, we have that

$$
\begin{aligned}
I_2^p(k) &= \left\langle \frac{\partial s_p(k)}{\partial \boldsymbol{w}}, \boldsymbol{w}(k+1) - \boldsymbol{w}(k) \right\rangle \\
&= \left\langle \frac{\partial s_p(k)}{\partial \boldsymbol{w}}, -\eta \frac{\partial L(k)}{\partial \boldsymbol{w}} \right\rangle \\
&= -\sum_{r=1}^m \eta \left\langle \frac{\partial s_p(k)}{\partial \boldsymbol{w}_r}, \frac{\partial L(k)}{\partial \boldsymbol{w}_r} \right\rangle \\
&= -\sum_{r=1}^m \eta \left\langle \frac{\partial s_p(k)}{\partial \boldsymbol{w}_r}, \sum_{t=1}^{n_1} s_t(k) \frac{\partial s_t(k)}{\partial \boldsymbol{w}_r} + \sum_{j=1}^{n_2} h_j(k) \frac{\partial h_j(k)}{\partial \boldsymbol{w}_r} \right\rangle \\
&= -\eta \left[ \sum_{t=1}^{n_1} \left\langle \frac{\partial s_p(k)}{\partial \boldsymbol{w}_r}, \frac{\partial s_t(k)}{\partial \boldsymbol{w}_r} \right\rangle s_t(k) + \sum_{j=1}^{n_2} \left\langle \frac{\partial s_p(k)}{\partial \boldsymbol{w}_r}, \frac{\partial h_j(k)}{\partial \boldsymbol{w}_r} \right\rangle h_j(k) \right] \\
&= -\eta [\boldsymbol{H}(k)]_p \begin{pmatrix} \boldsymbol{s}(k) \\ \boldsymbol{h}(k) \end{pmatrix},
\end{aligned}
\tag{49}
$$

where $[\boldsymbol{H}(k)]_p$ denotes the $p$-row of $\boldsymbol{H}(k)$.

Similarly, for $h(k)$, we have

$$
\begin{aligned}
h_j(k+1) - h_j(k) &= \left[ h_j(k+1) - h_j(k) - \left\langle \frac{\partial h_j(k)}{\partial \boldsymbol{w}}, \boldsymbol{w}(k+1) - \boldsymbol{w}(k) \right\rangle \right] + \left\langle \frac{\partial h_j(k)}{\partial \boldsymbol{w}}, \boldsymbol{w}(k+1) - \boldsymbol{w}(k) \right\rangle \\
&:= I_1^{n_1+j}(k) + I_2^{n_1+j}(k)
\end{aligned}
\tag{50}
$$

and

$$
I_2^{n_1+j}(k) = -\eta [\boldsymbol{H}(k)]_{n_1+j} \begin{pmatrix} \boldsymbol{s}(k) \\ \boldsymbol{h}(k) \end{pmatrix}.
\tag{51}
$$

Combining (48), (49), (50) and (51) yields that

$$
\begin{aligned}
\begin{pmatrix} \boldsymbol{s}(k+1) \\ \boldsymbol{h}(k+1) \end{pmatrix} - \begin{pmatrix} \boldsymbol{s}(k) \\ \boldsymbol{h}(k) \end{pmatrix} &= \boldsymbol{I}_1(k) + \boldsymbol{I}_2(k) \\
&= \boldsymbol{I}_1(k) - \eta \boldsymbol{H}(k) \begin{pmatrix} \boldsymbol{s}(k) \\ \boldsymbol{h}(k) \end{pmatrix}.
\end{aligned}
$$

A simple transformation directly leads to

$$
\begin{pmatrix} \boldsymbol{s}(k+1) \\ \boldsymbol{h}(k+1) \end{pmatrix} = (\boldsymbol{I} - \eta \boldsymbol{H}(k)) \begin{pmatrix} \boldsymbol{s}(k) \\ \boldsymbol{h}(k) \end{pmatrix} + \boldsymbol{I}_1(k).
$$

$\square$

### A.4. Proof of Theorem 3.8

*Proof.* For the sake of completeness in the proof, we restate Condition 1 and Corollary 3.11 from the main text, and label them as Condition 3 and Corollary A.1, respectively.

*Condition* 3. At the $t$-th iteration, we have that for each $r \in [m]$, $\|\boldsymbol{w}_r(t)\|_2 \leq B$ and

$$L(t) \leq \left(1 - \frac{\eta\lambda_0}{2}\right)^t L(0), \tag{52}$$

where $B = \sqrt{2(d+2)\log\left(\frac{2m(d+2)}{\delta}\right)} + 1$ and $L(k)$ is an abbreviation of $L(\boldsymbol{w}(k))$.

From (47), we know that with probability at least $1 - \delta$, $\|\boldsymbol{w}_r(0)\|_2 \leq \sqrt{2(d+2)\log\left(\frac{2m(d+2)}{\delta}\right)}$ holds for all $r \in [m]$. Thus, if we can prove that $\boldsymbol{w}_r(t)$ is closed enough to $\boldsymbol{w}_r(0)$, then $\|\boldsymbol{w}_r(t)\|_2 \leq B$ holds.

**Corollary A.1** (Lemma 4.1 in (Gao et al., 2023)). *If Condition 3 holds for $t = 0, \cdots, k$, then we have for every $r \in [m]$,*

$$\|\boldsymbol{w}_r(k+1) - \boldsymbol{w}_r(0)\|_2 \leq \frac{CB^2\sqrt{L(0)}}{\sqrt{m}\lambda_0} := R^{'}, \tag{53}$$

*where $C$ is a universal constant.*

Corollary A.1 implies that when $m$ is large enough, we have $\|\boldsymbol{w}_r(k+1) - \boldsymbol{w}_r(0)\|_2 \leq 1$ and then $\|\boldsymbol{w}_r(k+1)\|_2 \leq B$. Thus, in induction, we only need to prove that (52) also holds for $t = k+1$, which relies on the recursion formula (14).

Recall that the recursion formula is

$$\begin{pmatrix} \boldsymbol{s}(k+1) \\ \boldsymbol{h}(k+1) \end{pmatrix} = (\boldsymbol{I} - \eta\boldsymbol{H}(k)) \begin{pmatrix} \boldsymbol{s}(k) \\ \boldsymbol{h}(k) \end{pmatrix} + \boldsymbol{I}_1(k).$$

From Corollary A.1 and Lemma 3.5, taking $CM^2R < \frac{\lambda_0}{4}$ in (12) and $R^{'} \leq R$ in (53) yields that $\lambda_{min}(\boldsymbol{H}(k)) \geq \lambda_{min}(\boldsymbol{H}(0)) - \frac{\lambda_0}{4} \geq \frac{\lambda_0}{2}$ and

$$\|\boldsymbol{H}(k)\|_2 \leq \|\boldsymbol{H}(0)\|_2 + \frac{\lambda_0}{4} \leq \|\boldsymbol{H}^\infty\|_2 + \frac{\lambda_0}{2} \leq \frac{3}{2}\|\boldsymbol{H}^\infty\|_2.$$

Therefore, if we take $\eta \leq \frac{2}{3}\frac{1}{\|\boldsymbol{H}^\infty\|_2}$, then $\boldsymbol{I} - \eta\boldsymbol{H}(k)$ is positive definite and $\|I - \eta\boldsymbol{H}(k)\|_2 \leq 1 - \frac{\eta\lambda_0}{2}$.

Combining these facts with the recursion formula, we have that

$$\left\|\begin{pmatrix} \boldsymbol{s}(k+1) \\ \boldsymbol{h}(k+1) \end{pmatrix}\right\|_2^2$$

$$= \left\|(\boldsymbol{I} - \eta\boldsymbol{H}(k))\begin{pmatrix} \boldsymbol{s}(k) \\ \boldsymbol{h}(k) \end{pmatrix}\right\|_2^2 + \|\boldsymbol{I}_1(k)\|_2^2 + 2\left\langle(\boldsymbol{I} - \eta\boldsymbol{H}(k))\begin{pmatrix} \boldsymbol{s}(k) \\ \boldsymbol{h}(k) \end{pmatrix}, \boldsymbol{I}_1(k)\right\rangle \tag{54}$$

$$\leq \left(1 - \frac{\eta\lambda_0}{2}\right)^2\left\|\begin{pmatrix} \boldsymbol{s}(k) \\ \boldsymbol{h}(k) \end{pmatrix}\right\|_2^2 + \|\boldsymbol{I}_1(k)\|_2^2 + 2\left(1 - \frac{\eta\lambda_0}{2}\right)\left\|\begin{pmatrix} \boldsymbol{s}(k) \\ \boldsymbol{h}(k) \end{pmatrix}\right\|_2 \|\boldsymbol{I}_1(k)\|_2.$$

Thus, it remains only to bound $\|\boldsymbol{I}_1(k)\|_2$.

For $\boldsymbol{I}_1(k)$, recall that $\boldsymbol{I}_1(k) = (I_1^1(k), \cdots, I_1^{n_1}(k), I_1^{n_1+1}(k), \cdots, I_1^{n_1+n_2}(k))^T \in \mathbb{R}^{n_1+n_2}$ and for $p \in [n_1]$,

$$I_1^p(k) = s_p(k+1) - s_p(k) - \left\langle\frac{\partial s_p(k)}{\partial\boldsymbol{w}}, \boldsymbol{w}(k+1) - \boldsymbol{w}(k)\right\rangle,$$

for $j \in [n_2]$,

$$I_1^{n_1+j}(k) = h_j(k+1) - h_j(k) - \left\langle\frac{\partial h_j(k)}{\partial\boldsymbol{w}}, \boldsymbol{w}(k+1) - \boldsymbol{w}(k)\right\rangle.$$

Recall that

$$s_p(k) = \frac{1}{\sqrt{n_1}} \left( \frac{1}{\sqrt{m}} \left( \sum_{r=1}^{m} a_r \sigma'(\boldsymbol{w}_r(k)^T x_p) w_{r0}(k) - a_r \sigma''(\boldsymbol{w}_r(k)^T x_p) \|\boldsymbol{w}_{r1}(k)\|_2^2 \right) - f(x_p) \right)$$

and

$$\frac{\partial s_p(k)}{\partial \boldsymbol{w}_r} = \frac{a_r}{\sqrt{n_1 m}} \left[ \sigma''(\boldsymbol{w}_r(k)^T \boldsymbol{x}_p) w_{r0}(k) \boldsymbol{x}_p + \sigma'(\boldsymbol{w}_r(k)^T \boldsymbol{x}_p) \begin{pmatrix} 1 \\ \boldsymbol{0}_{d+2} \end{pmatrix} - \sigma'''(\boldsymbol{w}_r(k)^T \boldsymbol{x}_p) \|\boldsymbol{w}_{r1}(k)\|_2^2 \boldsymbol{x}_p \right.$$
$$\left. -2\sigma''(\boldsymbol{w}_r(k)^T \boldsymbol{x}_p) \begin{pmatrix} 0 \\ \boldsymbol{w}_{r1}(k) \end{pmatrix} \right].$$

Define $\chi_{pr}^1(k) := \sigma'(\boldsymbol{w}_r(k)^T \boldsymbol{x}_p) w_{r0}(k)$ and $\chi_{pr}^2(k) := \sigma''(\boldsymbol{w}_r(k)^T \boldsymbol{x}_p) \|\boldsymbol{w}_{r1}(k)\|_2^2$, i.e., $\chi_{pr}^1(k)$ and $\chi_{pr}^2(k)$ are related to the operators $\frac{\partial u}{\partial t}$ and $\Delta u$ respectively.

Then define

$$\hat{\chi}_{pr}^1(k) = \chi_{pr}^1(k+1) - \chi_{pr}^1(k) - \left\langle \frac{\partial \chi_{pr}^1(k)}{\partial \boldsymbol{w}_r}, \boldsymbol{w}_r(k+1) - \boldsymbol{w}_r(k) \right\rangle$$

and

$$\hat{\chi}_{pr}^2(k) = \chi_{pr}^2(k+1) - \chi_{pr}^2(k) - \left\langle \frac{\partial \chi_{pr}^2(k)}{\partial \boldsymbol{w}_r}, \boldsymbol{w}_r(k+1) - \boldsymbol{w}_r(k) \right\rangle.$$

At this time, we have

$$I_1^p(k) = \frac{1}{\sqrt{n_1 m}} \sum_{r=1}^{m} a_r \left[ \hat{\chi}_{pr}^1(k) - \hat{\chi}_{pr}^2(k) \right].$$

The purpose of defining $\hat{\chi}_{pr}^1(k)$ and $\hat{\chi}_{pr}^1(k)$ in this way is to enable us to handle the terms related to the operators $\frac{\partial u}{\partial t}$ and $\Delta u$ separately.

We first recall some definitions. For $p \in [n_1]$,

$$A_{p,r} = \{\exists \boldsymbol{w} : \|\boldsymbol{w} - \boldsymbol{w}_r(0)\|_2 \le R, I\{\boldsymbol{w}^T \boldsymbol{x}_p \ge 0\} \ne I\{\boldsymbol{w}_r(0)^T \boldsymbol{x}_p \ge 0\}\}$$

and $S_p = \{r \in [m] : I\{A_{p,r} = 0\}\}$, $S_p^\perp = [n_1] \backslash S_p$.

In the following, we are going to show that $|\hat{\chi}_{pr}^1(k)| = \mathcal{O}(\|\boldsymbol{w}_r(k+1) - \boldsymbol{w}_r(k)\|_2^2)$ for every $r \in [m]$ and $|\hat{\chi}_{pr}^2(k)| = \mathcal{O}(\|\boldsymbol{w}_r(k+1) - \boldsymbol{w}_r(k)\|_2^2)$ for $r \in S_p$, $|\hat{\chi}_{pr}^2(k)| = \mathcal{O}(\|\boldsymbol{w}_r(k+1) - \boldsymbol{w}_r(k)\|_2)$ for $r \in S_p^\perp$. Thus, we can prove that $\|\boldsymbol{I}_1(k)\|_2 = \mathcal{O}\left( \frac{\sqrt{L(k)}}{\sqrt{m}} \right)$. Then combining with (69) leads to the conclusion.

For $\hat{\chi}_{pr}^1(k)$, from its definition, we have that

$$\hat{\chi}_{pr}^1(k) = \sigma'(\boldsymbol{w}_r(k+1)^T \boldsymbol{x}_p) w_{r0}(k+1) - \sigma'(\boldsymbol{w}_r(k)^T \boldsymbol{x}_p) w_{r0}(k)$$
$$- \langle \boldsymbol{w}_r(k+1) - \boldsymbol{w}_r(k), \boldsymbol{x}_p \rangle \sigma''(\boldsymbol{w}_r(k)^T \boldsymbol{x}_p) w_{r0}(k) - (w_{r0}(k+1) - w_{r0}(k)) \sigma'(\boldsymbol{w}_r(k)^T \boldsymbol{x}_p)$$
$$= (\sigma'(\boldsymbol{w}_r(k+1)^T \boldsymbol{x}_p) - \sigma'(\boldsymbol{w}_r(k)^T \boldsymbol{x}_p)) w_{r0}(k+1) - \langle \boldsymbol{w}_r(k+1) - \boldsymbol{w}_r(k), \boldsymbol{x}_p \rangle \sigma''(\boldsymbol{w}_r(k)^T \boldsymbol{x}_p) w_{r0}(k).$$

From the mean value theorem, we can deduce that there exists $\zeta(k) \in \mathbb{R}$ such that

$$\sigma'(\boldsymbol{w}_r(k+1)^T \boldsymbol{x}_p) - \sigma'(\boldsymbol{w}_r(k)^T \boldsymbol{x}_p) = \sigma''(\zeta(k)) \langle \boldsymbol{w}_r(k+1) - \boldsymbol{w}_r(k), \boldsymbol{x}_p \rangle$$

and

$$|\sigma''(\zeta(k)) - \sigma''(\boldsymbol{w}_r(k)^T \boldsymbol{x}_p)| \le |\zeta(k) - \boldsymbol{w}_r(k)^T \boldsymbol{x}_p|$$
$$\le \sqrt{2} \|\boldsymbol{w}_r(k+1) - \boldsymbol{w}_r(k)\|_2.$$

Then, for $\hat{\chi}_{pr}^1(k)$, we can rewrite it as follows.

$$\hat{\chi}_{pr}^1(k) = \sigma^{''}(\zeta(k))\langle \boldsymbol{w}_r(k+1) - \boldsymbol{w}_r(k), \boldsymbol{x}_p \rangle w_{r0}(k+1) - \langle \boldsymbol{w}_r(k+1) - \boldsymbol{w}_r(k), \boldsymbol{x}_p \rangle \sigma^{''}(\boldsymbol{w}_r(k)^T \boldsymbol{x}_p) w_{r0}(k)$$

$$= \left[ \left( \sigma^{''}(\zeta(k)) - \sigma^{''}(\boldsymbol{w}_r(k)^T \boldsymbol{x}_p) \right) \langle \boldsymbol{w}_r(k+1) - \boldsymbol{w}_r(k), \boldsymbol{x}_p \rangle w_{r0}(k+1) \right]$$

$$+ \left[ \langle \boldsymbol{w}_r(k+1) - \boldsymbol{w}_r(k), \boldsymbol{x}_p \rangle \sigma^{''}(\boldsymbol{w}_r(k)^T \boldsymbol{x}_p)(w_{r0}(k+1) - w_{r0}(k)) \right].$$

This implies that

$$|\hat{\chi}_{pr}^1(k)| \lesssim B \|\boldsymbol{w}_r(k+1) - \boldsymbol{w}_r(k)\|_2^2.$$

For $\hat{\chi}_{pr}^2(k)$, we write it as follows explicitly.

$$\hat{\chi}_{pr}^2(k) = \sigma^{''}(\boldsymbol{w}_r(k+1)^T \boldsymbol{x}_p)\|\boldsymbol{w}_{r1}(k+1)\|_2^2 - \sigma^{''}(\boldsymbol{w}_r(k)^T \boldsymbol{x}_p)\|\boldsymbol{w}_{r1}(k)\|_2^2$$

$$- \langle \boldsymbol{w}_r(k+1) - \boldsymbol{w}_r(k), \boldsymbol{x}_p \rangle \sigma^{'''}(\boldsymbol{w}_r(k)^T \boldsymbol{x}_p)\|\boldsymbol{w}_{r1}(k)\|_2^2 \tag{55}$$

$$- 2\langle \boldsymbol{w}_{r1}(k+1) - \boldsymbol{w}_{r1}(k), \boldsymbol{w}_{r1}(k) \rangle \sigma^{''}(\boldsymbol{w}_r(k)^T \boldsymbol{x}_p).$$

Note that for the term $\sigma^{''}(\boldsymbol{w}_r(k)^T \boldsymbol{w}_p)\|\boldsymbol{w}_{r1}(k)\|_2^2$, we can rewrite it as follows.

$$\sigma^{''}(\boldsymbol{w}_r(k)^T \boldsymbol{x}_p)\|\boldsymbol{w}_{r1}(k)\|_2^2$$

$$= \sigma^{''}(\boldsymbol{w}_r(k)^T \boldsymbol{x}_p)\|\boldsymbol{w}_{r1}(k) - \boldsymbol{w}_{r1}(k+1) + \boldsymbol{w}_{r1}(k+1)\|_2^2 \tag{56}$$

$$= \sigma^{''}(\boldsymbol{w}_r(k)^T \boldsymbol{x}_p)[\|\boldsymbol{w}_{r1}(k) - \boldsymbol{w}_{r1}(k+1)\|_2^2 + \|\boldsymbol{w}_{r1}(k+1)\|_2^2 - 2\langle \boldsymbol{w}_{r1}(k+1) - \boldsymbol{w}_{r1}(k), \boldsymbol{w}_{r1}(k+1) \rangle],$$

where the first term $\sigma^{''}(\boldsymbol{w}_r(k)^T \boldsymbol{x}_p)\|\boldsymbol{w}_{r1}(k) - \boldsymbol{w}_{r1}(k+1)\|_2^2 = \mathcal{O}(B\|\boldsymbol{w}_r(k+1) - \boldsymbol{w}_r(k)\|_2^2)$.

Plugging (56) into (55) yields that

$$\hat{\chi}_{pr}^2(k) = [\sigma^{''}(\boldsymbol{w}_r(k+1)^T \boldsymbol{x}_p) - \sigma^{''}(\boldsymbol{w}_r(k)^T \boldsymbol{x}_p)]\|\boldsymbol{w}_{r1}(k+1)\|_2^2$$

$$- \langle \boldsymbol{w}_r(k+1) - \boldsymbol{w}_r(k), \boldsymbol{x}_p \rangle \sigma^{'''}(\boldsymbol{w}_r(k)^T \boldsymbol{x}_p)\|\boldsymbol{w}_{r1}(k)\|_2^2$$

$$+ 2\langle \boldsymbol{w}_{r1}(k+1) - \boldsymbol{w}_{r1}(k), \boldsymbol{w}_{r1}(k+1) - \boldsymbol{w}_{r1}(k) \rangle \sigma^{''}(\boldsymbol{w}_r(k)^T \boldsymbol{x}_p) + \mathcal{O}(B\|\boldsymbol{w}_r(k+1) - \boldsymbol{w}_r(k)\|_2^2)$$

$$= [\sigma^{''}(\boldsymbol{w}_r(k+1)^T \boldsymbol{x}_p) - \sigma^{''}(\boldsymbol{w}_r(k)^T \boldsymbol{x}_p) - \langle \boldsymbol{w}_r(k+1) - \boldsymbol{w}_r(k), \boldsymbol{x}_p \rangle \sigma^{'''}(\boldsymbol{w}_r(k)^T \boldsymbol{x}_p)]\|\boldsymbol{w}_{r1}(k+1)\|_2^2 \tag{57}$$

$$+ \langle \boldsymbol{w}_r(k+1) - \boldsymbol{w}_r(k), \boldsymbol{x}_p \rangle \sigma^{'''}(\boldsymbol{w}_r(k)^T \boldsymbol{x}_p)(\|\boldsymbol{w}_{r1}(k+1)\|_2^2 - \|\boldsymbol{w}_{r1}(k)\|_2^2)$$

$$+ \mathcal{O}(B\|\boldsymbol{w}_r(k+1) - \boldsymbol{w}_r(k)\|_2^2)$$

$$= \left[ \sigma^{''}(\boldsymbol{w}_r(k+1)^T \boldsymbol{x}_p) - \sigma^{''}(\boldsymbol{w}_r(k)^T \boldsymbol{x}_p) - \langle \boldsymbol{w}_r(k+1) - \boldsymbol{w}_r(k), \boldsymbol{x}_p \rangle \sigma^{'''}(\boldsymbol{w}_r(k)^T \boldsymbol{x}_p) \right] \|\boldsymbol{w}_{r1}(k+1)\|_2^2$$

$$+ \mathcal{O}(B\|\boldsymbol{w}_r(k+1) - \boldsymbol{w}_r(k)\|_2^2).$$

Thus, we only need to consider the term

$$\sigma^{''}(\boldsymbol{w}_r(k+1)^T \boldsymbol{x}_p) - \sigma^{''}(\boldsymbol{w}_r(k)^T \boldsymbol{x}_p) - \langle \boldsymbol{w}_r(k+1) - \boldsymbol{w}_r(k), \boldsymbol{x}_p \rangle \sigma^{'''}(\boldsymbol{w}_r(k)^T \boldsymbol{x}_p).$$

For $r \in S_p$, since $\|\boldsymbol{w}_r(k+1) - \boldsymbol{w}_r(0)\|_2 \leq R, \|\boldsymbol{w}_r(k) - \boldsymbol{w}_r(0)\|_2 \leq R$, we have that $I\{\boldsymbol{w}_r(k+1)^T \boldsymbol{x}_p \geq 0\} = I\{\boldsymbol{w}_r(k)^T \boldsymbol{x}_p \geq 0\}$, which yields that

$$\sigma^{''}(\boldsymbol{w}_r(k+1)^T \boldsymbol{x}_p) - \sigma^{''}(\boldsymbol{w}_r(k)^T \boldsymbol{x}_p) - \langle \boldsymbol{w}_r(k+1) - \boldsymbol{w}_r(k), \boldsymbol{x}_p \rangle \sigma^{'''}(\boldsymbol{w}_r(k)^T \boldsymbol{x}_p)$$

$$= [(\boldsymbol{w}_r(k+1)^T \boldsymbol{x}_p)I\{\boldsymbol{w}_r(k+1)^T \boldsymbol{x}_p \geq 0\} - (\boldsymbol{w}_r(k)^T \boldsymbol{x}_p)I\{\boldsymbol{w}_r(k)^T \boldsymbol{x}_p \geq 0\}]$$

$$- \langle \boldsymbol{w}_r(k+1) - \boldsymbol{w}_r(k), \boldsymbol{x}_p \rangle I\{\boldsymbol{w}_r(k)^T \boldsymbol{x}_p \geq 0\} \tag{58}$$

$$= [(\boldsymbol{w}_r(k+1)^T \boldsymbol{x}_p)I\{\boldsymbol{w}_r(k)^T \boldsymbol{x}_p \geq 0\} - (\boldsymbol{w}_r(k)^T \boldsymbol{x}_p)I\{\boldsymbol{w}_r(k)^T \boldsymbol{x}_p \geq 0\}]$$

$$- \langle \boldsymbol{w}_r(k+1) - \boldsymbol{w}_r(k), \boldsymbol{x}_p \rangle I\{\boldsymbol{w}_r(k)^T \boldsymbol{x}_p \geq 0\}$$

$$= 0.$$

For $r \in S_p^{\perp}$, the Lipschitz continuity of $\sigma''$ implies that

$$\sigma''(\boldsymbol{w}_r(k+1)^T\boldsymbol{x}_p) - \sigma''(\boldsymbol{w}_r(k)^T\boldsymbol{x}_p) - \langle \boldsymbol{w}_r(k+1) - \boldsymbol{w}_r(k), \boldsymbol{x}_p\rangle\sigma'''(\boldsymbol{w}_r(k)^T\boldsymbol{x}_p) = \mathcal{O}(\|\boldsymbol{w}_r(k+1) - \boldsymbol{w}_r(k)\|_2). \quad (59)$$

Combining (57), (58) and (59), we can deduce that for $r \in S_p$,

$$|\hat{\chi}_{pr}^2(k)| \lesssim B\|\boldsymbol{w}_r(k+1) - \boldsymbol{w}_r(k)\|_2^2$$

and for $r \in S_p^{\perp}$,

$$|\hat{\chi}_{pr}^2(k)| \lesssim B\|\boldsymbol{w}_r(k+1) - \boldsymbol{w}_r(k)\|_2^2 + B^2\|\boldsymbol{w}_r(k+1) - \boldsymbol{w}_r(k)\|_2.$$

With the estimations for $\hat{\chi}_{pr}^1(k)$ and $\hat{\chi}_{pr}^2(k)$, we have

$$\begin{aligned}
|I_1^p(k)| &\leq \frac{1}{\sqrt{n_1 m}} \sum_{r=1}^m (|\hat{\chi}_{pr}^1(k)| + |\hat{\chi}_{pr}^2(k)|) \\
&\lesssim \frac{1}{\sqrt{n_1 m}} \sum_{r=1}^m B\|\boldsymbol{w}_r(k+1) - \boldsymbol{w}_r(k)\|_2^2 + \frac{1}{\sqrt{n_1 m}} \sum_{r \in S_p^{\perp}} B^2\|\boldsymbol{w}_r(k+1) - \boldsymbol{w}_r(k)\|_2.
\end{aligned} \quad (60)$$

For $j \in [n_2]$, we consider $I_1^{n_1+j}(k)$, which can be written as follows.

$$\begin{aligned}
I_1^{n_1+j}(k) &= h_j(k+1) - h_j(k) - \left\langle \boldsymbol{w}(k+1) - \boldsymbol{w}(k), \frac{\partial h_j(k)}{\partial \boldsymbol{w}} \right\rangle \\
&= \sum_{r=1}^m \frac{a_r}{\sqrt{n_2 m}} \left[ \sigma(\boldsymbol{w}_r(k+1)^T\boldsymbol{y}_j) - \sigma(\boldsymbol{w}_r(k)^T\boldsymbol{y}_j) - \langle \boldsymbol{w}_r(k+1) - \boldsymbol{w}_r(k), \boldsymbol{y}_j\rangle\sigma'(\boldsymbol{w}_r(k)^T\boldsymbol{y}_j) \right].
\end{aligned}$$

From the mean value theorem, we have that there exists $\zeta(k) \in \mathbb{R}$ such that

$$\sigma(\boldsymbol{w}_r(k+1)^T\boldsymbol{y}_j) - \sigma(\boldsymbol{w}_r(k)^T\boldsymbol{y}_j) = \sigma'(\zeta(k))\langle \boldsymbol{w}_r(k+1) - \boldsymbol{w}_r(k), \boldsymbol{y}_j\rangle$$

and

$$\begin{aligned}
|\sigma'(\zeta(k)) - \sigma'(\boldsymbol{w}_r(k)^T\boldsymbol{y}_j)| &\leq 2B|\zeta(k) - \boldsymbol{w}_r(k)^T\boldsymbol{y}_j| \\
&\leq 2\sqrt{2}B\|\boldsymbol{w}_r(k+1) - \boldsymbol{w}_r(k)\|_2.
\end{aligned}$$

Thus,

$$\begin{aligned}
&|\sigma(\boldsymbol{w}_r(k+1)^T\boldsymbol{y}_j) - \sigma(\boldsymbol{w}_r(k)^T\boldsymbol{y}_j) - \langle \boldsymbol{w}_r(k+1) - \boldsymbol{w}_r(k), \boldsymbol{y}_j\rangle\sigma'(\boldsymbol{w}_r(k)^T\boldsymbol{y}_j)| \\
&= |\sigma'(\zeta(k))\langle \boldsymbol{w}_r(k+1) - \boldsymbol{w}_r(k), \boldsymbol{y}_j\rangle - \sigma(\boldsymbol{w}_r(k)^T\boldsymbol{y}_j) - \langle \boldsymbol{w}_r(k+1) - \boldsymbol{w}_r(k), \boldsymbol{y}_j\rangle\sigma'(\boldsymbol{w}_r(k)^T\boldsymbol{y}_j)| \\
&= |(\sigma'(\zeta(k)) - \sigma'(\boldsymbol{w}_r(k)^T\boldsymbol{y}_j))\langle \boldsymbol{w}_r(k+1) - \boldsymbol{w}_r(k), \boldsymbol{y}_j\rangle| \\
&\lesssim B\|\boldsymbol{w}_r(k+1) - \boldsymbol{w}_r(k)\|_2.
\end{aligned}$$

Therefore, for $j \in [n_2]$,

$$|I_1^{n_1+j}(k)| \lesssim \frac{B}{\sqrt{n_2 m}} \sum_{r=1}^m \|\boldsymbol{w}_r(k+1) - \boldsymbol{w}_r(k)\|_2^2. \quad (61)$$

From the updating rule of gradient descent, we can deduce that for every $r \in [m]$,

$$\|\boldsymbol{w}_r(k+1) - \boldsymbol{w}_r(k)\|_2 = \left\| -\eta \frac{\partial L(k)}{\partial \boldsymbol{w}_r} \right\|_2 \lesssim \frac{\eta B^2}{\sqrt{m}}\sqrt{L(k)}. \quad (62)$$

Plugging (62) into (61) and (60), we can deduce that

$$|I_1^p(k)| \lesssim \frac{B}{\sqrt{n_1 m}} \sum_{r=1}^{m} \|\boldsymbol{w}_r(k+1) - \boldsymbol{w}_r(k)\|_2^2 + \frac{B^2}{\sqrt{n_1 m}} \sum_{r \in S_p^\perp} \|\boldsymbol{w}_r(k+1) - \boldsymbol{w}_r(k)\|_2$$

$$\lesssim \frac{B}{\sqrt{n_1 m}} \sum_{r=1}^{m} \frac{\eta^2 B^4}{m} L(k) + \frac{B^2}{\sqrt{n_1 m}} \sum_{r \in S_p^\perp} \frac{\eta B^2}{\sqrt{m}} \sqrt{L(k)}$$

$$= \frac{\eta^2 B^5 L(k)}{\sqrt{n_1 m}} + \frac{\eta B^4 \sqrt{L(k)}}{\sqrt{n_1}} \frac{1}{m} \sum_{r=1}^{m} I\{r \in S_p^\perp\} \tag{63}$$

$$\leq \frac{\eta^2 B^5 \sqrt{L(0)} \sqrt{L(k)}}{\sqrt{n_1 m}} + \frac{\eta B^4 \sqrt{L(k)}}{\sqrt{n_1}} \frac{1}{m} \sum_{r=1}^{m} I\{r \in S_p^\perp\}$$

and

$$|I_1^{n_1+j}(k)| \lesssim \frac{B}{\sqrt{n_2 m}} \sum_{r=1}^{m} \|\boldsymbol{w}_r(k+1) - \boldsymbol{w}_r(k)\|_2^2$$

$$\lesssim \frac{B}{\sqrt{n_2 m}} \sum_{r=1}^{m} \frac{\eta^2 B^4}{m} L(k) \tag{64}$$

$$\leq \frac{\eta^2 B^5 \sqrt{L(0)} \sqrt{L(k)}}{\sqrt{n_2 m}}.$$

Note that

$$P(A_{p,r}) \leq \frac{2R}{\sqrt{2\pi}}, \ S_p = \{r \in [m] : I\{A_{p,r}\} = 0\}.$$

Thus, from Bernstein's inequality, we have that with probability at least $1 - e^{-mR}$,

$$\frac{1}{m} \sum_{r=1}^{m} I\{r \in S_p^\perp\} = \frac{1}{m} \sum_{r=1}^{m} I\{A_{pr}\} \lesssim 4R.$$

Then the inequality holds for all $p \in [n_1]$ with probability at least $1 - n_1 e^{-mR}$. Plugging this into (63), we can conclude that for every $p \in [n_1]$

$$|I_1^p(k)| \lesssim \frac{\eta^2 B^5 \sqrt{L(0)} \sqrt{L(k)}}{\sqrt{n_1 m}} + \frac{\eta B^4 \sqrt{L(k)}}{\sqrt{n_1}} R. \tag{65}$$

Combining (64) and (65), we have that

$$\|\boldsymbol{I}_1(k)\|_2 = \sqrt{\sum_{p=1}^{n_1} |I_1^p(k)|^2 + \sum_{j=1}^{n_2} |I_1^{n_1+j}(k)|^2}$$

$$\lesssim \frac{\eta^2 B^5 \sqrt{L(0)} \sqrt{L(k)}}{\sqrt{m}} + \eta B^4 \sqrt{L(k)} R.$$

Plugging this into (54) yields that

$$\left\| \begin{pmatrix} \boldsymbol{s}(k+1) \\ \boldsymbol{h}(k+1) \end{pmatrix} \right\|_2^2$$

$$\leq \left(1 - \frac{\eta \lambda_0}{2}\right)^2 \left\| \begin{pmatrix} \boldsymbol{s}(k) \\ \boldsymbol{h}(k) \end{pmatrix} \right\|_2^2 + \|\boldsymbol{I}_1(k)\|_2^2 + 2\left(1 - \frac{\eta \lambda_0}{2}\right) \left\| \begin{pmatrix} \boldsymbol{s}(k) \\ \boldsymbol{h}(k) \end{pmatrix} \right\|_2 \|\boldsymbol{I}_1(k)\|_2$$

$$\leq \left[ \left(1 - \frac{\eta \lambda_0}{2}\right)^2 + C^2 \left( \frac{\eta^2 B^5 \sqrt{L(0)}}{\sqrt{m}} + \eta B^4 R \right)^2 + 2C \left( \frac{\eta^2 B^5 \sqrt{L(0)}}{\sqrt{m}} + \eta B^4 R \right) \right] \left\| \begin{pmatrix} \boldsymbol{s}(k) \\ \boldsymbol{h}(k) \end{pmatrix} \right\|_2^2$$

$$\leq \left(1 - \frac{\eta \lambda_0}{2}\right) \left\| \begin{pmatrix} \boldsymbol{s}(k) \\ \boldsymbol{h}(k) \end{pmatrix} \right\|_2^2,$$

where $C$ is a universal constant and the last inequality requires that

$$\frac{\eta^2 B^5 \sqrt{L(0)}}{\sqrt{m}} \lesssim \eta\lambda_0, \ \eta B^4 R \lesssim \eta\lambda_0.$$

Recall that we also require $CM^2 R < \frac{\lambda_0}{4}$ for $R$ in (12) and

$$R^{'} = \frac{CB^2\sqrt{L(0)}}{\sqrt{m}\lambda_0} < R$$

for $R^{'}$ in (53) to make sure $\|\boldsymbol{H}(k) - \boldsymbol{H}(0)\|_2 \leq \frac{\lambda_0}{4}$.

Finally, with $R = \mathcal{O}(\frac{\lambda_0}{M^2})$ and Lemma C.4 for the upper bound of $L(0)$, $m$ needs to satisfies that

$$m = \Omega\left(\frac{M^4 B^4 L(0)}{\lambda_0^4}\right) = \Omega\left(\frac{d^8}{\lambda_0^4}\log^6\left(\frac{md}{\delta}\right)\log\left(\frac{n_1 + n_2}{\delta}\right)\right).$$

$\square$

# B. Proof of Section 4

## B.1. Proof of Lemma 4.3

*Proof.* Recall that

$$\boldsymbol{H}(\boldsymbol{w}) = \boldsymbol{D}^T\boldsymbol{D}, \quad \boldsymbol{D} = \left[\frac{\partial s_1(\boldsymbol{w})}{\partial\boldsymbol{w}}, \cdots, \frac{\partial s_{n_1}(\boldsymbol{w})}{\partial\boldsymbol{w}}, \frac{\partial h_1(\boldsymbol{w})}{\partial\boldsymbol{w}}, \cdots, \frac{\partial h_{n_2}(\boldsymbol{w})}{\partial\boldsymbol{w}}\right],$$

and $\boldsymbol{H}^\infty = \mathbb{E}_{\boldsymbol{w}\sim\mathcal{N}(\boldsymbol{0},\boldsymbol{I})}\boldsymbol{G}(\boldsymbol{w})$.

We denote $\varphi(\boldsymbol{x};\boldsymbol{w}) = \sigma^{'}(\boldsymbol{w}^T\boldsymbol{x})w_0 - \sigma^{''}(\boldsymbol{w}^T\boldsymbol{x})\|\boldsymbol{w}_1\|_2^2$, where $\boldsymbol{w} = (w_0, \boldsymbol{w}_1^T)^T$, $w_0 \in \mathbb{R}$, $\boldsymbol{w}_1 \in \mathbb{R}^d$, then

$$\frac{\partial s_p(\boldsymbol{w})}{\partial\boldsymbol{w}_r} = \frac{1}{\sqrt{n_1}}\frac{a_r}{\sqrt{m}}\frac{\partial\varphi(\boldsymbol{x}_p;\boldsymbol{w}_r)}{\partial\boldsymbol{w}_r}.$$

Similarly, we denote $\psi(\boldsymbol{y};\boldsymbol{w}) = \sigma(\boldsymbol{w}^T\boldsymbol{y})$, then

$$\frac{\partial h_j(\boldsymbol{w})}{\partial\boldsymbol{w}_r} = \frac{1}{\sqrt{n_2}}\frac{a_r}{\sqrt{m}}\frac{\partial\psi(\boldsymbol{y}_j,\boldsymbol{w}_r)}{\partial\boldsymbol{w}_r}.$$

With the notations, we can deduce that

$$H_{p,j}^\infty = \begin{cases} \dfrac{1}{n_1}\mathbb{E}_{\boldsymbol{w}\sim\mathcal{N}(\boldsymbol{0},\boldsymbol{I})}\left\langle\dfrac{\partial\varphi(\boldsymbol{x}_p;\boldsymbol{w})}{\partial\boldsymbol{w}}, \dfrac{\partial\varphi(\boldsymbol{x}_j;\boldsymbol{w})}{\partial\boldsymbol{w}}\right\rangle, & 1 \leq p \leq n_1, 1 \leq j \leq n_1, \\[3mm] \dfrac{1}{\sqrt{n_1 n_2}}\mathbb{E}_{\boldsymbol{w}\sim\mathcal{N}(\boldsymbol{0},\boldsymbol{I})}\left\langle\dfrac{\partial\varphi(\boldsymbol{x}_p;\boldsymbol{w})}{\partial\boldsymbol{w}}, \dfrac{\partial\psi(\boldsymbol{y}_j;\boldsymbol{w})}{\partial\boldsymbol{w}}\right\rangle, & 1 \leq p \leq n_1, n_1 + 1 \leq j \leq n_1 + n_2, \\[3mm] \dfrac{1}{n_2}\mathbb{E}_{\boldsymbol{w}\sim\mathcal{N}(\boldsymbol{0},\boldsymbol{I})}\left\langle\dfrac{\partial\psi(\boldsymbol{y}_p;\boldsymbol{w})}{\partial\boldsymbol{w}}, \dfrac{\partial\psi(\boldsymbol{y}_j;\boldsymbol{w})}{\partial\boldsymbol{w}}\right\rangle, & n_1 + 1 \leq p \leq n_1 + n_2, n_1 + 1 \leq j \leq n_1 + n_2, \end{cases}$$

where $H_{p,j}^\infty$ is the $(p,j)$-th entry of $\boldsymbol{H}^\infty$.

The proof of this lemma requires tools from functional analysis. Let $\mathcal{H}$ be a Hilbert space of integrable $(d+2)$-dimensional vector fields on $\mathbb{R}^{d+2}$, i.e., $f \in \mathcal{H}$ if $\mathbb{E}_{\boldsymbol{w}\sim\mathcal{N}(\boldsymbol{0},\boldsymbol{I})}[\|f(\boldsymbol{w})\|_2^2] < \infty$. The inner product for any two elements $f, g$ in $\mathcal{H}$ is $\mathbb{E}_{\boldsymbol{w}\sim\mathcal{N}(\boldsymbol{0},\boldsymbol{I})}[\langle f(\boldsymbol{w}), g(\boldsymbol{w})\rangle]$. Thus, proving $\boldsymbol{H}^\infty$ is strictly positive definite is equivalent to show that

$$\frac{\partial\varphi(\boldsymbol{x}_1;\boldsymbol{w})}{\partial\boldsymbol{w}}, \cdots, \frac{\partial\varphi(\boldsymbol{x}_{n_1};\boldsymbol{w})}{\partial\boldsymbol{w}}, \frac{\partial\psi(\boldsymbol{y}_1;\boldsymbol{w})}{\partial\boldsymbol{w}}, \cdots, \frac{\partial\psi(\boldsymbol{y}_{n_2};\boldsymbol{w})}{\partial\boldsymbol{w}} \in \mathcal{H}$$

are linearly independent. Suppose that there are $\alpha_1, \cdots, \alpha_{n_1}, \beta_1, \cdots, \beta_{n_2} \in \mathbb{R}$ such that

$$\alpha_1 \frac{\partial \varphi(\boldsymbol{x}_1; \boldsymbol{w})}{\partial \boldsymbol{w}} + \cdots + \alpha_{n_1} \frac{\partial \varphi(\boldsymbol{x}_{n_1}; \boldsymbol{w})}{\partial \boldsymbol{w}} + \beta_1 \frac{\partial \psi(\boldsymbol{y}_1; \boldsymbol{w})}{\partial \boldsymbol{w}} + \cdots + \beta_{n_2} \frac{\partial \psi(\boldsymbol{y}_{n_2}; \boldsymbol{w})}{\partial \boldsymbol{w}} = 0 \ in \ \mathcal{H}.$$

This implies that

$$\alpha_1 \frac{\partial \varphi(\boldsymbol{x}_1; \boldsymbol{w})}{\partial \boldsymbol{w}} + \cdots + \alpha_{n_1} \frac{\partial \varphi(\boldsymbol{x}_{n_1}; \boldsymbol{w})}{\partial \boldsymbol{w}} + \beta_1 \frac{\partial \psi(\boldsymbol{y}_1; \boldsymbol{w})}{\partial \boldsymbol{w}} + \cdots + \beta_{n_2} \frac{\partial \psi(\boldsymbol{y}_{n_2}; \boldsymbol{w})}{\partial \boldsymbol{w}} = 0 \tag{66}$$

holds for all $\boldsymbol{w} \in \mathbb{R}^{d+1}$, as $\sigma(\cdot)$ is smooth.

We first compute the derivatives of $\varphi$ and $\psi$. Differentiating $\psi(\boldsymbol{y}; \boldsymbol{w})$ $k$ times with respect to $\boldsymbol{w}$, we have

$$\frac{\partial^k \psi(\boldsymbol{y}; \boldsymbol{w})}{\partial \boldsymbol{w}^k} = \sigma^{(k)}(\boldsymbol{w}^T \boldsymbol{y}) \boldsymbol{y}^{\otimes(k)},$$

where $\otimes$ denotes tensor product.

For $\varphi(\boldsymbol{x}; \boldsymbol{w})$, let $\varphi_0(\boldsymbol{x}; \boldsymbol{w}) = \sigma'(\boldsymbol{w}^T \boldsymbol{x}) w_0$, $\varphi_i(\boldsymbol{x}; \boldsymbol{w}) = \sigma''(\boldsymbol{w}^T \boldsymbol{x}) w_i^2$ for $1 \le i \le d$, then

$$\varphi(\boldsymbol{x}; \boldsymbol{w}) = \varphi_0(\boldsymbol{x}; \boldsymbol{w}) - \sum_{i=1}^{d} \varphi_i(\boldsymbol{x}; \boldsymbol{w}).$$

Differentiating $\varphi_0(\boldsymbol{x}; \boldsymbol{w})$ $k$ times with respect to $\boldsymbol{w}$, similar to the Leibniz rule for the $k$-th derivative of the product of two scalar functions, we have

$$\frac{\partial^k \varphi_0(\boldsymbol{x}; \boldsymbol{w})}{\partial \boldsymbol{w}^k} = \sigma^{(k+1)}(\boldsymbol{w}^T \boldsymbol{x}) w_0 \boldsymbol{x}^{\otimes(k)} + k \sigma^{(k)}(\boldsymbol{w}^T \boldsymbol{x}) \boldsymbol{e}_0 \otimes \boldsymbol{x}^{\otimes(k-1)}, \tag{67}$$

where $\boldsymbol{e}_0 = (1, 0, \cdots, 0)^T \in \mathbb{R}^{d+2}$.

Similarly, for $\varphi_i(\boldsymbol{x}; \boldsymbol{w})$, $1 \le i \le d$, we have

$$\frac{\partial^k \varphi_i(\boldsymbol{x}; \boldsymbol{w})}{\partial \boldsymbol{w}^k} = \sigma^{(k+2)}(\boldsymbol{w}^T \boldsymbol{x}) w_i^2 \boldsymbol{x}^{\otimes(k)} + C_k^1 \sigma^{(k+1)}(\boldsymbol{w}^T \boldsymbol{x}) 2 w_i \boldsymbol{e}_i \otimes \boldsymbol{x}^{\otimes(k-1)} + C_k^2 \sigma^{(k)}(\boldsymbol{w}^T \boldsymbol{x}) 2 \boldsymbol{e}_i^{\otimes(2)} \otimes \boldsymbol{x}^{\otimes(k-2)}, \tag{68}$$

where $\boldsymbol{e}_i \in \mathbb{R}^{d+2}$, is a vector where all other components are 0, and only the $(i+1)$-th component is 1.

Combining the results in (67) and (68) for the derivatives of $\varphi_0(\boldsymbol{x}; \boldsymbol{w}), \cdots, \varphi_d(\boldsymbol{x}; \boldsymbol{w})$ yields that

$$\frac{\partial^k \varphi(\boldsymbol{x}; \boldsymbol{w})}{\partial \boldsymbol{w}^k} = \frac{\partial^k \varphi_0(x; \boldsymbol{w})}{\partial \boldsymbol{w}^k} - \sum_{i=1}^{d} \frac{\partial^k \varphi_i(x; \boldsymbol{w})}{\partial \boldsymbol{w}^k}$$

$$= \sigma^{(k+1)}(\boldsymbol{w}^T \boldsymbol{x}) w_0 \boldsymbol{x}^{\otimes(k)} + k \sigma^{(k)}(\boldsymbol{w}^T \boldsymbol{x}) \boldsymbol{e}_0 \otimes \boldsymbol{x}^{\otimes(k-1)}$$

$$- \sum_{i=1}^{d} \left( \sigma^{(k+2)}(\boldsymbol{w}^T \boldsymbol{x}) w_i^2 \boldsymbol{x}^{\otimes(k)} + C_k^1 \sigma^{(k+1)}(\boldsymbol{w}^T \boldsymbol{x}) 2 w_i \boldsymbol{e}_i \otimes \boldsymbol{x}^{\otimes(k-1)} + C_k^2 \sigma^{(k)}(\boldsymbol{w}^T \boldsymbol{x}) 2 \boldsymbol{e}_i^{\otimes(2)} \otimes \boldsymbol{x}^{\otimes(k-2)} \right). \tag{69}$$

Note that when no two points in $\{\boldsymbol{x}_1, \cdots, \boldsymbol{x}_{n_1}, \boldsymbol{y}_1, \cdots, \boldsymbol{y}_{n_2}\}$ are parallel,

$$\boldsymbol{x}_1^{\otimes(n_1+n_2-1)}, \cdots, \boldsymbol{x}_{n_1}^{\otimes(n_1+n_2-1)}, \boldsymbol{y}_1^{\otimes(n_1+n_2-1)}, \cdots, \boldsymbol{y}_{n_2}^{\otimes(n_1+n_2-1)}$$

are independent (see Lemma G.6 in (Du et al., 2018)). It motivates us to differentiate both sides in (66) $(k-1)$ times for $\boldsymbol{w}$ with $k = n_1 + n_2 + 1$, then we have

$$\alpha_1 \frac{\partial^k \varphi(\boldsymbol{x}_1; \boldsymbol{w})}{\partial \boldsymbol{w}^k} + \cdots + \alpha_{n_1} \frac{\partial^k \varphi(\boldsymbol{x}_{n_1}; \boldsymbol{w})}{\partial \boldsymbol{w}^k} + \beta_1 \frac{\partial^k \psi(\boldsymbol{y}_1; \boldsymbol{w})}{\partial \boldsymbol{w}^k} + \cdots + \beta_{n_2} \frac{\partial^k \psi(\boldsymbol{y}_{n_2}; \boldsymbol{w})}{\partial \boldsymbol{w}^k} = 0. \tag{70}$$

Since

$$\frac{\partial^{(n_1+n_2+1)} \psi(\boldsymbol{y}_j; \boldsymbol{w})}{\partial \boldsymbol{w}^{(n_1+n_2+1)}} = \sigma^{(n_1+n_2+1)}(\boldsymbol{w}^T \boldsymbol{y}_j) \boldsymbol{y}_j^{\otimes(n_1+n_2+1)},$$

we can deduce from the independence of the tensors that for any $j \in [n_2]$ and $\boldsymbol{w} \in \mathbb{R}^{d+2}$,

$$\beta_j \sigma^{(n_1+n_2+1)}(\boldsymbol{w}^T \boldsymbol{y}_j) \boldsymbol{y}_j^{\otimes(2)} = 0.$$

Now, we can choose a $\boldsymbol{w}$ such that $\sigma^{(n_1+n_2+1)}(\boldsymbol{w}^T \boldsymbol{y}_j) \neq 0$, thus

$$\beta_j \sigma^{(n_1+n_2+1)}(\boldsymbol{w}^T \boldsymbol{y}_j) \|\boldsymbol{y}_j\|_2^2 = Trace\left(\beta_j \sigma^{(n_1+n_2+1)}(\boldsymbol{w}^T \boldsymbol{y}_j) \boldsymbol{y}_j^{\otimes(2)}\right) = 0,$$

which implies $\beta_j = 0$ and then this holds for all $j \in [n_2]$.

Similarly, for $\alpha_i$, $i \in [n_1]$, from (67) and (68), we have

$$\alpha_i [\sigma^{(k+1)}(\boldsymbol{w}^T \boldsymbol{x}_i) w_0 \boldsymbol{x}_i^{\otimes(2)} + k\sigma^{(k)}(\boldsymbol{w}^T \boldsymbol{x}_i) \boldsymbol{e}_0 \otimes \boldsymbol{x}_i$$
$$- \sum_{j=1}^{d} \left( \sigma^{(k+2)}(\boldsymbol{w}^T \boldsymbol{x}_i) w_j^2 \boldsymbol{x}_i^{\otimes(2)} + C_k^1 \sigma^{(k+1)}(\boldsymbol{w}^T \boldsymbol{x}_i) 2 w_j \boldsymbol{e}_j \otimes \boldsymbol{x}_i + C_k^2 \sigma^{(k)}(\boldsymbol{w}^T \boldsymbol{x}_i) 2 \boldsymbol{e}_j^{\otimes(2)}\right)] = 0.$$

For fixed $i$, denote $\boldsymbol{x}_i = (x_{i0}, \boldsymbol{x}_{i1}^T)^T$ with $x_{i0} \in \mathbb{R}$ and $\boldsymbol{x}_{i1} \in \mathbb{R}^{d+1}$. We consider two cases: (1) $\boldsymbol{x}_{i1} \neq 0$; (2) $\boldsymbol{x}_{i1} = 0$. Although $\boldsymbol{x}_i$ has been augmented so that $\boldsymbol{x}_{i1} \neq 0$, we still consider Case 2 to account for scenarios where we might need to use neural networks without a bias term.

In the case(1), we can let $w_0 = 0$, thus for $k = n_1 + n_2 + 1$,

$$\alpha_i \left[ k\sigma^{(k)}(\boldsymbol{w}^T \boldsymbol{x}_i) \boldsymbol{e}_0 \otimes \boldsymbol{x}_i - \sum_{j=1}^{d} \left( \sigma^{(k+2)}(\boldsymbol{w}^T \boldsymbol{x}_i) w_j^2 \boldsymbol{x}_i^{\otimes(2)} + C_k^1 \sigma^{(k+1)}(\boldsymbol{w}^T \boldsymbol{x}_i) 2 w_j \boldsymbol{e}_j \otimes \boldsymbol{x}_i + C_k^2 \sigma^{(k)}(\boldsymbol{w}^T \boldsymbol{x}_i) 2 \boldsymbol{e}_j^{\otimes(2)}\right) \right] = 0,$$

which implies that the trace is 0, i.e.,

$$\alpha_i \left[ k\sigma^{(k)}(\boldsymbol{w}^T \boldsymbol{x}_i) x_{i,0} - \sum_{j=1}^{d} \left( \sigma^{(k+2)}(\boldsymbol{w}^T \boldsymbol{x}_i) w_j^2 \|\boldsymbol{x}_i\|_2^2 + C_k^1 \sigma^{(k+1)}(\boldsymbol{w}^T \boldsymbol{x}_i) 2 w_j \boldsymbol{x}_{ij} + C_k^2 \sigma^{(k)}(\boldsymbol{w}^T \boldsymbol{x}_i) 2 \right) \right] = 0.$$

Rearranging it yields that

$$\alpha_i \left[ k\sigma^{(k)}(w^T \boldsymbol{x}_i) \boldsymbol{x}_{i0} - \left( \sigma^{(k+2)}(\boldsymbol{w}^T \boldsymbol{x}_i) \|\boldsymbol{w}_1\|_2^2 \|\boldsymbol{x}_i\|_2^2 + 2 C_k^1 \sigma^{(k+1)}(\boldsymbol{w}^T \boldsymbol{x}_i) \boldsymbol{w}_1^T \boldsymbol{x}_{i1} + 2 C_k^2 \sigma^{(k)}(\boldsymbol{w}^T \boldsymbol{x}_i) \right) \right] = 0.$$

Now, we can set $\boldsymbol{w}_1^T \boldsymbol{x}_{i1} = c$ such that $\sigma^{(k+1)}(c) \neq 0$. Since $\boldsymbol{w}^T \boldsymbol{x}_i = \boldsymbol{w}_1^T \boldsymbol{x}_{i1} = c$, we have

$$\alpha_i \left[ k\sigma^{(k)}(c) \boldsymbol{x}_{i,0} - \left( \sigma^{(k+2)}(c) \|\boldsymbol{w}_1\|_2^2 \|\boldsymbol{x}_i\|_2^2 + 2 C_k^1 \sigma^{(k+1)}(c) c + 2 C_k^2 \sigma^{(k)}(c) \right) \right] = 0.$$

Note that the only variable in above equation is $\boldsymbol{w}_1$ and in the hyperplane $\{\boldsymbol{w}_1 : \boldsymbol{w}_1^T \boldsymbol{x}_{i,1} = c\}$, $\|\boldsymbol{w}_1\|_2$ can be selected to tend infinite, thus $\alpha_i = 0$.

In the case(2), we have that

$$\alpha_i \left[ \sigma^{(k+1)}(\boldsymbol{w}^T \boldsymbol{x}_i) w_0 \boldsymbol{x}_i^{\otimes(2)} + k\sigma^{(k)}(\boldsymbol{w}^T \boldsymbol{x}_i) \boldsymbol{e}_0 \otimes \boldsymbol{x}_i - \sum_{j=1}^{d} \left( \sigma^{(k+2)}(\boldsymbol{w}^T \boldsymbol{x}_i) w_j^2 \boldsymbol{x}_i^{\otimes(2)} + C_k^2 \sigma^{(k)}(\boldsymbol{w}^T \boldsymbol{x}_i) 2 \boldsymbol{e}_j^{\otimes(2)}\right) \right] = 0.$$

From the observation of the $(t, t)$-th entry of the matrix above with $t \geq 2$, we have that

$$\alpha_i(-C_k^2 \sigma^{(k)}(\boldsymbol{w}^T \boldsymbol{x}_i) 2) = -2\alpha_i C_k^2 \sigma^{(k)}(w_0 x_{i0}) = 0.$$

Then, taking a $w_0$ such that $\sigma^{(k)}(w_0 x_{i0}) \neq 0$ yields the conclusion. □

**B.2. Proof of Lemma 4.5**

*Proof.* Recall that

$$\frac{\partial s_p(\boldsymbol{w})}{\partial \boldsymbol{w}_r} = \frac{a_r}{\sqrt{n_1 m}} \left[ \sigma''(\boldsymbol{w}_r^T \boldsymbol{x}_p) w_{r0} \boldsymbol{x}_p + \sigma'(\boldsymbol{w}_r^T \boldsymbol{x}_p) \begin{pmatrix} 1 \\ \boldsymbol{0}_{d+1} \end{pmatrix} - \sigma'''(\boldsymbol{w}_r^T \boldsymbol{x}_p) \|\boldsymbol{w}_{r1}\|_2^2 \boldsymbol{x}_p \right.$$

$$\left. -2\sigma''(\boldsymbol{w}_r^T \boldsymbol{x}_p) \begin{pmatrix} 0 \\ \boldsymbol{w}_{r1} \end{pmatrix} \right]$$

and

$$\frac{\partial h_j(\boldsymbol{w})}{\partial \boldsymbol{w}_r} = \frac{a_r}{\sqrt{n_2 m}} \sigma'(\boldsymbol{w}_r^T \boldsymbol{y}_j) \boldsymbol{y}_j.$$

(1) When $\sigma(\cdot)$ is the ReLU$^3$ activation function.

From the form of $\frac{\partial s_p(\boldsymbol{w})}{\partial \boldsymbol{w}_r}$, we can deduce that

$$\left\| \frac{\partial s_p(\boldsymbol{w})}{\partial \boldsymbol{w}_r} - \frac{\partial s_p(0)}{\partial \boldsymbol{w}_r} \right\|_2$$

$$\lesssim \frac{1}{\sqrt{n_1 m}} \left[ R(\|\boldsymbol{w}_r(0)\|_2 + 1) + |I\{\boldsymbol{w}_r^T \boldsymbol{x}_p \geq 0\} - I\{\boldsymbol{w}_r(0)^T \boldsymbol{x}_p \geq 0\}|(\|\boldsymbol{w}_r(0)\|_2^2 + 1)\right] \tag{71}$$

$$\leq \frac{1}{\sqrt{n_1 m}} \left[ R(\|\boldsymbol{w}_r(0)\|_2 + 1) + I\{A_{pr}\}(\|\boldsymbol{w}_r(0)\|_2^2 + 1)\right],$$

where the second inequality follows from the fact $\|\boldsymbol{w} - \boldsymbol{w}_r(0)\|_2 < R \leq 1$ and the definition of $A_{pr}$ in (36).

Similarly, we have that

$$\left\| \frac{\partial h_j(\boldsymbol{w})}{\partial \boldsymbol{w}_r} - \frac{\partial h_j(0)}{\partial \boldsymbol{w}_r} \right\|_2 \lesssim \frac{1}{\sqrt{n_2 m}} R(\|\boldsymbol{w}_r(0)\|_2 + 1). \tag{72}$$

Combining (71) and (72), we can deduce that

$$\|\boldsymbol{J}(\boldsymbol{w}) - \boldsymbol{J}(0)\|_2^2$$

$$\leq \|\boldsymbol{J}(\boldsymbol{w}) - \boldsymbol{J}(0)\|_F^2$$

$$= \sum_{i=1}^{n_1+n_2} \|\boldsymbol{J}_i(\boldsymbol{w}) - \boldsymbol{J}_i(0)\|_2^2$$

$$= \sum_{r=1}^{m} \left( \sum_{p=1}^{n_1} \left\| \frac{\partial s_p(\boldsymbol{w})}{\partial \boldsymbol{w}_r} - \frac{\partial s_p(0)}{\partial \boldsymbol{w}_r} \right\|_2^2 + \sum_{j=1}^{n_2} \left\| \frac{\partial h_j(\boldsymbol{w})}{\partial \boldsymbol{w}_r} - \frac{\partial h_j(0)}{\partial \boldsymbol{w}_r} \right\|_2^2 \right)$$

$$\lesssim \sum_{r=1}^{m} \left( \sum_{p=1}^{n_1} \frac{1}{n_1 m} \left( R(\|\boldsymbol{w}_r(0)\|_2 + 1) + I\{A_{pr}\}(\|\boldsymbol{w}_r(0)\|_2^2 + 1)\right)^2 + \sum_{j=1}^{n_2} \frac{1}{n_2 m} (R\|\boldsymbol{w}_r(0)\|_2 + R)^2 \right)$$

$$\lesssim \frac{R^2}{m} \sum_{r=1}^{m} (\|\boldsymbol{w}_r(0)\|_2^2 + 1) + \frac{1}{n_1 m} \sum_{p=1}^{n_1} \sum_{r=1}^{m} I\{A_{pr}\}(\|\boldsymbol{w}_r(0)\|_2^4 + 1)$$

$$= \frac{R^2}{m} \sum_{r=1}^{m} (\|\boldsymbol{w}_r(0)\|_2^2 + 1)$$

$$+ \frac{1}{n_1 m} \sum_{p=1}^{n_1} \sum_{r=1}^{m} I\{A_{pr}\} \left( \|\boldsymbol{w}_r(0)\|_2^4 I\{\|\boldsymbol{w}_r(0)\|_2^2 \leq M\} + \|\boldsymbol{w}_r(0)\|_2^4 I\{\|\boldsymbol{w}_r(0)\|_2^2 > M\} + 1\right)$$

$$\lesssim \frac{R^2}{m} \sum_{r=1}^{m} (\|\boldsymbol{w}_r(0)\|_2^2 + 1) + \frac{M^2}{n_1 m} \sum_{p=1}^{n_1} \sum_{r=1}^{m} I\{A_{pr}\} + \frac{1}{m} \sum_{r=1}^{m} \|\boldsymbol{w}_r(0)\|_2^4 I\{\|\boldsymbol{w}_r(0)\|_2^2 > M\},$$

where $M = 2(d+2)\log(2m(d+2)/\delta)$. Note that from (47), we have

$$P\left(\exists r \in [m], \|\boldsymbol{w}_r(0)\|_2^2 \geq 2(d+2)\log\left(\frac{2m(d+2)}{\delta}\right)\right) \leq \delta.$$

On the other hand, applying Bernstein's inequality yields that with probability at least $1 - n_1 e^{-mR}$,

$$\frac{1}{m}\sum_{r=1}^{m} I\{A_{pr}\} < 4R$$

holds for all $p \in [n_1]$.

Therefore, we have that

$$\|\boldsymbol{J}(\boldsymbol{w}) - \boldsymbol{J}(0)\|_2^2 \lesssim MR^2 + R^2 + M^2R \lesssim M^2R$$

holds with probability at least $1 - \delta - n_1 e^{-mR}$.

(2) Note that when $\sigma$ satisfies Assumption 4.2, $\sigma', \sigma''$ and $\sigma'''$ are all Lipschitz continuous and bounded. Thus, we can obtain that

$$\left\|\frac{\partial s_p(\boldsymbol{w})}{\partial \boldsymbol{w}_r} - \frac{\partial s_p(0)}{\partial \boldsymbol{w}_r}\right\|_2 \lesssim \frac{1}{\sqrt{n_1 m}} R(\|\boldsymbol{w}_r(0)\|_2^2 + \|\boldsymbol{w}_r(0)\|_2 + 1) \lesssim \frac{1}{\sqrt{n_1 m}} R(\|\boldsymbol{w}_r(0)\|_2^2 + 1), \tag{73}$$

where the second inequality is from Young's inequality.

Similarly, we have

$$\left\|\frac{\partial h_j(\boldsymbol{w})}{\partial \boldsymbol{w}_r} - \frac{\partial h_j(0)}{\partial \boldsymbol{w}_r}\right\|_2 \lesssim \frac{1}{\sqrt{n_2 m}} R(\|\boldsymbol{w}_r(0)\|_2 + 1). \tag{74}$$

Combining (73) and (74) yields that

$$\|\boldsymbol{J}(\boldsymbol{w}) - \boldsymbol{J}(0)\|_2^2$$

$$\leq \sum_{r=1}^{m}\left(\sum_{p=1}^{n_1}\left\|\frac{\partial s_p(\boldsymbol{w})}{\partial \boldsymbol{w}_r} - \frac{\partial s_p(0)}{\partial \boldsymbol{w}_r}\right\|_2^2 + \sum_{j=1}^{n_2}\left\|\frac{\partial h_j(\boldsymbol{w})}{\partial \boldsymbol{w}_r} - \frac{\partial h_j(0)}{\partial \boldsymbol{w}_r}\right\|_2^2\right)$$

$$\lesssim \sum_{r=1}^{m}\left(\sum_{p=1}^{n_1}\frac{1}{n_1 m}(R\|\boldsymbol{w}_r(0)\|_2^2 + R)^2 + \sum_{j=1}^{n_2}\frac{1}{n_2 m}(R\|\boldsymbol{w}_r(0)\|_2 + R)^2\right)$$

$$\lesssim \frac{R^2}{m}\sum_{r=1}^{m}(\|\boldsymbol{w}_r(0)\|_2^4 + 1)$$

$$\lesssim R^2\left[d^2 + \frac{d^2}{\sqrt{m}}\sqrt{\log\left(\frac{1}{\delta}\right)} + \frac{d^2}{m}\left(\log\left(\frac{1}{\delta}\right)\right)^2\right],$$

where the last inequality follows from the fact that $\left\|\|\boldsymbol{w}_r(0)\|_2^4\right\|_{\psi_{\frac{1}{2}}} \lesssim d^2$ and Lemma C.4. $\qquad\square$

## B.3. Proof of Theorem 4.7

For the sake of completeness in the proof, we restate Condition 2 and Corollary 4.11 from the main text, and label them as Condition 4 and Corollary B.1, respectively.

*Condition* 4. At the $t$-th iteration, we have $\|\boldsymbol{w}_r(t)\|_2 \leq B$ and

$$\|\boldsymbol{w}_r(t) - \boldsymbol{w}_r(0)\|_2 \leq \frac{CB^2\sqrt{L(0)}}{\sqrt{m}\lambda_0} := R'$$

for all $r \in [m]$, where $C$ is a universal constant and $B = \sqrt{2(d+2)\log\left(\frac{2m(d+2)}{\delta}\right)} + 1$.

**Corollary B.1.** *If Condition 3 holds for $t = 0, \cdots, k$ and $R^{'} \leq R$ and $R^{''} \lesssim \sqrt{1-\eta}\sqrt{\lambda_0}$, then*

$$L(t) \leq (1-\eta)^t L(0),$$

*holds for $t = 0, \cdots, k$, where $R$ is the constant in Lemma 4.5 and $R^{''} = CM\sqrt{R}$ in (28) when $\sigma$ is the ReLU$^3$ activation function, $R^{''} = CdR$ in (30) when $\sigma$ satisfies Assumption 4.2.*

Thanks to Corollary B.1, it is sufficient to prove that Condition 4 also holds for $t = k + 1$. For readability, we defer the proof of Corollary B.1 to the end of this section. In the following, we are going to show that the Condition 4 also holds for $t = k + 1$, thus combining Condition 4 and Corollary B.1 leads to Theorem 4.7.

*Proof of Theorem 4.7.* Recall that we let $R^{''} = CM\sqrt{R}$ in (28) when $\sigma$ is the ReLU$^3$ activation function and let $R^{''} = CdR$ in (30) when $\sigma$ satisfies Assumption 4.2.

First, we can set $R^{'} \leq R$ and $R^{''} \leq \frac{\sqrt{3\lambda_0}}{6}$, since $R^{''} \lesssim \sqrt{1-\eta}\sqrt{\lambda_0}$. Then from Lemma 4.5 we have $\|J(t) - J(0)\|_2 \leq \frac{\sqrt{3\lambda_0}}{6}$, thus

$$\sigma_{min}(J(t)) \geq \sigma_{min}(J(0)) - \|J(t) - J(0)\|_2 \geq \frac{\sqrt{3\lambda_0}}{2} - \frac{\sqrt{3\lambda_0}}{6} = \frac{\sqrt{3\lambda_0}}{3}$$

and then $\lambda_{min}(H(t)) \geq \frac{\lambda_0}{3}$ for $t = 0, \cdots, k$, where $\sigma_{min}(\cdot)$ denotes the least singular value.

From the updating rule of NGD, we have

$$w_r(t+1) = w_r(t) - \eta \left[ J(t)^T \right]_r (H(t))^{-1} \begin{pmatrix} s(t) \\ h(t) \end{pmatrix},$$

where

$$\left[ J(t)^T \right]_r = \left[ \frac{\partial s_1(t)}{\partial w_r}, \cdots, \frac{\partial s_{n_1}(t)}{\partial w_r}, \frac{\partial h_1(t)}{\partial w_r}, \cdots, \frac{\partial h_{n_2}(t)}{\partial w_r} \right].$$

Therefore, for $t = 0, \cdots, k$ and any $r \in [m]$, we have

$$
\begin{aligned}
&\|w_r(t+1) - w_r(t)\|_2 \\
&\leq \eta \| \left[ J(t)^T \right]_r \|_2 \|H(t)^{-1}\|_2 \sqrt{L(t)} \\
&\leq \frac{3\eta}{\lambda_0} \| \left[ J(t)^T \right]_r \|_2 \sqrt{L(t)} \\
&\leq \frac{3\eta}{\lambda_0} \| \left[ J(t)^T \right]_r \|_F \sqrt{L(t)} \\
&= \frac{3\eta}{\lambda_0} \sqrt{ \sum_{p=1}^{n_1} \left\| \frac{\partial s_p(t)}{\partial w_r} \right\|_2^2 + \sum_{j=1}^{n_2} \left\| \frac{\partial h_j(t)}{\partial w_r} \right\|_2^2 } \sqrt{L(t)} \\
&\lesssim \frac{\eta}{\lambda_0} \sqrt{ \frac{B^4 + 1}{m} } \sqrt{L(t)} \\
&\lesssim \frac{\eta B^2}{\sqrt{m}\lambda_0} \sqrt{L(t)} \\
&\leq \frac{\eta B^2}{\sqrt{m}\lambda_0} (1-\eta)^{t/2} \sqrt{L(0)},
\end{aligned}
\tag{75}
$$

where the last inequality is due to Corollary B.1.

Summing $t$ from $0$ to $k$ yields that

$$\|\boldsymbol{w}_r(k+1) - \boldsymbol{w}_r(0)\|_2$$

$$\leq \sum_{t=0}^{k} \|\boldsymbol{w}_r(t+1) - \boldsymbol{w}_r(t)\|_2$$

$$\leq C \frac{\eta B^2}{\sqrt{m}\lambda_0} \sum_{t=0}^{k} (1-\eta)^{t/2} \sqrt{L(0)}$$

$$\leq \frac{CB^2 \sqrt{L(0)}}{\sqrt{m}\lambda_0},$$

where $C$ is a universal constant.

Now, when $R' \leq 1$, we can deduce that $\|\boldsymbol{w}_r(k+1)\|_2 \leq B$, implying that Condition 4 also holds for $t = k+1$. Thus, it remains only to derive the requirement for $m$.

Recall that we need $m$ to satisfy that $R' = \frac{CB^2\sqrt{L(0)}}{\sqrt{m}\lambda_0} \leq R$ and $R'' \lesssim \sqrt{1-\eta}\sqrt{\lambda_0}$.

(1) When $\sigma$ is the ReLU$^3$ activation function, in Corollary B.1, $R'' = CM\sqrt{R} \lesssim \sqrt{1-\eta}\sqrt{\lambda_0}$, implying that $R \lesssim \frac{(1-\eta)\lambda_0}{M^2}$. Then $R' = \frac{CB^2\sqrt{L(0)}}{\sqrt{m}\lambda_0} \leq R$ implies that

$$m = \Omega\left(\frac{1}{(1-\eta)^2} \frac{M^4 B^4 L(0)}{\lambda_0^4}\right).$$

From Lemma C.4 for the estimation of $L(0)$, i.e.,

$$L(0) \lesssim d^2 \log\left(\frac{n_1 + n_2}{\delta}\right),$$

we can deduce that

$$m = \Omega\left(\frac{1}{(1-\eta)^2} \frac{d^8}{\lambda_0^4} \log^6\left(\frac{md}{\delta}\right) \log\left(\frac{n_1 + n_2}{\delta}\right)\right).$$

(2) When $\sigma$ satisfies Assumption 4.2, we have that

$$R \lesssim \frac{\sqrt{(1-\eta)\lambda_0}}{d}, R' = \frac{CB^2\sqrt{L(0)}}{\sqrt{m}\lambda_0} \leq R.$$

From Lemma C.4, we can deduce that

$$m = \Omega\left(\frac{1}{1-\eta} \frac{d^6}{\lambda_0^3} \log^2\left(\frac{md}{\delta}\right) \log\left(\frac{n_1 + n_2}{\delta}\right)\right).$$

$\square$

*Proof of Corollary B.1.* Similar as before, when $R' \leq R$ and $R'' \leq \frac{\sqrt{3\lambda_0}}{6}$, we have $\sigma_{min}(\boldsymbol{J}(t)) \geq \frac{\sqrt{3\lambda_0}}{3}$ and then $\lambda_{min}(\boldsymbol{H}(t)) \geq \frac{\lambda_0}{3}$ for $t = 0, \cdots, k$.

Let $\boldsymbol{u}(t) = \begin{pmatrix} \boldsymbol{s}(t) \\ \boldsymbol{h}(t) \end{pmatrix}$, then

$$
\begin{aligned}
&\boldsymbol{u}(t+1) - \boldsymbol{u}(t) \\
&= \boldsymbol{u}\left(\boldsymbol{w}(t) - \eta \boldsymbol{J}(t)^T \boldsymbol{H}(t)^{-1} \boldsymbol{u}(\boldsymbol{w}(t))\right) - \boldsymbol{u}(\boldsymbol{w}(t)) \\
&= -\int_0^1 \left\langle \frac{\partial \boldsymbol{u}(\boldsymbol{w}(s))}{\partial \boldsymbol{w}}, \eta \boldsymbol{J}(t)^T \boldsymbol{H}(t)^{-1} \boldsymbol{u}(\boldsymbol{w}(t)) \right\rangle ds \\
&= -\int_0^1 \left\langle \frac{\partial \boldsymbol{u}(\boldsymbol{w}(t))}{\partial \boldsymbol{w}}, \eta \boldsymbol{J}(t)^T \boldsymbol{H}(t)^{-1} \boldsymbol{u}(\boldsymbol{w}(t)) \right\rangle ds \\
&\quad + \int_0^1 \left\langle \frac{\partial \boldsymbol{u}(\boldsymbol{w}(t))}{\partial \boldsymbol{w}} - \frac{\partial \boldsymbol{u}(\boldsymbol{w}(s))}{\partial \boldsymbol{w}}, \eta \boldsymbol{J}(t)^T \boldsymbol{H}(t)^{-1} \boldsymbol{u}(\boldsymbol{w}(t)) \right\rangle ds \\
&:= \boldsymbol{I}_1(t) + \boldsymbol{I}_2(t),
\end{aligned}
\tag{76}
$$

where the second equality is from the fundamental theorem of calculus and $\boldsymbol{w}(s) = s\boldsymbol{w}(t+1) + (1-s)\boldsymbol{w}(t) = \boldsymbol{w}(t) - s\eta \boldsymbol{J}(t)^T \boldsymbol{H}(t)^{-1} \boldsymbol{u}(t)$.

Note that $\frac{\partial \boldsymbol{u}(\boldsymbol{w}(t))}{\partial \boldsymbol{w}} = \boldsymbol{J}(t)$, thus $\boldsymbol{I}_1(t) = \eta \boldsymbol{u}(t)$. Plugging this into (76) yields that

$$
\boldsymbol{u}(t+1) = (1-\eta)\boldsymbol{u}(t) + \boldsymbol{I}_2(t).
\tag{77}
$$

Therefore, it remains only to bound $\|\boldsymbol{I}_2(t)\|_2$.

$$
\begin{aligned}
\|\boldsymbol{I}_2(t)\|_2 &= \left\| \int_0^1 \left\langle \frac{\partial \boldsymbol{u}(\boldsymbol{w}(t))}{\partial \boldsymbol{w}} - \frac{\partial \boldsymbol{u}(\boldsymbol{w}(s))}{\partial \boldsymbol{w}}, \eta \boldsymbol{J}(t)^T \boldsymbol{H}(t)^{-1} \boldsymbol{u}(\boldsymbol{w}(t)) \right\rangle ds \right\|_2 \\
&\leq \int_0^1 \|\boldsymbol{J}(\boldsymbol{w}(t)) - \boldsymbol{J}(\boldsymbol{w}(s))\|_2 \|\eta \boldsymbol{J}(t)^T \boldsymbol{H}(t)^{-1} \boldsymbol{u}(\boldsymbol{w}(t))\|_2 ds \\
&\leq \eta \|\boldsymbol{J}(t)^T \boldsymbol{H}(t)^{-1}\|_2 \|\boldsymbol{u}(\boldsymbol{w}(t))\|_2 \int_0^1 \|\boldsymbol{J}(\boldsymbol{w}(t)) - \boldsymbol{J}(\boldsymbol{w}(s))\|_2 ds \\
&\lesssim \frac{\eta \sqrt{L(t)}}{\sqrt{\lambda_0}} \int_0^1 \|\boldsymbol{J}(\boldsymbol{w}(t)) - \boldsymbol{J}(\boldsymbol{w}(s))\|_2 ds \\
&\lesssim \frac{\eta \sqrt{L(t)}}{\sqrt{\lambda_0}} \int_0^1 (\|\boldsymbol{J}(\boldsymbol{w}(t)) - \boldsymbol{J}(0)\|_2 + \|\boldsymbol{J}(\boldsymbol{w}(s)) - \boldsymbol{J}(0)\|_2) ds \\
&\lesssim \frac{\eta \sqrt{L(t)}}{\sqrt{\lambda_0}} R'',
\end{aligned}
\tag{78}
$$

where the last inequality follows from the fact that

$$
\|\boldsymbol{w}_r(s) - \boldsymbol{w}_r(0)\|_2 \leq s\|\boldsymbol{w}_r(t+1) - \boldsymbol{w}_r(0)\|_2 + (1-s)\|\boldsymbol{w}_r(t) - \boldsymbol{w}_r(0)\|_2 \leq R' \leq R
$$

and Lemma 4.5.

Plugging (78) into the recursion formula (77) yields that

$$
\begin{aligned}
\|\boldsymbol{u}(t+1)\|_2^2 &= \|(1-\eta)\boldsymbol{u}(t) + \boldsymbol{I}_2(t)\|_2^2 \\
&= (1-\eta)^2 \|\boldsymbol{u}(t)\|_2^2 + \|\boldsymbol{I}_2(t)\|_2^2 + 2\langle (1-\eta)\boldsymbol{u}(t), \boldsymbol{I}_2(t) \rangle \\
&\leq (1-\eta)^2 \|\boldsymbol{u}(t)\|_2^2 + \|\boldsymbol{I}_2(t)\|_2^2 + 2(1-\eta)\|\boldsymbol{u}(t)\|_2 \|\boldsymbol{I}_2(t)\|_2 \\
&\leq \left[ (1-\eta)^2 + \frac{C^2 \eta^2 (R'')^2}{\lambda_0} + 2(1-\eta)\frac{C\eta R''}{\sqrt{\lambda_0}} \right] \|\boldsymbol{u}(t)\|_2^2,
\end{aligned}
$$

where $C$ is a universal constant.

Then we can choose $R^{''}$ such that

$$\|\boldsymbol{I}_2(t)\|_2 \leq \frac{C\eta\sqrt{L(t)}R^{''}}{\sqrt{\lambda_0}} \leq C_1\eta\sqrt{L(t)} = C_1\eta\sqrt{\boldsymbol{u}(t)},$$

where $C$ is a universal constant and $C_1$ is a constant to be determined.

Thus, we can deduce that

$$\begin{aligned}
\|\boldsymbol{u}(t+1)\|_2^2 &\leq \left[(1-\eta)^2 + (C_1\eta)^2 + 2(1-\eta)C_1\eta\right]\|\boldsymbol{u}(t)\|_2^2 \\
&= \left[(1-\eta) + \eta(\eta C_1^2 + 2(1-\eta)C_1 + \eta - 1)\right]\|\boldsymbol{u}(t)\|_2^2 \\
&\leq (1-\eta)\|\boldsymbol{u}(t)\|_2^2,
\end{aligned}$$

where in the last inequality is due to that we can choose $C_1$ such that $\eta C_1^2 + 2(1-\eta)C_1 + \eta - 1 \leq 0$.

Note that since $\eta \in (0,1)$, the quadratic equation $\eta x^2 + 2(1-\eta)x + \eta - 1 = 0$ has one negative root and one positive root, denoted as $x_0$ and $x_1$ respectively. Therefore, the condition $C_1 \leq x_1$ is sufficient to satisfy the requirement. The explicit form of $x_1$ can be written as:

$$x_1 = \frac{2(\eta-1) + \sqrt{4(1-\eta)^2 - 4\eta(\eta-1)}}{2\eta} = \frac{\sqrt{1-\eta}}{1+\sqrt{1-\eta}} \geq \frac{\sqrt{1-\eta}}{2}.$$

Thus, $C_1 = \frac{\sqrt{1-\eta}}{2}$ is sufficient to satisfy that $\eta C_1^2 + 2(1-\eta)C_1 + \eta - 1 \leq 0$.

From this, we can deduce that

$$R^{''} \lesssim C_1\sqrt{\lambda_0} \lesssim \sqrt{1-\eta}\sqrt{\lambda_0}.$$

Therefore, we can conclude that $\|\boldsymbol{u}(t)\|_2^2 \leq (1-\eta)^t\|\boldsymbol{u}(0)\|_2^2$ holds for $t = 0, \cdots, k$.

$\square$

### B.4. Proof of Corollary 4.9

*Proof.* In the proof of Theorem 4.7, we have proved that Condition 4 holds for all $t \in \mathbb{N}$. Thus, it is sufficient to prove that Condition 4 can lead to the conclusion in Corollary 4.9.

Setting $\eta = 1$ in (77) yields that

$$\boldsymbol{u}(t+1) = \boldsymbol{I}_2(t).$$

From (78), we have that

$$\|\boldsymbol{I}_2(t)\|_2 \lesssim \frac{\sqrt{L(t)}}{\sqrt{\lambda_0}} \int_0^1 \|\boldsymbol{J}(\boldsymbol{w}(t)) - \boldsymbol{J}(\boldsymbol{w}(s))\|_2 ds. \tag{79}$$

Since $\boldsymbol{w}(s) = s\boldsymbol{w}(t+1) + (1-s)\boldsymbol{w}(t)$, then for any $r \in [m]$, we have $\|\boldsymbol{w}_r(s)\|_2 \leq s\|\boldsymbol{w}_r(t+1)\|_2 + (1-s)\|\boldsymbol{w}_r(t)\|_2 \leq B$.

When $\sigma(\cdot)$ is smooth, we can deduce that for any $r \in [m]$,

$$\left\|\frac{\partial s_p(\boldsymbol{w}(s))}{\partial \boldsymbol{w}_r} - \frac{\partial s_p(\boldsymbol{w}(t))}{\partial \boldsymbol{w}_r}\right\|_2 \lesssim \frac{1}{\sqrt{n_1 m}}(B^2+1)\|\boldsymbol{w}_r(s) - \boldsymbol{w}_r(t)\|_2 \leq \frac{1}{\sqrt{n_1 m}}(B^2+1)\|\boldsymbol{w}_r(t+1) - \boldsymbol{w}_r(t)\|_2$$

and

$$\left\|\frac{\partial h_j(\boldsymbol{w}(s))}{\partial \boldsymbol{w}_r} - \frac{\partial h_j(\boldsymbol{w}(t))}{\partial \boldsymbol{w}_r}\right\|_2 \lesssim \frac{1}{\sqrt{n_1 m}}(B+1)\|\boldsymbol{w}_r(s) - \boldsymbol{w}_r(t)\|_2 \leq \frac{1}{\sqrt{n_1 m}}(B+1)\|\boldsymbol{w}_r(t+1) - \boldsymbol{w}_r(t)\|_2.$$

From (75), we know that for any $r \in [m]$,

$$\|\boldsymbol{w}_r(t+1) - \boldsymbol{w}_r(t)\|_2 \lesssim \frac{B^2}{\sqrt{m\lambda_0}}\sqrt{L(t)}.$$

Thus for any $s \in [0,1]$, we have

$$\|\boldsymbol{J}(\boldsymbol{w}(s)) - \boldsymbol{J}(\boldsymbol{w}(t))\|_2^2$$

$$\leq \sum_{r=1}^{m} \left( \sum_{p=1}^{n_1} \left\| \frac{\partial s_p(\boldsymbol{w}(s))}{\partial \boldsymbol{w}_r} - \frac{\partial s_p(\boldsymbol{w}(t))}{\partial \boldsymbol{w}_r} \right\|_2^2 + \left\| \frac{\partial h_j(\boldsymbol{w}(s))}{\partial \boldsymbol{w}_r} - \frac{\partial h_j(\boldsymbol{w}(t))}{\partial \boldsymbol{w}_r} \right\|_2^2 \right)$$

$$\lesssim \frac{1}{m} \sum_{r=1}^{m} \left( (B^4 + 1) \|\boldsymbol{w}_r(t+1) - \boldsymbol{w}_r(t)\|_2^2 + (B^2 + 1) \|\boldsymbol{w}_r(t+1) - \boldsymbol{w}_r(t)\|_2^2 \right)$$

$$\lesssim B^4 \left( \frac{B^2}{\sqrt{m}\lambda_0} \sqrt{L(t)} \right)^2.$$

Plugging this into (79), we have

$$\|\boldsymbol{I}_2(t)\|_2 \lesssim \frac{\sqrt{L(t)}}{\sqrt{\lambda_0}} \int_0^1 \|\boldsymbol{J}(\boldsymbol{w}(t)) - \boldsymbol{J}(\boldsymbol{w}(s))\|_2 ds$$

$$\lesssim \frac{\sqrt{L(t)}}{\sqrt{\lambda_0}} \frac{B^4}{\sqrt{m}\lambda_0} \sqrt{L(t)}$$

$$= \frac{B^4}{\sqrt{m\lambda_0^3}} L(t).$$

Combining with the fact $\boldsymbol{u}(t+1) = \boldsymbol{I}_2(t)$ yields that

$$\left\| \begin{pmatrix} \boldsymbol{s}(t+1) \\ \boldsymbol{h}(t+1) \end{pmatrix} \right\|_2 \leq \frac{CB^4}{\sqrt{m\lambda_0^3}} \left\| \begin{pmatrix} \boldsymbol{s}(t) \\ \boldsymbol{h}(t) \end{pmatrix} \right\|_2^2$$

holds for $t \in \mathbb{N}$, where $C$ is a universal constant.

In the proof above, we only require that $R' \leq R$ and $R'' = CdR \leq \frac{\sqrt{3\lambda_0}}{6}$, leading to the requirement for $m$ that

$$m = \Omega \left( \frac{d^6}{\lambda_0^3} \log^2 \left( \frac{md}{\delta} \right) \log \left( \frac{n_1 + n_2}{\delta} \right) \right).$$

$\square$

## C. Auxiliary Lemmas

**Lemma C.1** (Theorem 3.1 in Kuchibhotla & Chakrabortty (2022)). *If $X_1, \cdots, X_n$ are independent mean zero random variables with $\|X_i\|_{\psi_\alpha} < \infty$ for all $1 \leq i \leq n$ and some $\alpha > 0$, then for any vector $a = (a_1, \cdots, a_n) \in \mathbb{R}^n$, the following holds true:*

$$P \left( \left| \sum_{i=1}^{n} a_i X_i \right| \geq 2eC(\alpha) \|b\|_2 \sqrt{t} + 2eL_n^*(\alpha) t^{1/\alpha} \|b\|_{\beta(\alpha)} \right) \leq 2e^{-t}, \; for \; all \; t \geq 0,$$

*where $b = (a_1 \|X_1\|_{\psi_\alpha}, \cdots, a_n \|X_n\|_{\psi_\alpha}) \in \mathbb{R}^n$,*

$$C(\alpha) := \max\{\sqrt{2}, 2^{1/\alpha}\} \begin{cases} \sqrt{8}(2\pi)^{1/4} e^{1/24} (e^{2/e}/\alpha)^{1/\alpha}, & if \; \alpha < 1, \\ 4e + 2(\log 2)^{1/\alpha}, & if \; \alpha \geq 1. \end{cases}$$

*and for $\beta(\alpha) = \infty$ when $\alpha \leq 1$ and $\beta(\alpha) = \alpha/(\alpha - 1)$ when $\alpha > 1$,*

$$L_n(\alpha) := \frac{4^{1/\alpha}}{\sqrt{2}\|b\|_2} \times \begin{cases} \|b\|_{\beta(\alpha)}, & if \; \alpha < 1, \\ 4e\|b\|_{\beta(\alpha)}/C(\alpha), & if \alpha \geq 1. \end{cases}$$

*and $L_n^*(\alpha) = L_n(\alpha) C(\alpha) \|b\|_2 / \|b\|_{\beta(\alpha)}$.*

In the following, we will provide some preliminary information about Orlicz norms.

Let $f : [0, \infty) \to [0, \infty)$ be a non-decreasing function with $f(0) = 0$. The $f$-Orlicz norm of a real-valued random variable $X$ is given by

$$\|X\|_f := \inf\{C > 0 : \mathbb{E}\left[f\left(\frac{|X|}{C}\right)\right] \leq 1\}.$$

If $\|X\|_{\psi_\alpha} < \infty$, we say that $X$ is sub-Weibull of order $\alpha > 0$, where

$$\psi_\alpha(x) := e^{x^\alpha} - 1.$$

Note that when $\alpha \geq 1$, $\|\cdot\|_{\psi_\alpha}$ is a norm and when $0 < \alpha < 1$, $\|\cdot\|_{\psi_\alpha}$ is a quasi-norm. Moreover, since $(|a| + |b|)^\alpha \leq |a|^\alpha + |b|^\alpha$ holds for any $a, b \in \mathbb{R}$ and $0 < \alpha < 1$, we can deduce that

$$\mathbb{E}e^{\frac{|X+Y|^\alpha}{|C|^\alpha}} \leq \mathbb{E}e^{\frac{|X|^\alpha+|Y|^\alpha}{|C|^\alpha}} = \mathbb{E}e^{\frac{|X|^\alpha}{|C|^\alpha}}e^{\frac{|Y|^\alpha}{|C|^\alpha}} \leq \left(\mathbb{E}e^{\frac{2|X|^\alpha}{|C|^\alpha}}\right)^{1/2}\left(\mathbb{E}e^{\frac{2|Y|^\alpha}{|C|^\alpha}}\right)^{1/2}.$$

This implies that

$$\|X + Y\|_{\psi_\alpha} \leq 2^{1/\alpha}\max\{\|X\|_{\psi_\alpha}, \|Y\|_{\psi_\alpha}\} \leq 2^{1/\alpha}(\|X\|_{\psi_\alpha} + \|Y\|_{\psi_\alpha}).$$

Furthermore, for $p, q > 0$, we have $\||X|\|_{\psi_p} = \||X|^{p/q}\|_{\psi_q}^{q/p}$. And in the related proofs, we may frequently use the fact that for real-valued random variable $X \sim \mathcal{N}(0, 1)$, we have $\|X\|_{\psi_2} \leq \sqrt{6}$ and $\|X^2\|_{\psi_1} = \|X\|_{\psi_2}^2 \leq 6$.

**Lemma C.2.** *If $\|X\|_{\psi_\alpha}, \|Y\|_{\psi_\beta} < \infty$ with $\alpha, \beta > 0$, then we have $\|XY\|_{\psi_\gamma} \leq \|X\|_{\psi_\alpha}\|Y\|_{\psi_\beta}$, where $\gamma$ satisfies that*

$$\frac{1}{\gamma} = \frac{1}{\alpha} + \frac{1}{\beta}.$$

*Proof.* Without loss of generality, we can assume that $\|X\|_{\psi_\alpha} = \|Y\|_{\psi_\beta} = 1$. To prove this, let us use Young's inequality, which states that

$$xy \leq \frac{x^p}{p} + \frac{y^q}{q}, for \ x, y \geq 0, p, q > 1.$$

Let $p = \alpha/\gamma, q = \beta/\gamma$, then

$$\mathbb{E}[\exp(|XY|^\gamma)] \leq \mathbb{E}\left[\exp\left(\frac{|X|^{\gamma p}}{p} + \frac{|Y|^{\gamma q}}{q}\right)\right]$$
$$= \mathbb{E}\left[\exp\left(\frac{|X|^\alpha}{p}\right)\exp\left(\frac{|Y|^\beta}{q}\right)\right]$$
$$\leq \mathbb{E}\left[\frac{\exp(|X|^\alpha)}{p} + \frac{\exp(|Y|^\beta)}{q}\right]$$
$$\leq \frac{2}{p} + \frac{2}{q}$$
$$= 2,$$

where the first and second inequality follow from Young's inequality. From this, we have that $\|XY\|_{\psi_\gamma} \leq \|X\|_{\psi_\alpha}\|Y\|_{\psi_\beta}$.

$\square$

**Lemma C.3** (Bernstein inequality, Theorem 3.1.7 in Giné & Nickl (2021)). *Let $X_i$, $1 \leq i \leq n$ be independent centered random variables a.s. bounded by $c < \infty$ in absolute value. Set $\sigma^2 = 1/n \sum_{i=1}^n \mathbb{E}X_i^2$ and $S_n = 1/n \sum_{i=1}^n X_i$. Then, for all $t \geq 0$,*

$$P\left(S_n \geq \sqrt{\frac{2\sigma^2 t}{n}} + \frac{ct}{3n}\right) \leq e^{-u}.$$

**Lemma C.4.** *For $0 < \delta < 1$, with probability at least $1 - \delta$, we have that when $m \geq \log^2\left(\frac{n_1 + n_2}{\delta}\right)$,*

$$L(0) = \left\| \begin{pmatrix} s(0) \\ h(0) \end{pmatrix} \right\|_2^2 = \mathcal{O}\left( d^2 \log\left(\frac{n_1 + n_2}{\delta}\right) \right).$$

*Proof.* Recall that for $p \in [n_1]$,

$$s_p(0) = \frac{1}{\sqrt{n_1}} \left[ \frac{1}{\sqrt{m}} \sum_{r=1}^m a_r \left( \sigma'(\boldsymbol{w}_r(0)^T \boldsymbol{x}_p) w_{r0}(0) - \sigma''(\boldsymbol{w}_r(0)^T \boldsymbol{x}_p) \|\boldsymbol{w}_{r1}(0)\|_2^2 \right) - f(x_p) \right]$$

and for $j \in [n_2]$,

$$h_j(0) = \frac{1}{\sqrt{n_2}} \left[ \frac{1}{\sqrt{m}} \sum_{r=1}^m a_r \sigma(\boldsymbol{w}_r(0)^T \boldsymbol{y}_j) - g(\boldsymbol{y}_j) \right].$$

Then

$$L(0) = \sum_{p=1}^{n_1} \frac{1}{2} (s_p(0))^2 + \sum_{j=1}^{n_2} \frac{1}{2} (h_j(0))^2$$

$$\leq \frac{1}{n_1} \sum_{p=1}^{n_1} \left( \frac{1}{\sqrt{m}} \sum_{r=1}^m a_r \left( \sigma'(\boldsymbol{w}_r(0)^T \boldsymbol{x}_p) w_{r0}(0) - \sigma''(\boldsymbol{w}_r(0)^T \boldsymbol{x}_p) \|\boldsymbol{w}_{r1}(0)\|_2^2 \right) \right)^2 + \frac{1}{n_1} \sum_{p=1}^{n_1} f^2(x_p)$$

$$+ \frac{1}{n_2} \sum_{j=1}^{n_2} \left( \frac{1}{\sqrt{m}} \sum_{r=1}^m a_r \sigma(\boldsymbol{w}_r(0)^T \boldsymbol{y}_j) \right)^2 + \frac{1}{n_2} \sum_{j=1}^{n_2} g^2(\boldsymbol{y}_j).$$

Note that

$$\left| a_r \left( \sigma'(\boldsymbol{w}_r(0)^T \boldsymbol{x}_p) w_{r0} - \sigma''(\boldsymbol{w}_r(0)^T \boldsymbol{x}_p) \|\boldsymbol{w}_{r1}(0)\|_2^2 \right) \right| \lesssim \|\boldsymbol{w}_r(0)\|_2^2 |\boldsymbol{w}_r(0)^T \boldsymbol{x}_p|$$

and $\left| a_r \sigma(\boldsymbol{w}_r(0)^T \boldsymbol{y}_j) \right| \lesssim \|\boldsymbol{w}_r(0)\|_2^2 |\boldsymbol{w}_r(0)^T \boldsymbol{y}_j|$.

Since $\left\| \|\boldsymbol{w}_r(0)\|_2^2 \right\|_{\psi_1} = \mathcal{O}(d)$ and $\|\boldsymbol{w}_r(0)^T \boldsymbol{y}_j\|_{\psi_2}, \|\boldsymbol{w}_r(0)^T \boldsymbol{x}_p\|_{\psi_2} = \mathcal{O}(1)$, from Lemma C.2, we have that

$$\left\| \|\boldsymbol{w}_r(0)\|_2^2 |\boldsymbol{w}_r(0)^T \boldsymbol{x}_p| \right\|_{\psi_{\frac{2}{3}}} = \mathcal{O}(d), \left| \boldsymbol{w}_r(0)^T \boldsymbol{y}_j \right| \|_{\psi_{\frac{2}{3}}} = \mathcal{O}(d).$$

Applying Lemma C.1 yields that for fixed $p \in [n_1]$ and $j \in [n_2]$ with probability at least $1 - 2e^{-t}$,

$$\left| \frac{1}{\sqrt{m}} \sum_{r=1}^m a_r \left( \sigma'(\boldsymbol{w}_r(0)^T \boldsymbol{x}_p) w_{r0}(0) - \sigma''(\boldsymbol{w}_r(0)^T \boldsymbol{x}_p) \|\boldsymbol{w}_{r1}(0)\|_2^2 \right) \right| \lesssim d\sqrt{t} + \frac{d}{\sqrt{m}} t^{\frac{3}{2}}$$

and with probability at least $1 - 2e^{-t}$,

$$\left| \frac{1}{\sqrt{m}} \sum_{r=1}^m a_r \sigma(\boldsymbol{w}_r(0)^T \boldsymbol{y}_j) \right| \lesssim d\sqrt{t} + \frac{d}{\sqrt{m}} t^{\frac{3}{2}}.$$

Then taking a union bound for all $p \in [n_1]$ and $j \in [n_2]$ with $2(n_1 + n_2)e^{-t} = \delta$ yields that

$$L(0) \lesssim \left( d\sqrt{t} + \frac{d}{\sqrt{m}} t^{\frac{3}{2}} \right)^2$$

$$\lesssim d^2 t + \frac{d^2 t^3}{m}$$

$$= d^2 \left( \log\left(\frac{n_1 + n_2}{\delta}\right) + \frac{1}{m} \log^3\left(\frac{n_1 + n_2}{\delta}\right) \right)$$

$$\lesssim d^2 \log\left(\frac{n_1 + n_2}{\delta}\right),$$

since $m \geq \log^2\left(\frac{n_1+n_2}{\delta}\right)$.

$\square$

