# OpenReview forum: "Convergence Analysis of Natural Gradient Descent for Over-parameterized Physics-Informed Neural Networks"
_ICML.cc/2025/Conference — Submitted to ICML 2025_

### Official Review · Reviewer_4G6r · 2025-03-10

**Overall Recommendation:** 4

**Summary:**

This paper considers the convergences of a certain class of PINNs with 2 layers in the overparametrization regime (NTK) and makes two contributions: 1) improves the convergence of GD (conditions for LR) and 2) shows quadratic convergence of natural gradient descent. Specifically for 1) the LR dependency on the smallest eigen value of the gram matrix (fisher information matrix) is removed and for 2) it considers a different stability criterion for the jacobian for the derivation.

**Claims And Evidence:**

The claims are justified via theoretical derivations.

**Essential References Not Discussed:**

N/A

**Experimental Designs Or Analyses:**

N/A

**Methods And Evaluation Criteria:**

There was no evaluation as this is a purely theoretical paper

**Other Comments Or Suggestions:**

N/A

**Other Strengths And Weaknesses:**

## Strengths
1. The contributions are interesting and will be useful to the theoretical community, and it may be useful in practice. The proof ideas appear novel.

## Weaknesses
1. The paper jumps to the derivations straightway, but it might be better to contextualize it first. This would make the paper more accessible to a general audience. also, please define the nations when they were introduced.

**Questions For Authors:**

N/A

**Relation To Broader Scientific Literature:**

The main focus is on the niche domain of PINNs and the paper is targeted towards such a niche audience. In fact, the paper assumes familiarity of the previous theoretical work and reuses notations (even in the abstract without defining them properly until in the method section). The paper may not be easily accessible for a general audience, even though the topic (convergence of NGD) might be interesting to them.

**Theoretical Claims:**

Theoretical claims are built on top of previous works and appear correct. I could largely follow the arguments, however, I didn't check for the correctness of derivations rigorously.

---

> ### Author Rebuttal · Authors · 2025-03-30
>
> Thank you for taking the time to review our article and for your insightful comments. We apologize for the oversight in notations. We agree that adding some context before diving into derivations would improve readability, especially for a broader audience. In the revised version, we will make sure to clarify the definitions of the notations when they are first introduced and add more introductory context.
>
> In this work, for gradient descent, we improve both the requirements for the learning rate $\eta$ and the width $m$. Furthermore, we establish convergence guarantees for natural gradient descent (NGD) in training over-parameterized two-layer PINNs, demonstrating that NGD achieves: (1) an $\mathcal{O}(1)$ learning rate, and (2) faster convergence compared to gradient descent.
>
> Although this is a purely theoretical paper, we can provide some experiments to justfy our theoretical results.  When we are able to revise our paper, we will add the detailed experimental results as a separate section in the manuscript, and **the code to reproduce the experiments will be added to the Github**.
>
> We conduct experiments on three problems, the 2D Poisson equation with reference solution $u _{ref}=\sin(\pi x)\sin(\pi y)$, the 1D Heat equation with reference solution $u _{ref}=e^{-\frac{\pi^2t}{4}}\sin(\pi x)$ and the 2D Helmholtz equation with wave number $k=4$, and the reference solution $u _{ref}=\sin(\pi x)\sin(k\pi y)$.
> All codes are conducted by the Pytorch framework. The configurations used in these examples are listed in Table 1. We report the relative $L^2$-error of the NGD optimizer, the SGD optimizer, the Adam optimizer, and the L-BFGS optimizer in Table 2.
> The relative $L^2$-error is defined as follows:
> $$ \frac{||\hat{u}-u _{ref}|| _2}{||u _{ref}|| _2},$$
> where $\hat{u}$ denotes the predicted solution and $u _{ref}$ represents the reference solution. $N _f$ is the number of interior sampling points, and $N _b$ is the number of boundary sampling points.
>
> **Table 1: Configurations of Different Equations**
>
> |    | $N _f $ | $N _b$ | batch size | hidden layers | hidden neurons | activation function |
> |---|---|---|---|---|---|---|
> | 2D Poisson   | 1000    | 200    | 100    | 1    | 128    | tanh(·)    |
> | 1D Heat   | 1000    | 200    | 100    | 1    | 128    | tanh(·)    |
> | 2D Helmholtz | 1000    | 200    | 100    | 1    | 128    | tanh(·)    |
>
>
> **Table 2: Relative $L^2$-error of Different Optimizers**
>
> |    | SGD    | Adam    | L-BFGS | NGD    |
> |---|---|---|---|---|
> | 2D Poisson   | 1.45e-01    | 5.32e-03    | 3.17e-03    | **1.12e-04**  |
> | 1D Heat   | 5.43e-01    | 6.91e-03    | 4.98e-03    | **3.42e-04**    |
> | 2D Helmholtz | 8.48e+00    | 1.06e+00    | 3.35e+00    | **6.67e-03** |
>
> In all the experiments，we run the NGD and L-BFGS method for 500 epochs, while the SGD and Adam are trained for 10, 000 epochs. The loss decay during training demonstrating that the NGD method converges significantly faster than other optimization methods.
>
> Table 3 presents the convergence performance of the NGD method with different learning rates on the 2D Poisson equation. The experimental results demonstrate that NGD maintains stable convergence across a wide range of learning rates without significant degradation in final accuracy.
>
> **Table 3: Relative $ L^2 $-error Comparison Across Different Learning Rates for NGD method**
>
> | Learning Rate       |   0.5   |   0.1   |  0.05   |  0.01   | 0.005   | 0.001   |
> |---|---|---|---|---|---|---|
> | Relative $L^2$-error | 1.18e-03 | 3.24e-04 | 1.87e-04 | 1.12e-04 | 1.22e-04 | 1.68e-04 |
>
> In addition, a comparative analysis of the model performance is performed with progressively increasing network widths. Table 4 presents the variation of $L^2$-error with respect to network width  for 1D Poisson equation with $u_{ref}=\sin(4\pi x)$. The results demonstrate that increasing network width leads to accuracy improvements.
>
> **Table 4: Relative $ L^2$-error Comparison Across Different Network Width for NGD method**
>
> | Width $m$       |   20   |   40   |   80   |  160  |  320  |  640  |  1280  |  2560  |
> |---|---|---|---|---|---|---|---|---|
> | Relative $L^2$-error | 1.59e-03 | 7.21e-04 | 5.18e-04 | 3.8e-04 | 3.08e-04 | 2.76e-04 | 1.78e-04 | 7.05e-05 |
>
> From an experimental perspective, NGD demonstrates rapid convergence during the training process. Compared to other optimization algorithms, it requires significantly fewer epochs to converge. Furthermore, the experimental results illustrate the strong robustness of the NGD method with respect to hyperparameter selection. Therefore, the empirical findings validate our theoretical conclusions.

---

### Official Review · Reviewer_4Wzk · 2025-03-11

**Overall Recommendation:** 3

**Summary:**

The manuscript concerns convergence results for PINNs for shallow neural networks with $\operatorname{ReLU}^3$ (or certain smooth) activation functions in the overparametrized setting. Both gradient descent and natural gradient descent are considered. The considered model PDE is a heat equation.

## update after rebuttal
I maintain my score. The authors claimed that the natural gradient they are considering is different from Gauss-Newton's method, which is incorrect. I pointed this out, but never received an answer.

**Claims And Evidence:**

Full proofs for every statement are provided.

**Essential References Not Discussed:**

I am not aware of any.

**Experimental Designs Or Analyses:**

Not applicable.

**Methods And Evaluation Criteria:**

Not applicable.

**Other Comments Or Suggestions:**

My main concern with the article is the lack of simulations.

**Other Strengths And Weaknesses:**

The article would greatly benefit from simulations, especially for the natural gradient part. From a practioners point of view, it is important to know to what extent theory and practice meet. Even simple experiments in one spatial dimension would already be interesting (this keeps the scaling with the dimension d under control). Examining  equation (23) shows that the cost of NG is manageable: The Jacobians are of size $n \times p$, where $n = n_1 + n_2$ is the sample size and $p = m(d+2)$ is the number of trainable parameters. Consequently, assuming $n > p$ the dominating cost is $\mathcal O(n^2p)$ to compute $J\cdot J^\top$ which should be feasible for $p$ in the millions and $n$ in the hundreds — which is already quite interesting.

**Questions For Authors:**

The authors frequently compare to Gao et al. when discussing their results and the improved  constants. The cited results in Gao do not contain a dependence on the dimension $d$ of the computational domain. Can the authors comment on this? Is this dependence hidden in the constants of Gao et al?

Can the authors comment on the dependence of the results on the norm of the right-hand side $f$? A complicated right-hand side — for example oscillatory — leads to a complicated solution. It is well known that PINNs can struggle in such situations.

**Relation To Broader Scientific Literature:**

The discussion of the recent literature on convergence results in the NTK regime seems reasonable, although I am no expert in the area. In the broader context, some contextualization may help:

- The form of natural gradient descent considered here is exactly the classical Gauss-Newton method. Including a short remark reminding readers of this fact should be considered.

- The analyzed method agrees with energy natural gradients as proposed by Müller and Zeinhofer **in the case of linear PDEs**. As this is also the setting considered in this paper it can be mentioned.

**Theoretical Claims:**

I checked the proof strategy but I am not familiar enough with the mathematical machinery to certify the correctness of the proofs.

---

> ### Author Rebuttal · Authors · 2025-03-30
>
> We sincerely appreciate the reviewer's time and valuable feedback on our manuscript.  Let us address your questions point by point.
>
> **Q1: Relations to other methods.**
>
> **A1**:  We apologize for the insufficient background discussion of Natural Gradient Descent (NGD) in our work. Although NGD shares some similarities with the classical Gauss-Newton method, key differences exist. For instance, in our problem, the Gauss-Newton iteration is given by
> $$w(k+1)=w(k)-(J(k)^T J(k))^{-1} J(k)^T u(k),$$
> whereas the NGD iteration follows
> $$w(k+1)=w(k)-\eta J(k)^T (J(k) J(k)^T )^{-1} u(k).$$
>
> Müller and Zeinhofer proposed ENGD for PINNs and empirically demonstrated its ability to yield highly
> accurate solutions.  However, NGD formulation differs from ENGD, and they did not provide theoretical convergence guarantees.
>
> **Q2: Lack of simulations.**
>
> **A2**: Thanks for the suggestion. We provide some experiments to validate our theoretical results. In the revised manuscript, we will add a separate section presenting detailed experimental results, and **the code to reproduce the experiments will be added to the Github**.
>
> The configurations used in these examples are listed in Table 1. We report the relative $L^2$-error of NGD, SGD, Adam and L-BFGS optimizers in Table 2. $N _f$ is the number of interior sampling points, and $N _b$ is the number of boundary sampling points.
>
> **Table 1: Configurations of Different Equations**
>
> |    | $ N _f$ | $N _b$ | batch size | hidden layers | hidden neurons | activation function |
> |---|---|---|---|---|---|---|
> | 2D Poisson   | 1000    | 200    | 100    | 1    | 128    | tanh(·)    |
> | 1D Heat   | 1000    | 200    | 100    | 1    | 128    | tanh(·)    |
> | 2D Helmholtz | 1000    | 200    | 100    | 1    | 128    | tanh(·)    |
>
>
> **Table 2: Relative $L^2$-error of Different Optimizers**
>
> |    | SGD    | Adam    | L-BFGS | NGD  |
> |---|---|---|---|---|
> | 2D Poisson   | 1.45e-01    | 5.32e-03    | 3.17e-03    | **1.12e-04**  |
> | 1D Heat   | 5.43e-01    | 6.91e-03    | 4.98e-03    | **3.42e-04**    |
> | 2D Helmholtz | 8.48e+00    | 1.06e+00    | 3.35e+00    | **6.67e-03** |
>
> In all the experiments，we run the NGD and L-BFGS method for 500 epochs, while the SGD and Adam are trained for 10, 000 epochs. The loss decay during training demonstrating that the NGD method converges significantly faster than other optimization methods.
>
> Table 3 presents the convergence performance of the NGD method with different learning rates on the 2D Poisson equation. The experimental results demonstrate that NGD maintains stable convergence across a wide range of learning rates without significant degradation in final accuracy.
>
> **Table 3: Relative $ L^2 $-error Comparison Across Different Learning Rates for NGD method**
> |    |     |       |       |       |       |       |
> |-------|-------|-------|-------|-------|-------|-------|
> | Learning Rate                | 0.5   | 0.1   | 0.05  | 0.01  | 0.005 | 0.001 |
> | Relative $L^2$-error           | 1.18e-03 | 3.24e-04 | 1.87e-04 | 1.12e-04 | 1.22e-04 | 1.68e-04 |
>
> In addition, a comparative analysis of the model performance is performed
> with progressively increasing network widths. Table 4 presents the variation of $L^2$-error with respect to network
> width for 1D Poisson equation with $u_{ref}=\sin(4\pi x)$. The results demonstrate that increasing network width leads to accuracy improvements.
>
> **Table 4: Relative $ L^2$-error Comparison Across Different Network Width for NGD method**
> |  |       |       |       |      |       |       |        |        |
> |------|-------|-------|-------|------|-------|-------|--------|--------|
> | Width $m $   | 20    | 40    | 80    | 160  | 320   | 640   | 1280   | 2560   |
> | Relative $L^2 $-error    | 1.59e-03 | 7.21e-04 | 5.18e-04 | 3.8e-04 | 3.08e-04 | 2.76e-04 | 1.78e-04 | 7.05e-05 |
>
> **Q3: Dependence on $d$.**
>
> **A3**: The dimension-related terms in Gao's results are implicitly hidden in their outcomes. However, our results outperform theirs. For instance, when estimating the initial value $L(0)$, for one of the terms $||w|| _2^2\sigma(w^Tx)$, Gao employed the bounded differences inequality, specifically applying $| ||w|| _2^2\sigma(w^Tx)|\lesssim ||w|| _2^3\sim d^{3/2}$. This leads to their estimate of $L(0)$ being at least $L(0)=\Omega(d^3)$. In contrast, our analysis, by leveraging Lemma C.2, achieves a better bound of $L(0)=\mathcal{O}(d^2)$. In their analysis, similar bounding techniques were employed in other parts as well, thus our results concerning the dimension $d$ are superior to theirs.
>
> **Q4: Dependence on $f$.**
>
> **A4**: From the analysis, we can see that the initial value $L(0)$ depends linearly on $f$, and the requirement for the width $m$ is $m=\Omega(L(0))$. Therefore, a complicated $f$ leads to a higher requirement on $m$, making convergence more challenging.

---

### Official Review · Reviewer_t3g2 · 2025-03-13

**Overall Recommendation:** 3

**Summary:**

The paper investigates the theoretical convergence of natural gradient descent for overparameterized physics-informed neural networks under NTK regime. The paper extends the previous works on gradient descent to natural gradient descent, with improved bounds. The improvement is based on some inequality techniques adopted and the inherent benefits of natural gradient descent over the vanilla gradient descent. The mathematical proof is overall correct, and the study is meaningful for understanding and improving the training of PINNs.


## update after rebuttal
I expect authors can include experiments/implementation details, discussions, and strong motivations of the study, since NGD is not popular in PINNs training.
Since my score has already been positive, I will keep it. I think the paper should be accepted.

**Claims And Evidence:**

Yes

**Essential References Not Discussed:**

NA

**Experimental Designs Or Analyses:**

No experiments in the paper. But I think experimental investigations on natural gradient descent for PINNs is necessary. Otherwise, the analysis conducted in the paper does not make sense.

**Methods And Evaluation Criteria:**

The performance of natural gradient descent for PINNs has not been well investigated. Researchers prefer to use gradient-based methods (e.g., gradient descent, SGD, Adam) or quasi-Newton's methods (e.g., LBFGS) to train PINNs. The natural gradient descent is rarely adopted in PINNs literature. Therefore, I am concerned about the applicability of the optimization algorithm studied in this paper.

**Other Comments Or Suggestions:**

NA

**Other Strengths And Weaknesses:**

The motivation for the analysis is not strong, since the algorithm (natural gradient descent) is never adopted for training PINNs. I think experimental investigations on natural gradient descent for PINNs is required to support your motivation and analysis. For example, natural gradient descent is exactly outperforming other methods, which is somewhat supported by your theoretical results. Does the convergence results improve with wider networks? Therefore, I think this paper's motivation and theoretical results are not supported and accompanied by some experimental evidence.

**Questions For Authors:**

Why do you consider natural gradient descent, which is rarely applied for training PINNs? Why not consider LBFGS, which I think is more interesting.

**Relation To Broader Scientific Literature:**

NA

**Theoretical Claims:**

The proof should be correct.

---

> ### Author Rebuttal · Authors · 2025-03-30
>
> We sincerely appreciate the reviewer's time and valuable feedback on our manuscript.  Let us address your questions point by point.
>
> **Q1: Experimental investigations on NGD for PINNs.**
>
> **A1**: Thanks for the suggestion. We provide some experiments to validate our theoretical results. In the revised manuscript, we will add a separate section presenting detailed experimental results, and **the code to reproduce the experiments will be added to the Github**.
>
> The configurations used in these examples are listed in Table 1. We report the relative $L^2$-error of NGD, SGD, Adam and L-BFGS optimizers in Table 2. $N _f$ is the number of interior sampling points, and $N _b$ is the number of boundary sampling points.
>
> **Table 1: Configurations of Different Equations**
>
> |    | $ N _f$ | $N _b$ | batch size | hidden layers | hidden neurons | activation function |
> |---|---|---|---|---|---|---|
> | 2D Poisson   | 1000    | 200    | 100    | 1    | 128    | tanh(·)    |
> | 1D Heat   | 1000    | 200    | 100    | 1    | 128    | tanh(·)    |
> | 2D Helmholtz | 1000    | 200    | 100    | 1    | 128    | tanh(·)    |
>
>
> **Table 2: Relative $L^2$-error of Different Optimizers**
>
> |    | SGD    | Adam    | L-BFGS | NGD  |
> |---|---|---|---|---|
> | 2D Poisson   | 1.45e-01    | 5.32e-03    | 3.17e-03    | **1.12e-04**  |
> | 1D Heat   | 5.43e-01    | 6.91e-03    | 4.98e-03    | **3.42e-04**    |
> | 2D Helmholtz | 8.48e+00    | 1.06e+00    | 3.35e+00    | **6.67e-03** |
>
> In all the experiments，we run the NGD and L-BFGS method for 500 epochs, while the SGD and Adam are trained for 10, 000 epochs. The loss decay during training demonstrating that the **NGD method converges significantly faster than other optimization methods.**
>
> Table 3 presents the convergence performance of the NGD method with different learning rates on the 2D Poisson equation. The experimental results demonstrate that **NGD maintains stable convergence across a wide range of learning rates without significant degradation in final accuracy**.
>
> **Table 3: Relative $ L^2 $-error Comparison Across Different Learning Rates for NGD method**
> |    |     |       |       |       |       |       |
> |-------|-------|-------|-------|-------|-------|-------|
> | Learning Rate                | 0.5   | 0.1   | 0.05  | 0.01  | 0.005 | 0.001 |
> | Relative $L^2$-error           | 1.18e-03 | 3.24e-04 | 1.87e-04 | 1.12e-04 | 1.22e-04 | 1.68e-04 |
>
> In addition, a comparative analysis of the model performance is performed
> with progressively increasing network widths. Table 4 presents the variation of $L^2$-error with respect to network
> width for 1D Poisson equation with $u_{ref}=\sin(4\pi x)$. The results demonstrate that **increasing network width leads to accuracy improvements** .
>
> **Table 4: Relative $ L^2$-error Comparison Across Different Network Width for NGD method**
> |  |       |       |       |      |       |       |        |        |
> |------|-------|-------|-------|------|-------|-------|--------|--------|
> | Width $m $   | 20    | 40    | 80    | 160  | 320   | 640   | 1280   | 2560   |
> | Relative $L^2 $-error    | 1.59e-03 | 7.21e-04 | 5.18e-04 | 3.8e-04 | 3.08e-04 | 2.76e-04 | 1.78e-04 | 7.05e-05 |
>
>
> **Q2: The influence of width for convergence results.**
>
> **A2:** Theoretically, the algorithm achieves guaranteed convergence once the network width exceeds a sufficient threshold, with no further improvement in convergence rate from additional width increases. This fact is consistent with established results on gradient descent for PINNs and regression problems. Empirically, wider networks exhibit lower training and test errors.
>
> **Q3: Why consider NGD?**
>
> **A3**: From a theoretical perspective, existing convergence analyses of optimization algorithms for PINNs have primarily focused on gradient descent. However, as shown in this paper's improvements on gradient descent, gradient descent imposes relatively stringent requirements on certain hyperparameters (e.g., learning rate) and exhibits slow convergence rates. Therefore, we turned our attention to NGD, which we think may account for curvature information of the function. Theoretically, NGD has more relaxed requirements for the learning rate and achieves faster convergence rates. Experiments also demonstrate that NGD converges more rapidly during training.
>
> As for why L-BFGS was not considered, it is because for non-convex optimization problems like PINNs, L-BFGS is more complex and harder to analyze compared to NGD. Given that recent studies have shown quasi-Newton methods (e.g., SOAP [1]) perform well in optimizing PINNs, analyzing the convergence of such algorithms for PINNs will be an important future direction. Additionally, more efficient implementations or variants of NGD also represent a key area for future research.
>
> [1]: Vyas N, Morwani D, Zhao R, et al. Soap: Improving and stabilizing shampoo using adam[J]. arXiv preprint arXiv:2409.11321, 2024.

---

> > ### Comment · Reviewer_t3g2 · 2025-04-02
> >
> > Thank you so much for adding experiments comparing Adam, LBFSG, and NGD. The experimental results show the superior performances of NGD, which supports the motivation of this paper to investigate the theoretical convergence of NGD for PINNs. I am just curious about one more thing: how do you deal with the inverse of the Gram matrix? It should be extremely high-dimensional and singular. I don't think the vanilla NGD works well in practice. What kind of practical tricks do you adopt? Thank you.

---

> > > ### Author Response · Authors · 2025-04-02
> > >
> > > Thanks for the very timely comment. We can understand your concern about the computing of the Gram matrix.
> > > However, for the Gram matrix in NGD, we should note that it differs from the classical Gauss-Newton method.
> > > **Practical tricks can make computing the inverse of the Gram matrix manageable and stable**. We explain it as follows.
> > >
> > > In our problem, the classical Gauss-Newton iteration is given by $$w(k+1)=w(k)-(J(k)^T J(k))^{-1} J(k)^T u(k),$$ whereas the NGD iteration follows $$w(k+1)=w(k)-\eta J(k)^T (J(k) J(k)^T )^{-1} u(k).$$
> > > Here $\eta$ is the learning rate, the Jacobian $J(k)\in R^{n\times p}$, where $n=n_1+n_2$ is the sample data size and $p=m(d+2)$ is the neural network parameters, as given by Equation (23) in our paper.
> > > The dominating computing cost is the inverse of $J\cdot J^T \in R^{n\times n}$, which can be manageable and stable for small $n$ and large $p$ (it is also noticed by Reviewer 4Wzk).
> > > In practice, we apply **stochastic batch technique to choose small sample size $n$ in every iteration, and large network papameters $p$ to make $J\cdot J^T$ invertible** (we also use torch.pinverse($J\cdot J^T$) in the code). $$
> > >
> > > For example, for the 2D Poisson equation in above Table 2, the size in $J$ is $n=100$ and $p=128*(2+2)=512$, and it is enough to make the computing stable. Much larger $p=m(d+2)$ is also given in above Table 4, shows accuracy improved when network parameter $p$ incresed.
> > >
> > > We note that for the classical Gauss-Newton iteration, the Gram matrix is $J^T\cdot J \in R^{p\times p}$.
> > > For larger network parameters $p$, the Gram matrix of Gauss-Newton method will be extremely high-dimensional and singular as you said.
> > > However, this does not occur in our NGD method.
> > >
> > > Finally, we believe that practical implementation techniques for NGD in PINNs—similar to the classical Newton method's evolution into L-BFGS—require further investigation and will be a focus of our future work.
> > >
> > > We would be most grateful for your input if any additional revisions are needed during the remaining time of the discussion period.

---

### Official Review · Reviewer_XQEe · 2025-03-14

**Overall Recommendation:** 4

**Summary:**

The paper investigates the convergence properties of gradient descent (GD) and natural gradient descent (NGD) for training two-layer Physics-Informed Neural Networks (PINNs). The authors improve the learning rate of GD from $\mathcal{O}(\lambda_0)$ to $\mathcal{O}(1/\|H^{\infty}\|_2)$, where $\lambda_0$ is the least eigenvalue of the Gram matrix $H^{\infty}$, leading to faster convergence. They also establish the positive definiteness of Gram matrices for various smooth activation functions, such as logistic, softplus, and hyperbolic tangent, applicable to a wide range of PDEs. The paper demonstrates that NGD can achieve a learning rate of $\mathcal{O}(1)$, resulting in a convergence rate independent of the Gram matrix, with quadratic convergence for smooth activation functions. By introducing a new recursion formula for GD, the authors reduce the requirements on learning rate and network width, improving convergence results. The study highlights the advantages of NGD over GD, showing faster convergence rates and less stringent network width requirements, making it a promising approach for efficiently training PINNs in scientific computing applications involving PDEs.

**Claims And Evidence:**

Yes.

**Essential References Not Discussed:**

No specific related work not discussed.

**Experimental Designs Or Analyses:**

N/A.

**Methods And Evaluation Criteria:**

N/A

**Other Comments Or Suggestions:**

No.

**Other Strengths And Weaknesses:**

The current paper is limited to a convergence analysis, based on the improvement proposed in the paper, it may conclude a method to guide the training of PINNs in practice. However, we don't see such a method or guidance.

**Questions For Authors:**

Could the authors propose a method that helps the design of PINNs?

Can the authors conduct experiments to verify the correctness in practice?

**Relation To Broader Scientific Literature:**

The key contributions of the paper are closely related to several strands of prior research in the broader scientific literature, particularly in the fields of optimization, neural networks, and PINNs.

The paper builds on prior work that demonstrates the convergence of GD for over-parameterized neural networks. Specifically, it extends findings from Du et al. (2018, 2019) and Gao et al. (2023), which show that GD can achieve zero training loss under over-parameterization. The authors improve upon these results by providing a better learning rate and milder requirements on network width.

**Theoretical Claims:**

Yes.

---

> ### Author Rebuttal · Authors · 2025-03-30
>
> We sincerely thank the reviewer for their valuable time and constructive suggestions on our work.
>
> **Q1: Methods to help the PINNs' design.**
>
> **A1:** Thanks for the kind suggestion. The results of the paper motivate us to establish "good" Gram matrix $H^{\infty}$ to reduce the strictly learning rate requirement for GD optimizers, and this can be achieved by properly design the loss function, the network architecture, and so on.
> For example, since the learning rate requirement for GD is related to $\mathcal{O}(1/||H^{\infty}||_2)$, existing works [1] have adaptively adjusted the weight in the different loss components of PINNs, to improve the eigenvalue distribution of the Gram matrix $H^{\infty}$, thus we can use a normal learning rate to accelerate PINNs' training and convergence.
>
> [1]Wang S, Yu X, Perdikaris P. When and why PINNs fail to train: A neural tangent kernel perspective[J]. Journal of Computational Physics, 2022, 449: 110768.
>
> **Q2: Add experiments to verify theoretical findings.**
>
> **A2:** Thanks for the suggestion. Here, we provide some experiments to validate our theoretical results. In the revised version, we will add the detailed experimental results as a separate section, and **the code to reproduce the experiments will be added to the Github**.
>
> We conduct experiments on three problems, the 2D Poisson equation with reference solution $u_{ref}=\sin(\pi x)\sin(\pi y)$, the 1D Heat equation with reference solution $u_{ref}=e^{-\frac{\pi^2 t}{4}}\sin(\pi x)$, and the 2D Helmholtz equation with wave number
> $k=4$, and the reference solution $u_{ref}=\sin(\pi x)\sin(k\pi y)$.
> All codes are conducted by the Pytorch framework. The configurations used in these examples are listed in Table 1. We report the relative $L^2$-error of the NGD optimizer, the SGD optimizer, the Adam optimizer, and the L-BFGS optimizer in Table 2.
> The relative $L^2$-error is defined as follows:
> $$ \frac{||\hat{u}-u _{ref}|| _2}{||u _{ref}|| _2},$$
> where $\hat{u}$ denotes the predicted solution and $u _{ref}$ represents the reference solution. $N _f$ is the number of interior sampling points  and $N _b$ is the number of boundary sampling points.
>
> **Table 1: Configurations of Different Equations**
>
> |    | $ N _f$ | $N _b$ | batch size | hidden layers | hidden neurons | activation function |
> |---|---|---|---|---|---|---|
> | 2D Poisson   | 1000    | 200    | 100    | 1    | 128    | tanh(·)    |
> | 1D Heat   | 1000    | 200    | 100    | 1    | 128    | tanh(·)    |
> | 2D Helmholtz | 1000    | 200    | 100    | 1    | 128    | tanh(·)    |
>
>
> **Table 2: Relative $L^2$-error of Different Optimizers**
>
> |    | SGD    | Adam    | L-BFGS | NGD    |
> |---|---|---|---|---|
> | 2D Poisson   | 1.45e-01    | 5.32e-03    | 3.17e-03    | **1.12e-04**  |
> | 1D Heat   | 5.43e-01    | 6.91e-03    | 4.98e-03    | **3.42e-04**    |
> | 2D Helmholtz | 8.48e+00    | 1.06e+00    | 3.35e+00    | **6.67e-03** |
>
> In all the experiments，we run the NGD and L-BFGS method for 500 epochs, while the SGD and Adam are trained for 10, 000 epochs. The loss decay during training demonstrating that the NGD method converges significantly faster than other optimization methods.
>
> Table 3 presents the convergence performance of the NGD method with different learning rates on the 2D Poisson equation. The experimental results demonstrate that NGD maintains stable convergence across a wide range of learning rates without significant degradation in final accuracy.
>
> **Table 3: Relative $ L^2 $-error Comparison Across Different Learning Rates for NGD method**
>
> | Learning Rate       |   0.5   |   0.1   |  0.05   |  0.01   | 0.005   | 0.001   |
> |---|---|---|---|---|---|---|
> | Relative $L^2$-error | 1.18e-03 | 3.24e-04 | 1.87e-04 | 1.12e-04 | 1.22e-04 | 1.68e-04 |
>
> In addition, a comparative analysis of the model performance is performed with progressively increasing network widths. Table 4 presents the variation of $L^2$-error with respect to network width  for 1D Poisson equation with $u_{ref}=\sin(4\pi x)$. The results demonstrate that increasing network width leads to accuracy improvements.
>
> **Table 4: Relative $ L^2$-error Comparison Across Different Network Width for NGD method**
>
> | Width $m$       |   20   |   40   |   80   |  160  |  320  |  640  |  1280  |  2560  |
> |---|---|---|---|---|---|---|---|---|
> | Relative $L^2$-error | 1.59e-03 | 7.21e-04 | 5.18e-04 | 3.8e-04 | 3.08e-04 | 2.76e-04 | 1.78e-04 | 7.05e-05 |
>
> From an experimental perspective, NGD demonstrates rapid convergence during the training process. Compared to other optimization algorithms, it requires significantly fewer epochs to converge. Furthermore, the experimental results illustrate the strong robustness of the NGD method with respect to hyperparameter selection. Therefore, the empirical findings validate our theoretical conclusions.

---

### Decision · Program_Chairs · 2025-05-01

**Decision:**

Reject

**Comment:**

This paper considers training two-layer PINNs in the NTK regime. The paper extends previous work on gradient descent to natural gradient descent and shows improved convergence bounds. The reviews are generally positive. However, there are several weaknesses identified during the review process:
- Limited novelty: The theoretical contribution is perceived as an incremental extension of existing NTK analysis techniques to NGD within the PINN context, rather than a substantial conceptual advance. The analysis is fairly straightforward given the literature.
- Weak motivation & practical relevance: The paper's motivation is unclear, as NGD is rarely used in practice for training PINNs. The justification for studying this specific optimizer over more prevalent methods (like Adam or L-BFGS) was weak in the original submission, and relying on post-hoc experimental results from the rebuttal doesn't fully address this issue.
- Lack of empirical validation in submission: The paper was initially submitted without any experiments to support its theoretical claims about convergence rates and optimizer performance. This is a major shortcoming for work aiming to provide insights into practical optimization algorithms. Even with the added experiments, Reviewer 4Wzk noted the need for a stronger connection between the empirical results and the specific theoretical claims (e.g., how the experiments directly validate the derived rates or dependencies).
- Unresolved technical clarity: A question was raised regarding the equivalence of the paper's NGD formulation to the Gauss-Newton method in the studied setting. Reviewer 4Wzk noted the authors seemed to misunderstand this relationship and failed to respond to their final clarification, raising concerns about the paper's technical precision and positioning.

If the paper is not accepted, I suggest the authors include strong empirical contributions in the revision.